# LoRanPAC: Low-rank Random Features and Pre-trained Models for Bridging Theory and Practice in Continual Learning

**Liangzu Peng**
University of Pennsylvania
lpenn@seas.upenn.edu

**Juan Elenter Litwin** [1]
Spotify
juane@spotify.com

**Joshua Agterberg**
University of Illinois Urbana-Champaign
jagt@illinois.edu

**Alejandro Ribeiro**
University of Pennsylvania
aribeiro@seas.upenn.edu

**René Vidal**
University of Pennsylvania
vidalr@seas.upenn.edu

## Abstract

The goal of continual learning (CL) is to train a model that can solve multiple tasks presented sequentially. Recent CL approaches have achieved strong performance by leveraging large pre-trained models that generalize well to downstream tasks. However, such methods lack theoretical guarantees, making them prone to unexpected failures. Conversely, principled CL approaches often fail to achieve competitive performance. In this work, we aim to bridge this gap between theory and practice by designing a simple CL method that is theoretically sound and highly performant. Specifically, we lift pre-trained features into a higher dimensional space and formulate an over-parametrized minimum-norm least-squares problem. We find that the lifted features are highly ill-conditioned, potentially leading to large training errors (numerical instability) and increased generalization errors. We address these challenges by continually truncating the singular value decomposition of the lifted features. Our approach, termed LoRanPAC, is stable with respect to the choice of hyperparameters, can handle hundreds of tasks, and outperforms state-of-the-art CL methods on multiple datasets. Importantly, our method satisfies a recurrence relation throughout its continual learning process, which allows us to prove it maintains small training and test errors by appropriately truncating a fraction of SVD factors. This results in a stable continual learning method with strong empirical performance and theoretical guarantees. Code available: https://github.com/liangzu/loranpac.

## 1 Introduction

*Continual learning* (CL) requires training a model that performs well on multiple tasks presented sequentially. A primary challenge in CL is acquiring new knowledge without causing *catastrophic forgetting* (i.e., substantial performance degradation on previously learned tasks). However, the gap that prevails in the CL literature, as we review in Section 5 and Appendix H, is that theoretically grounded CL methods tend to be impractical (Evron et al., 2022; Peng & Risteski, 2022; Peng et al., 2023; Cai & Diakonikolas, 2024), while highly performant methods involve solving intricate, non-convex training problems, for which deriving informative theoretical guarantees is challenging (Wang et al., 2022b;c;a; Smith et al., 2023; Wang et al., 2023; Jung et al., 2023; Tang et al., 2023; Gao et al., 2024b; Roy et al., 2024; Kim et al., 2024). In this paper, we aim to bridge the gap by designing a simple CL method that is both theoretically grounded and highly performant.

---

[1]Work done while at the University of Pennsylvania.

Towards this goal, we consider learning downstream image classification tasks continually with large pre-trained models, which are widely available nowadays. As their weights are typically frozen, they provide highly generalizable features that significantly boost performance with little computational overhead (Wang et al., 2022d;a; McDonnell et al., 2023; Zhou et al., 2024a). Crucially, pre-trained models simplify network design, as concatenating a pre-trained model with a shallow trainable network often attains competitive performance (Zhou et al., 2023; McDonnell et al., 2023).

Motivated by the *RanPAC* method of McDonnell et al. (2023), we use a shallow trainable network that is now commonly known as the *random feature model*. With this network, we lift pre-trained features into a higher dimensional space and then train a linear classifier on the lifted features simply by least-squares fitting. Yet, the lifted features are *double-edged*: while they tend to boost performance by increasing feature separability (Telgarsky, 2022; Min et al., 2024), they are also highly ill-conditioned, making it computationally difficult to train the linear classifier. For example, the ill-conditioned features would decelerate the convergence of gradient-based methods such as *Orthogonal Gradient Descent* (OGD) and *PCA-OGD* (Farajtabar et al., 2020; Doan et al., 2021). Also due to ill-conditioning, the implementation of the *Ideal Continual Learner* (ICL) (Peng et al., 2023) based on incremental *Singular Value Decomposition* (SVD) will be numerically unstable. While RanPAC alleviates the numerical instability by using ridge regression, its performance is sensitive to the choice of the regularization parameter, which can make it ill-suited for long task sequences.

We identify that the ill-conditioning and instability arise as more tasks are observed and then the smallest singular values of the lifted features plummet, while the largest singular values remain almost constant. This finding motivates our method, termed *LoRanPAC*, which truncates these smallest singular values prior to least-squares fitting. LoRanPAC bridges the gap between theory and practice by delivering stable and strong performance with theoretical guarantees. Concretely:

• We provide a continual implementation of LoRanPAC to train an over-parameterized linear classifier with highly ill-conditioned features in a numerically stable fashion (Section 3). We show it is more stable, more scalable, and more efficient than RanPAC.

• We derive theoretical guarantees for LoRanPAC, proving that it has small training and test errors when a suitable fraction of SVD factors are truncated (Theorems 1 and 2, Section 4). These results stem from a non-trivial recurrence relation that allows us to capture the continual learning dynamics of LoRanPAC (Lemma 1, Appendix D).

• We conduct extensive experiments on multiple datasets, showing that LoRanPAC uniformly outperforms prior works and specifically RanPAC (Section 6). Thanks to our stable implementation, LoRanPAC outmatches RanPAC by a significant margin in the CIL setting with one class given at a time (Inc-1), where hundreds of tasks (classes) are sequentially presented (Table 2).

## 2 TECHNICAL BACKGROUND

**Problem Setting.** We consider classification tasks in the *class-incremental learning* (CIL) setting, where each incoming task contains only unseen classes. Following conventions (Yan et al., 2021; Zhou et al., 2023), we write B-$q_1$, Inc-$q_2$ to mean that the model is given $q_1$ classes in the first task and then $q_2$ classes in each of the subsequent tasks ($q_1 = 0$ means all tasks have $q_2$ distinct classes). We use *vision transformers* (ViTs) of Dosovitskiy et al. (2021) as pre-trained models.

**Pretrained Features and Labels.** Given $m_t$ images of task $t$, we feed them to pre-trained ViTs, obtaining the output features $\boldsymbol{X}_t \in \mathbb{R}^{d \times m_t}$. Here, $d$ is the feature dimension ($d = 768$ in the ViTs used). Corresponding to $\boldsymbol{X}_t$ is the label matrix $\boldsymbol{Y}_t \in \mathbb{R}^{c_t \times m_t}$. Every column of $\boldsymbol{Y}_t$ is a *one-hot vector*, that is some standard basis vector in $\mathbb{R}^{c_t}$, where $c_t$ is the total number of classes observed so far. We thus have $c_1 \leq \cdots \leq c_i \leq \cdots \leq c_t$. Let $M_t := m_1 + \cdots + m_t$. While $\boldsymbol{Y}_i \in \mathbb{R}^{c_i \times m_t}$ might have a different number of rows as $c_i$ varies, one can pad $c_t - c_i$ zero rows to $\boldsymbol{Y}_i$ when new class information is revealed; so, with a slight abuse of notation, $\boldsymbol{Y}_i$ is viewed as having $c_t$ rows. We denote by $\boldsymbol{Y}_{1:t}$ the label matrix of the first $t$ tasks: $\boldsymbol{Y}_{1:t} = [\boldsymbol{Y}_1, \ldots, \boldsymbol{Y}_t] \in \mathbb{R}^{c_t \times M_t}$.

**Random ReLU Features.** Let relu : $\xi \mapsto \max\{0, \xi\}$ be a ReLU layer and $\boldsymbol{P} \in \mathbb{R}^{E \times d}$ denote a random Gaussian matrix with i.i.d. $\mathcal{N}(0, 1)$ entries; here we assume $E > d$. These allow us to embed $\boldsymbol{X}_t$ into a higher dimensional space and get *random ReLU features* $\boldsymbol{H}_t \in \mathbb{R}^{E \times m_t}$ via

$$\boldsymbol{H}_t := \text{relu}(\boldsymbol{P}\boldsymbol{X}_t), \qquad \boldsymbol{H}_{1:t} := [\boldsymbol{H}_1, \ldots, \boldsymbol{H}_t] \in \mathbb{R}^{E \times M_t}. \qquad (1)$$

Note that relu is a pointwise non-linearity, applied to $\boldsymbol{PX}_t$ entry-wise. The goal is to learn a linear classifier $\boldsymbol{W} \in \mathbb{R}^{c_t \times E}$ continually, using features $\boldsymbol{H}_t$ and labels $\boldsymbol{Y}_t$ of task $t$.

An alternative way to view these setups is through a two-layer neural network

$$\boldsymbol{X} \mapsto \boldsymbol{W} \cdot \text{relu}(\boldsymbol{PX}),$$

where $\boldsymbol{P}$ are randomly generated, then fixed, and $\boldsymbol{W}$ are trainable weights. Networks of this form predate the early work of Schmidt et al. (1992); Pao & Takefuji (1992) and are later known as *extreme learning machines* and *random feature models*. In Appendix H.2 we review these lines of research. In principle, the randomness of $\boldsymbol{P}$ would have some effects on learning $\boldsymbol{W}$, but algorithmically, this randomness is irrelevant as $\boldsymbol{P}$ is fixed after random generation. In the paper we adopt this algorithmic viewpoint, treating $\boldsymbol{H}_{1:t} = \text{relu}(\boldsymbol{PX}_{1:t})$ as fixed, and largely ignore the randomness of $\boldsymbol{P}$.

Table 16 of our appendix shows random ReLU features $\boldsymbol{H}_{1:t}$ boosts the performance (compared to the original pre-trained features $\boldsymbol{X}_{1:t}$, ReLU features $\text{relu}(\boldsymbol{X}_t)$, or randomly embedded features $\boldsymbol{PX}_{1:t}$. Fig. 6 in the appendix furthermore shows that the performance improves as the embedding dimension $E$ increases. These experiments justify the use of random ReLU features in high embedding dimensions (we use $E = 10^5$ unless otherwise specified).

**Minimum Norm Solution and Its Instability.** If $E \gg M_t$, there are infinitely many $\boldsymbol{W}$ satisfying $\boldsymbol{WH}_i = \boldsymbol{Y}_i$ ($\forall i$). Among them, a common choice is the *minimum-norm* solution:

$$\min_{\boldsymbol{W} \in \mathbb{R}^{c_t \times E}} \|\boldsymbol{W}\|_{\text{F}}^2 \qquad \text{s.t.} \qquad \boldsymbol{WH}_i = \boldsymbol{Y}_i, \quad i = 1, \ldots, t, \qquad \text{(Min-Norm)}$$

where $\| \cdot \|_{\text{F}}$ denotes the Frobenius norm. While Min-Norm is an *offline formulation* that assumes the availability of all seen data $\{(\boldsymbol{H}_i, \boldsymbol{Y}_i)\}_{i=1}^t$, it can be implemented in a CL fashion, e.g., via *incremental SVD* (Remark 2). But such an intuitive implementation fails. Fig. 1a plots the eigenvalues of $\boldsymbol{H}_{1:t}^\top \boldsymbol{H}_{1:t}$, showing that $\boldsymbol{H}_{1:t}^\top \boldsymbol{H}_{1:t}$ is highly ill-conditioned: it has just a few largest and smallest eigenvalues, respectively of order $10^{11}$ and $10^{-5}$, outnumbered by the eigenvalues in between that decay more slowly. Fig. 1b plots extreme eigenvalues of $\boldsymbol{H}_{1:t}^\top \boldsymbol{H}_{1:t}$, revealing that the minimum eigenvalue drastically drops after a certain number of tasks. Comparing Fig. 1b, c, d, we see that the training MSE loss $\frac{1}{M_t} \|\boldsymbol{WH}_{1:t} - \boldsymbol{Y}_{1:t}\|_{\text{F}}^2$ explodes up, and the test accuracy plummets, exactly when the smallest eigenvalues (of order $10^{-5}$) emerge and begin to invade the spectrum. In summary, the incremental SVD solution to Min-Norm is unable to handle the highly ill-conditioned features $\boldsymbol{H}_{1:t}$, resulting in numerical errors.

## 3   LoRanPAC: Stable Continual Learning via Low-Rank Random Features

**Offline Formulation.** The numerical evidence collected in Fig. 1 suggests that the instability of Min-Norm relates to the emergence of very small eigenvalues that make $\boldsymbol{H}_{1:t} \in \mathbb{R}^{E \times M_t}$ ill-conditioned. This motivates a simple remedy, called *LoRanPAC* (*offline formulation*), which consists of truncating the smallest singular values (vectors) of $\boldsymbol{H}_{1:t}$ and then solving Min-Norm with its truncated version. More concretely, write the SVD of $\boldsymbol{H}_{1:t}$ as $\sigma_1 \boldsymbol{u}_1 \boldsymbol{v}_1^\top + \cdots + \sigma_{M_t} \boldsymbol{u}_{M_t} \boldsymbol{v}_{M_t}^\top$ with ordered singular values $\sigma_1 \geq \cdots \geq \sigma_{M_t}$. The truncation can then be described with some integer $k_t \in [0, M_t]$ by a function $\tau_{k_t}$ that maps $\boldsymbol{H}_{1:t}$ to $\sigma_1 \boldsymbol{u}_1 \boldsymbol{v}_1^\top + \cdots + \sigma_{k_t} \boldsymbol{u}_{k_t} \boldsymbol{v}_{k_t}^\top$, where $k_t$ is the number of top SVD factors preserved. Since $\tau_{k_t}(\cdot)$ preserves the shape of its input, $\tau_{k_t}(\boldsymbol{H}_{1:t})$ is of the same size $E \times M_t$ as $\boldsymbol{H}_{1:t}$. Applying this idea of truncation to Min-Norm means solving the following program:

$$\overline{\boldsymbol{W}}_t \in \underset{\boldsymbol{W} \in \mathbb{R}^{c_t \times E}}{\arg\min} \|\boldsymbol{W}\|_{\text{F}}^2 \qquad \text{s.t.} \qquad \boldsymbol{W}\tau_{k_t}(\boldsymbol{H}_{1:t}) = \boldsymbol{Y}_{1:t}. \qquad \text{(LoRanPAC)}$$

LoRanPAC is thus named, as it draws inspiration from RanPAC (McDonnell et al., 2023) and leverages low-rank random features. It is safe to assume $k_t \leq \text{rank}(\boldsymbol{H}_{1:t})$, for otherwise the truncation has no effects. For simplicity one might assume $\boldsymbol{H}_{1:t}$ has full rank, that is $\text{rank}(\boldsymbol{H}_{1:t}) = \min\{E, M_t\}$.

**Continual Implementation.** To solve LoRanPAC continually, we first write down the closed-form expression of $\overline{\boldsymbol{W}}_t$. Let $\overline{\boldsymbol{U}}_{1:t} \overline{\boldsymbol{\Sigma}}_{1:t} \overline{\boldsymbol{V}}_{1:t}^\top$ be a *compact* SVD of $\tau_{k_t}(\boldsymbol{H}_{1:t})$; here, $\overline{\boldsymbol{U}}_{1:t}$ is of size $E \times k_t$ and $\overline{\boldsymbol{\Sigma}}_{1:t}$ is invertible of size $k_t \times k_t$. Similarly, let $\boldsymbol{U}_{1:t} \boldsymbol{\Sigma}_{1:t} \boldsymbol{V}_{1:t}^\top$ be a compact SVD of $\boldsymbol{H}_{1:t}$, where

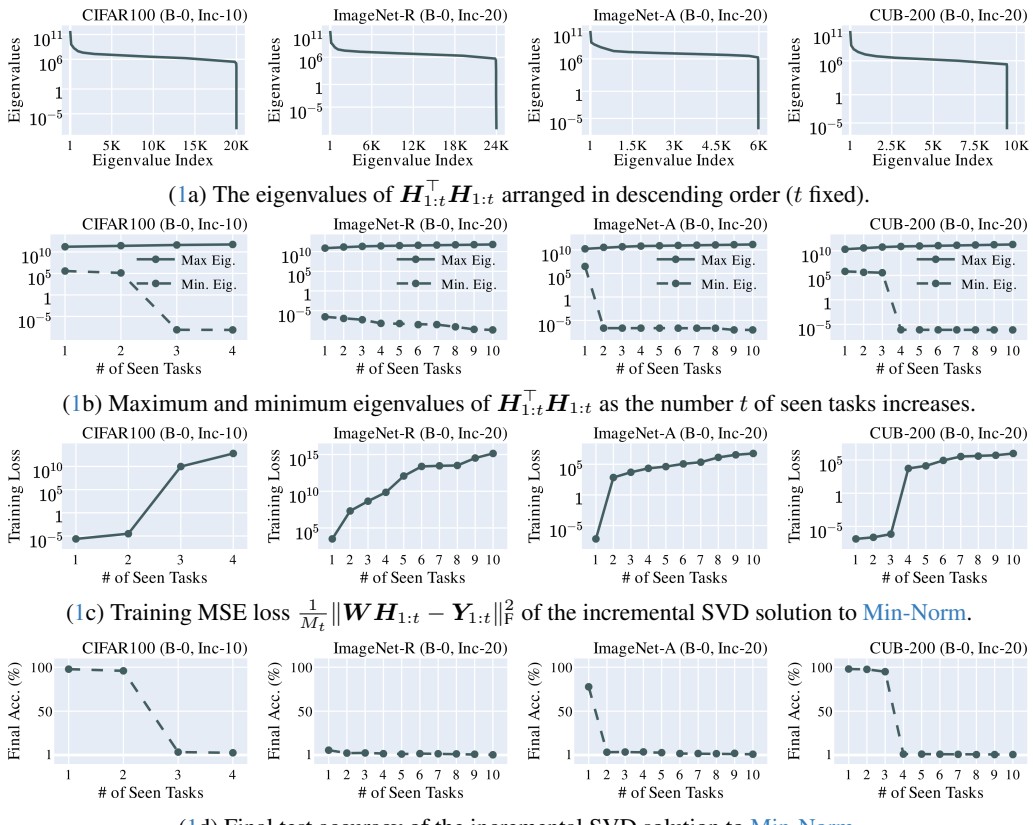

(1a) The eigenvalues of $\boldsymbol{H}_{1:t}^\top \boldsymbol{H}_{1:t}$ arranged in descending order ($t$ fixed).

(1b) Maximum and minimum eigenvalues of $\boldsymbol{H}_{1:t}^\top \boldsymbol{H}_{1:t}$ as the number $t$ of seen tasks increases.

(1c) Training MSE loss $\frac{1}{M_t}\|\boldsymbol{W}\boldsymbol{H}_{1:t} - \boldsymbol{Y}_{1:t}\|_F^2$ of the incremental SVD solution to Min-Norm.

(1d) Final test accuracy of the incremental SVD solution to Min-Norm.

Figure 1: Spectrum of $\boldsymbol{H}_{1:t}^\top \boldsymbol{H}_{1:t}$ and its impact on training losses & test accuracy ($E = 10^5$); see also Appendix K.8. The matrix $\boldsymbol{H}_{1:t}^\top \boldsymbol{H}_{1:t}$ is ill-conditioned (1a); training loss increases (1c) and test accuracy drops (1d), drastically, when small eigenvalues (of order $10^{-5}$) invade the spectrum (1b).

---

**Algorithm 1:** Continual Solver of LoRanPAC (detailed version in Algorithm 4, Appendix C)

1  *Input (Task t)*: Features $\boldsymbol{H}_t \in \mathbb{R}^{E \times m_t}$ (1), labels $\boldsymbol{Y}_t \in \mathbb{R}^{c_t \times m_t}$, truncation percentage $\zeta \in [0,1]$;
2  For $t \leftarrow 1, 2, \ldots$:
3  $\quad k_t \leftarrow \lceil(1-\zeta)\min\{E, M_t\}\rceil$;                    // $M_t := m_1 + \cdots + m_t$ can be updated online
4  $\quad \boldsymbol{J}_t \leftarrow \boldsymbol{Y}_{1:t}\boldsymbol{H}_{1:t}^\top$;                    // online update of $\boldsymbol{J}_t$ detailed in Algorithm 5, Appendix C
5  $\quad$ Form $\boldsymbol{B}_t$ as per (3);
6  $\quad (\widetilde{\boldsymbol{U}}_{1:t}, \widetilde{\boldsymbol{\Sigma}}_{1:t}) \leftarrow$ Top-$k_t$ SVD factors of $\boldsymbol{B}_t$;          // Algorithm 3 if $t=1$, or Algorithm 2 if $t>1$
7  $\quad$ Compute linear classifier $\widetilde{\boldsymbol{W}}_t := \boldsymbol{J}_t\widetilde{\boldsymbol{U}}_{1:t}\widetilde{\boldsymbol{\Sigma}}_{1:t}^{-2}\widetilde{\boldsymbol{U}}_{1:t}^\top$;                    // cf. (2) and (4)

---

$\boldsymbol{U}_{1:t}$ has $\mathrm{rank}(\boldsymbol{H}_{1:t})$ columns and contains $\overline{\boldsymbol{U}}_{1:t}$ as a submatrix. We can then write $\overline{\boldsymbol{W}}_t$ as

$$\overline{\boldsymbol{W}}_t = \boldsymbol{Y}_{1:t}\overline{\boldsymbol{V}}_{1:t}\overline{\boldsymbol{\Sigma}}_{1:t}^{-1}\overline{\boldsymbol{U}}_{1:t}^\top = \boldsymbol{Y}_{1:t}\overline{\boldsymbol{V}}_{1:t}\overline{\boldsymbol{\Sigma}}_{1:t}\overline{\boldsymbol{U}}_{1:t}^\top\left(\overline{\boldsymbol{U}}_{1:t}\overline{\boldsymbol{\Sigma}}_{1:t}^{-2}\overline{\boldsymbol{U}}_{1:t}^\top\right)$$
$$\stackrel{(i)}{=} \boldsymbol{Y}_{1:t}\boldsymbol{V}_{1:t}\boldsymbol{\Sigma}_{1:t}\boldsymbol{U}_{1:t}^\top\left(\overline{\boldsymbol{U}}_{1:t}\overline{\boldsymbol{\Sigma}}_{1:t}^{-2}\overline{\boldsymbol{U}}_{1:t}^\top\right) = \boldsymbol{Y}_{1:t}\boldsymbol{H}_{1:t}^\top\left(\overline{\boldsymbol{U}}_{1:t}\overline{\boldsymbol{\Sigma}}_{1:t}^{-2}\overline{\boldsymbol{U}}_{1:t}^\top\right), \tag{2}$$

where (i) holds as the column vectors of $\boldsymbol{U}_{1:t}$ not shown in $\overline{\boldsymbol{U}}_{1:t}$ are orthogonal to $\overline{\boldsymbol{U}}_{1:t}$. Given (2), it now suffices to update $\boldsymbol{J}_t := \boldsymbol{Y}_{1:t}\boldsymbol{H}_{1:t}^\top \in \mathbb{R}^{c_t \times E}$, $\overline{\boldsymbol{U}}_{1:t} \in \mathbb{R}^{E \times k_t}$, and $\overline{\boldsymbol{\Sigma}}_{1:t} \in \mathbb{R}^{k_t \times k_t}$ in an online fashion. This procedure is described in Algorithm 1, where the following points are considered:

• Since the columns of $\boldsymbol{Y}_{1:t}$ are one-hot vectors, we can compute $\boldsymbol{Y}_{1:t}\boldsymbol{H}_{1:t}^\top$ incrementally by matrix addition rather than (sparse) matrix multiplication.

• An exact update of $\overline{\boldsymbol{U}}_{1:t}$ and $\overline{\boldsymbol{\Sigma}}_{1:t}$ would require computing the SVD factors of the full data $\boldsymbol{H}_{1:t}$. However, past data $\boldsymbol{H}_{1:t-1}$ is not available when observing task $t$. Thus, we consider approximating

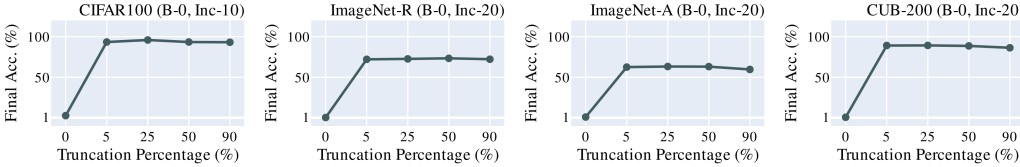

Figure 2: Final test accuracy as the truncation percentage $\zeta$ varies.

$\overline{U}_{1:t}, \overline{\Sigma}_{1:t}$ by two respective matrices $\widetilde{U}_{1:t}, \widetilde{\Sigma}_{1:t}$ of the same sizes. Specifically, as shown in Line 6, we set $\widetilde{U}_{1:t}, \widetilde{\Sigma}_{1:t}$ to be the top $k_t$ SVD factors of $B_t$, where $B_t$ is defined as

$$B_t := \begin{cases} H_1 & \text{if } t = 1; \\ [\widetilde{U}_{1:t-1}\widetilde{\Sigma}_{1:t-1}, \ H_t] & \text{otherwise.} \end{cases} \tag{3}$$

Note that $B_1$ is of size $E \times m_1$, while for $t > 1$ we have $B_t$ of size $E \times (k_{t-1} + m_t)$. The top-$k_t$ singular values of $B_t$ and $H_{1:t}$ are close to each other (cf. Fig. 1a, Fig. 8, Fig. 9, and Theorem 6), which indicates the effectiveness of our continual updating strategy. Then, as shown in Line 7 and recalling (2) and $J_t = Y_{1:t}H_{1:t}^\top$, we construct a linear classifier via

$$\widetilde{W}_t \leftarrow J_t\widetilde{U}_{1:t}\widetilde{\Sigma}_{1:t}^{-2}\widetilde{U}_{1:t}^\top. \tag{4}$$

• In Algorithm 1, $\zeta$ denotes the truncation percentage. Given $\zeta$, we set the number $k_t$ of top SVD factors preserved at task $t$ to $k_t = \lceil (1 - \zeta) \min\{E, M_t\}\rceil$, where $\lceil \cdot \rceil$ converts an input number to the closest integer no smaller than it. Fig. 2 visualizes the effect of $\zeta$ on the final test accuracy, highlighting nearly zero accuracy with $\zeta = 0$ and stable performance with $\zeta \geq 5\%$.

*Remark* 1. At test time, given a sample, we make a forward pass and obtain its random ReLU feature $h$. The predicted class is then set according to the maximum entry of $\widetilde{W}_t h$.

*Remark* 2. Algorithm 1 with $\zeta = 0$ is the incremental SVD method that we use to solve Min-Norm.

*Remark* 3 (Time and Space Complexity). The major cost of Algorithm 1 is to compute the top-$k_t$ SVD factors of the $E \times (k_{t-1} + m_t)$ matrix $B_t$, which takes $O(E(k_{t-1} + m_t)^2)$ time. Futhermore Algorithm 1 uses $O(2Ec_t + Ek_t + k_t^2)$ memory to store $J_t, \widetilde{U}_{1:t}$, and $\widetilde{\Sigma}_{1:t}$, and $\widetilde{W}_t$, and $O(Ek_{t-1} + Em_t)$ memory to construct $B_t$, and some extra working memory to compute the top-$k_t$ SVD factors of $B_t$. We refer the reader to Appendix C for more details.

## 4 PROVABLY CONTROLLED TRAINING AND TEST ERRORS

In this section, we present Theorems 1 and 2, which bound the training and test error of the output (4) of our approach (Algorithm 1).

**Notations.** Denote by $\mu_k(\cdot)$ the $k$-th largest eigenvalue of a symmetric matrix. Define

$$\gamma_1 := 1, \quad \gamma_t := \frac{\mu_{k_t}(B_tB_t^\top)}{\max_{i=1,\dots,t-1}\{\mu_{k_i+1}(B_iB_i^\top)\}}, \quad \forall t > 1. \tag{5}$$

The quantity $\gamma_t$ relates to the *stability-plasticity tradeoff*, as it is the ratio between the *minimum preserved eigenvalue* $\mu_{k_t}(B_tB_t^\top)$ at task $t$ and the *maximum eigenvalues being truncated in the past*, $\mu_{k_i+1}(B_iB_i^\top)$. Clearly $\gamma_t > 0$, as we truncate only non-zero eigenvalues. Furthermore, instead of determining the number of preserved SVD factors $k_t$ based on the truncation percentage $\zeta$, we can take a threshold hyperparameter $\delta$ and truncate eigenvalues smaller than $\delta$; this $\delta$ implicitly determines $k_t$'s, and we have $\mu_{k_t}(B_tB_t^\top) \geq \delta > \mu_{k_i+1}(B_iB_i^\top)$, which implies $\gamma_t \geq 1$. Finally, as suggested by Fig. 1a, $\gamma_t$ can be as large as $10^{10}$: If we set $\delta = 10^{-2}$ in the case of Fig. 1a, then the maximum truncated eigenvalue is of order $10^{-5}$ and the minimum preserved is of order $10^5$.

Then, the *accumulative error* $a_t$ is defined as

$$a_0 := 0, \qquad a_t := \sum_{i=1}^{t} \mu_{k_i+1}(B_iB_i^\top), \quad \forall t \geq 1. \tag{6}$$

The term $a_t$ reflects the information ignored by our algorithm, as $\mu_{k_i+1}(\boldsymbol{B}_i\boldsymbol{B}_i^\top)$ is the maximum eigenvalue truncated at task $i$. Note that, even when observing thousands of tasks (e.g., $t \approx 10^3$), if we truncate the smallest eigenvalues (of order $10^{-5}$), $a_t$ is in the order of $10^{-2}$.

**Model Assumption.** We consider a noisy linear regression model. Specifically, we assume there is some *ground-truth* weight matrix $\boldsymbol{W}_t^* \in \mathbb{R}^{c_t \times E}$ and noise $\mathcal{E}_{1:t} \in \mathbb{R}^{c_t \times M_t}$ satisfying

$$\boldsymbol{Y}_{1:t} = \boldsymbol{W}_t^*\boldsymbol{H}_{1:t} + \mathcal{E}_{1:t}. \tag{7}$$

The quantities $\boldsymbol{W}_t^*$ and $\mathcal{E}_{1:t}$ are colored to reflect the fact that they are unknown and not computable. The model in (7) is related to *probabilistic principal component analysis* (PPCA); cf. Tipping & Bishop (1999) and Chapter 2.2 of Vidal et al. (2016). The two main differences with PPCA are that we make no probabilistic assumptions on $\boldsymbol{H}_{1:t}$ or $\mathcal{E}_{1:t}$ (except in Appendix F); and we consider the over-parameterized case with large $E$, while PPCA assumes $\boldsymbol{W}_t^*$ is a tall matrix (i.e., $E < c_t$).

**Bound The Training MSE.** In the over-parametrized regime $E \gg M_t$, a solution to Min-Norm should, in principle, perfectly fit the data and achieve zero training MSE. However, solving Min-Norm is numerically unstable and empirically entails huge losses (Fig. 4). As a remedy, our approach truncates the data spectrum continually, trading off between perfectly fitting training data and increasing numerical stability. The following theorem, whose proof can be found in Appendix E, connects the eigenvalue ratio $\gamma_t$ and the accumulative error $a_t$ with our method's training loss, showing that the training MSE is provably under control:

**Theorem 1.** *Let $\boldsymbol{B}_t, \gamma_t, a_t$ be defined as in (3), (5), and (6) respectively. If $\boldsymbol{Y}_{1:t} = \boldsymbol{W}_t^*\boldsymbol{H}_{1:t} + \mathcal{E}_{1:t}$ (7), then the output $\widetilde{\boldsymbol{W}}_t = \boldsymbol{Y}_{1:t}\boldsymbol{H}_{1:t}^\top\widetilde{\boldsymbol{U}}_{1:t}\widetilde{\boldsymbol{\Sigma}}_{1:t}^{-2}\widetilde{\boldsymbol{U}}_{1:t}^\top$ of our method (4) satisfies*

$$\frac{1}{M_t}\left\|\widetilde{\boldsymbol{W}}_t\boldsymbol{H}_{1:t} - \boldsymbol{Y}_{1:t}\right\|_{\mathrm{F}}^2 \le 4 \cdot \|\boldsymbol{W}_t^*\|_{\mathrm{F}}^2\left(\frac{a_t}{M_t} + \frac{a_{t-1}(t-1)}{\gamma_t M_t} + \frac{a_{t-1}(t-1)^2}{\gamma_t^2 M_t}\right) \tag{8}$$

$$+ 2 \cdot \|\mathcal{E}_{1:t}\|^2\left(\frac{(M_t - k_t)}{M_t} + \frac{(t-1)\min\{M_{t-1} - k_{t-1}, (t-1)k_t\}}{\gamma_t^2 M_t}\right).$$

In (8), $\|\cdot\|$ is an overloaded notation, denoting the spectrum norm of a matrix and also the Euclidean norm of a vector. One of the main quantities governing the bound in Theorem 1 is $a_t/M_t$, which reflects the truncation process for the current task. When truncating the smallest eigenvalues (of order $10^{-5}$) and observing hundreds of tasks, $a_t$ is in the order of $10^{-3}$, which makes $a_t/M_t$ insignificant. Then, the terms $(t-1)/\gamma_t$ and $a_{t-1}/M_t$ capture the continual past truncations and are equal to zero for $t = 1$. Similarly to $a_t$, when truncating only the smallest eigenvalues, we have $a_{t-1} \approx 10^{-5}(t-1)$ and $(t-1)/\gamma_t \approx 10^{-10}(t-1)$. Hence, all terms involving $(t-1)/\gamma_t$ and $a_{t-1}/M_t$ are under control for hundreds- even thousands- of tasks. Finally, although the ground-truth $\boldsymbol{W}_t^*$ and noise $\mathcal{E}_{1:t}$ are unknown, we empirically verify that the minimum-norm solution to LoRanPAC achieves high accuracy (Section 6). This suggests the linear model assumption is adequate, and that $\|\mathcal{E}_{1:t}\|^2$ and $\|\boldsymbol{W}_t^*\|_{\mathrm{F}}^2$ are reasonably small. In summary, the upper bound (8) shown in Theorem 1 behaves well and is quite small if we truncate the eigenvalues suitably (which makes $\gamma_t$ large and $a_t$ small).

**Bound The Test MSE.** Consider a test sample $(\boldsymbol{h}, \boldsymbol{y})$ satisfying $\boldsymbol{y} = \boldsymbol{W}_t^*\boldsymbol{h} + \boldsymbol{\epsilon}$ for some noise vector $\boldsymbol{\epsilon}$. To derive a bound on the test MSE, we assume that $\boldsymbol{h}$ is randomly sampled from some distribution with a finite second-order moment ($\boldsymbol{\Lambda} := \mathbb{E}[\boldsymbol{h}\boldsymbol{h}^\top] < \infty$), and that $\boldsymbol{\epsilon}$ is random, independent of $\boldsymbol{h}$. Given the output $\widetilde{\boldsymbol{W}}_t$ of our method (4), we bound its test error $\mathbb{E}_{\boldsymbol{h},\boldsymbol{\epsilon}}\left[\|\widetilde{\boldsymbol{W}}_t\boldsymbol{h} - \boldsymbol{y}\|^2\right]$ over the randomness of $\boldsymbol{h}, \boldsymbol{\epsilon}$ as follows:

**Theorem 2.** *Let $\boldsymbol{B}_t, \gamma_t, a_t$ be defined as in (3), (5), and (6) respectively. Assume $\boldsymbol{Y}_{1:t} = \boldsymbol{W}_t^*\boldsymbol{H}_{1:t} + \mathcal{E}_{1:t}$ (7) and $\boldsymbol{y} = \boldsymbol{W}_t^*\boldsymbol{h} + \boldsymbol{\epsilon}$ with $\boldsymbol{\Lambda} := \mathbb{E}[\boldsymbol{h}\boldsymbol{h}^\top]$. The output $\widetilde{\boldsymbol{W}}_t$ of Algorithm 1 satisfies*

$$\mathbb{E}_{\boldsymbol{h},\boldsymbol{\epsilon}}\left\|\widetilde{\boldsymbol{W}}_t\boldsymbol{h} - \boldsymbol{y}\right\|^2 \le 4 \cdot \|\boldsymbol{W}_t^*\|_{\mathrm{F}}^2 \cdot \mathbb{B}_t + 4 \cdot \|\mathcal{E}_{1:t}\|^2 \cdot \mathbb{V}_t + 2 \cdot \mathbb{E}_{\boldsymbol{\epsilon}}\left[\|\boldsymbol{\epsilon}\|^2\right], \tag{9}$$

*where $\mathbb{B}_t$ and $\mathbb{V}_t$ are defined as follows:*

$$\mathbb{B}_t = \left\|\boldsymbol{\Lambda} - \frac{1}{M_t}\boldsymbol{H}_{1:t}\boldsymbol{H}_{1:t}^\top\right\|\left(1 + \frac{(t-1)^2}{\gamma_t^2}\right) + \left(\frac{a_t}{M_t} + \frac{a_{t-1}(t-1)}{\gamma_t M_t} + \frac{a_{t-1}(t-1)^2}{\gamma_t^2 M_t}\right)$$

$$\mathbb{V}_t = \left\|\boldsymbol{\Lambda} - \frac{1}{M_t}\boldsymbol{H}_{1:t}\boldsymbol{H}_{1:t}^\top\right\| \cdot \frac{\left(\frac{1}{\gamma_t}\min\{M_{t-1} - k_{t-1}, (t-1)k_t\} + k_t\right)}{\mu_{k_t}(\boldsymbol{B}_t\boldsymbol{B}_t^\top)} \tag{10}$$

$$+ \frac{k_t}{M_t} + \left(\frac{t-1}{\gamma_t^2 M_t} + \frac{2}{\gamma_t M_t}\right) \cdot \min\{M_{t-1} - k_{t-1}, (t-1)k_t\}.$$

There are two major terms in $\mathbb{B}_t$ (10). The term in the right-most large parenthesis also appears in the error bound of Theorem 1; and reflects the fact that training losses impact test errors. Then, the term $\left\| \mathbf{\Lambda} - \frac{1}{M_t} \boldsymbol{H}_{1:t} \boldsymbol{H}_{1:t}^\top \right\|$ is commonly seen in *covariance estimation* (Wainwright, 2019), where $\boldsymbol{h}$ and the columns of $\boldsymbol{H}_{1:t}$ are assumed to be independent i.i.d. Gaussian vectors. In this case, if $\mathbf{\Lambda}$ furthermore satisfies some boundedness condition, we can show $\left\| \mathbf{\Lambda} - \frac{1}{M_t} \boldsymbol{H}_{1:t} \boldsymbol{H}_{1:t}^\top \right\|$ converges to 0 as $M_t \to \infty$; cf. Theorem 9 of Koltchinskii & Lounici (2017). On the other hand, the Gaussian assumption is sufficient but not necessary, and a similar conclusion is reached if we take a much weaker assumption called *hypercontractivity* (Jirak et al., 2024). Note that $\mathbb{V}_t$ is independent of noise, so the rest of the terms in (9), which are weighted by noise magnitudes $\|\mathcal{E}_{1:t}\|^2$, $\mathbb{E}_\epsilon\left[\|\boldsymbol{\epsilon}\|^2\right]$, are negligible if the noise is sufficiently small.

## 5 RELATED WORK

We now discuss related works that are the most relevant to our method and theory. A more extensive review of the literature and context is in Appendix H.

**RanPAC.** The RanPAC method of McDonnell et al. (2023) motivates our use of random ReLU features $\boldsymbol{H}_1, \ldots, \boldsymbol{H}_t$. It amounts to solving the ridge regression problem (with some $\lambda > 0$)

$$\min_{\boldsymbol{W} \in \mathbb{R}^{c_t \times E}} \lambda \cdot \|\boldsymbol{W}\|_{\mathrm{F}}^2 + \|\boldsymbol{W}\boldsymbol{H}_{1:t} - \boldsymbol{Y}_{1:t}\|_{\mathrm{F}}^2. \tag{RanPAC}$$

The choice of hyperparameter $\lambda$ is crucial; RanPAC with a small regularization $\lambda$ fails to achieve competitive performance, while it might work well with a large enough $\lambda$ (e.g., $\lambda = 10^4$); cf. Fig. 12. In constrast, Prabhu et al. (2024) finds that small $\lambda$ (of order $10^{-5}$) works better if $\boldsymbol{H}_{1:t}$ is replaced with *random Fourier features*. This implies the optimal choice of $\lambda$ depends, among other factors, on the scale of the features and the noise level. Our method also has a hyperparameter, the truncation percentage $\zeta$, while the choice of $\zeta$ is less sensitive to these factors (Fig. 2). In the implementation of McDonnell et al. (2023), RanPAC selects $\lambda$ from the predefined set $\{10^{-8}, 10^{-7}, \ldots, 10^8\}$ via cross-validation on a small faction of training data. Although this stabilizes RanPAC in some cases, cross-validation can fail when the validation (or training) set is small and not representative of test data. Unfortunately, this failure occurs often in CIL with small increments (cf. Table 2, Section 6).

In more detail, for every task $t$ and every each candidate choice of $\lambda$, RanPAC maintains the covariances $\boldsymbol{H}_{1:t}\boldsymbol{H}_{1:t}^\top$, $\boldsymbol{Y}_{1:t}\boldsymbol{H}_{1:t}^\top$, to solve the normal equations $\boldsymbol{W}(\boldsymbol{H}_{1:t}\boldsymbol{H}_{1:t}^\top + \lambda \boldsymbol{I}_E) = \boldsymbol{Y}_{1:t}\boldsymbol{H}_{1:t}^\top$ in variable $\boldsymbol{W}$ using off-the-shelf solvers implemented in PyTorch, which in general takes $O(E^3)$ time. In contrast, LoRanPAC has $O(E(k_{t-1} + m_t)^2)$ time complexity, and this is why it is slower than LoRanPAC for the same $E$, particularly when $E$ is large (cf. Fig. 3, Section 6). Certainly, both RanPAC and LoRanPAC can potentially be implemented more efficiently. For example, RanPAC involves inverting the regularized covariance $\boldsymbol{H}_{1:t}\boldsymbol{H}_{1:t}^\top + \lambda \boldsymbol{I}_E$, and this inverse can be updated continually via the *Sherman–Morrison–Woodbury* formula. This formula is at the heart of the classic *recursive least-squares* method (Sayed, 2008), and its philosophy is also found in recent continual learning papers (Min et al., 2022; Zhuang et al., 2022; 2023). However, it is known that such a scheme can be numerically unstable, brittle for ill-conditioned data. Indeed, in our setting, We find the implementation based on the Sherman–Morrison–Woodbury formula suffers from numerical errors and is unable to maintain good accuracy. Moreover, numerical errors accumulate over time, leading to worse performance for longer sequences of tasks. Finally, even if we know the numerical errors might arise in these methods, there is no obvious way to remedy them. This is different from our implementation based on robust truncated SVD, which has the advantage that we could (empirically) reduce numerical errors by re-orthogonalizing $\widetilde{\boldsymbol{U}}_{1:t}$ (see Remark 4 and Algorithm 2).

One more component in RanPAC is a preprocessing step called *first-session adaptation*. That is, before using the pre-trained model for continual learning, one fine-tunes it with data from the first task in a *parameter-efficient* way (Panos et al., 2023). This needs extra hyperparameters and yields different features than $\boldsymbol{H}_{1:t}$. We study the impact of this step in Table 1, Section 6.

The final point that relates LoRanPAC to RanPAC is this: $\overline{\boldsymbol{W}}_t$ in (2) is a global minimizer of

$$\min_{\boldsymbol{W} \in \mathbb{R}^{c_t \times E}} \|\boldsymbol{W}\tau_{k_t}(\boldsymbol{H}_{1:t}) - \boldsymbol{Y}_{1:t}\|_{\mathrm{F}}^2. \tag{11}$$

Both LoRanPAC and RanPAC aim to minimize the MSE loss; the former uses truncation and the latter uses regularization to make the problem better conditioned. The MSE loss typically yields

similar performance to the cross-entropy loss in many settings (Janocha & Czarnecki, 2017; Hui & Belkin, 2021), and the MSE loss is preferred here as it allows for a closed-form least-squares solution to be rapidly computed and continually updated.

**ICL.** LoRanPAC is also related to the *Ideal Continual Learner* (ICL) of Peng et al. (2023), which in the linear, over-parameterized case is the following linearly constrained quadratic problem:

$$\min_{\boldsymbol{W} \in \mathbb{R}^{c_t \times E}} \|\boldsymbol{W}\boldsymbol{H}_t - \boldsymbol{Y}_t\|_{\mathrm{F}}^2 \qquad \text{s.t.} \qquad \boldsymbol{W}\boldsymbol{H}_i = \boldsymbol{Y}_i, \quad i = 1, \dots, t-1. \qquad \text{(ICL)}$$

Proposition 5 of Peng et al. (2023) gives a method based on SVD to solve ICL; it is proved in Peng & Vidal (2025) that this method implicitly finds the solution to Min-Norm. But we have seen in Fig. 1 that solving Min-Norm is numerically challenging due to highly ill-conditioned features $\boldsymbol{H}_{1:t}$. Proposition 6 of Peng et al. (2023) further suggests that solving ICL by a gradient-based method gives the approach of Farajtabar et al. (2020), known as *Orthogonal Gradient Descent* (OGD). Subsequently, OGD is combined with the idea of SVD truncation in the *PCA-OGD* method (Doan et al., 2021). As gradient-based methods, OGD and PCA-OGD converge slowly for ill-conditioned data and would be less efficient than our LoRanPAC implementation; the differences between PCA-OGD and our method are thoroughly discussed in our rebuttal. The OR-Fit method of Min et al. (2022) improves PCA-OGD by devising carefully chosen stepsizes that facilitate solving the current task. Their proposed stepsizes are related to ICL and recursive least-squares in an intriguing manner; we refer the reader to Peng & Vidal (2025) for the precise mathematical connections and a unifying perspective on the aforementioned methods.

**Principal Component Regression.** LoRanPAC combines *principal component analysis* and *ordinary least-squares*, which is analogous to *principal component regression* (PCR) (Xu & Hsu, 2019; Huang et al., 2022; Hucker & Wahl, 2023; Bach, 2024; Green & Romanov, 2024). These papers consider the offline setting, where truncation is performed only once. In contrast, we analyze the effect of continual truncation, which is most pertinent for CL. Indeed, for $t = 1$, $\mathbb{B}_1$ of Theorem 2 is equal to the corresponding term in Theorem 1 of Huang et al. (2022) up to a constant. More importantly, these papers have statistical assumptions on $\boldsymbol{H}_{1:t}$, which are potentially violated by generating $\boldsymbol{H}_{1:t}$ via $\boldsymbol{H}_t := \mathrm{relu}(\boldsymbol{P}\boldsymbol{X}_t)$, with $\boldsymbol{X}_t$ consisting of features from pre-trained models. In contrast, Theorems 1 and 2 have few assumptions, and so they apply, at least in principle, to the *full* architecture (i.e., a pre-trained model and random ReLU feature model in cascade).

## 6 NUMERICAL VALIDATION

This section highlights the performance and efficiency of LoRanPAC in the CIL setting across a diverse range of datasets and increments. For additional results, see Appendix K, particularly Appendix K.4 for experimental outcomes in the DIL (*domain-incremental learning*) setting.

### 6.1 SETUP

**Baselines.** The most relevant baseline to compare is RanPAC (McDonnell et al., 2023). Additional competitive baselines include L2P (Wang et al., 2022d), DualPrompt (Wang et al., 2022c), CodaPrompt (Smith et al., 2023), SimpleCIL, ADAM (Zhou et al., 2023) and EASE (Zhou et al., 2024a). We also compare LoRanPAC with a *joint linear classifier*, that is, a linear model trained using either the pre-trained features $\boldsymbol{X}_{1:T}$ of all $T$ tasks, or the random ReLU features $\boldsymbol{H}_{1:T}$. We denote these two methods by LC ($\boldsymbol{X}_{1:T}$) and LC ($\boldsymbol{H}_{1:T}$). To ensure a fair comparison, all experiments are conducted based on the PILOT GitHub repository of Sun et al. (2023). Additional experimental details, as well as a comprehensive review of relevant baselines is given in Appendix J and Appendix H.

**Pre-trained Models.** We use ViT models pre-trained on ImageNet-1K; specifically the model `vit_base_patch16_224` from the `timm` repository (Wightman, 2019). Experiments using ViTs pre-trained on ImageNet-21K are presented in Appendix K.2.

**Datasets.** Following prior works (Zhou et al., 2023; McDonnell et al., 2023), we run CIL experiments with B-$q_1$, Inc-$q_2$ on continual learning versions of the following datasets: CIFAR100 (Krizhevsky et al., 2009), ImageNet-R (Hendrycks et al., 2021a), ImageNet-A (Hendrycks et al., 2021b), CUB-200 (Wah et al., 2011), ObjectNet (Barbu et al., 2019), OmniBenchmark (Zhang et al., 2022), VTAB (Zhai et al., 2019), and StanfordCars (Krause et al., 2013). We set $q_1 = 0$ for most cases, but since

Table 1: Final accuracy with pre-trained ViTs. Large accuracy gaps between RanPAC and LoRanPAC (ours) are shown in bold. †: Methods using first-session adaptation with the hyperparameters set as per RanPAC†. ∗: Methods using first-session adaptation with the hyperparameters set as per EASE∗ (Zhou et al., 2024a). Table 14 reports standard deviation. Appendix J reports experimental details.

| (Part 1) | CIFAR100 (B-0) | | | ImageNet-R (B-0) | | | ImageNet-A (B-0) | | | CUB-200 (B-0) | | | Avg. |
|---|---|---|---|---|---|---|---|---|---|---|---|---|---|
| LC ($\boldsymbol{X}_{1:T}$) | | 87.56 | | | 72.42 | | | 58.85 | | | 88.76 | | 76.90 |
| LC ($\boldsymbol{H}_{1:T}$) | | 87.76 | | | 73.00 | | | 59.25 | | | 88.72 | | 77.18 |
| | Inc-5 | Inc-10 | Inc-20 | Inc-5 | Inc-10 | Inc-20 | Inc-5 | Inc-10 | Inc-20 | Inc-5 | Inc-10 | Inc-20 | |
| L2P | 80.25 | 83.53 | 83.57 | 67.92 | 71.78 | 73.42 | 44.50 | 48.52 | 51.28 | 53.60 | 59.20 | 67.81 | 65.45 |
| DualPrompt | 80.85 | 83.86 | 84.59 | 67.12 | 71.57 | 72.87 | 49.70 | 53.72 | 56.75 | 54.79 | 63.99 | 69.93 | 67.48 |
| CodaPrompt | 82.93 | 86.31 | 87.87 | 67.80 | 72.73 | 74.85 | 34.43 | 49.57 | 59.51 | 36.39 | 60.18 | 71.29 | 65.32 |
| SimpleCIL | 80.48 | 80.48 | 80.48 | 63.47 | 63.47 | 63.47 | 58.72 | 58.72 | 58.72 | 80.45 | 80.45 | 80.45 | 70.78 |
| RanPAC | 86.71 | 87.02 | 87.10 | 71.90 | 71.97 | 72.50 | **56.48** | 62.34 | 61.75 | 88.08 | 87.15 | 88.13 | 76.76 |
| LoRanPAC | 88.18 | 88.18 | 88.21 | 73.67 | 73.72 | 73.63 | **62.74** | 63.20 | 63.20 | 89.36 | 89.27 | 89.23 | 78.55 |
| ADAM† | 83.55 | 85.13 | 85.86 | 63.73 | 65.03 | 71.40 | 58.72 | 58.66 | 58.99 | 80.49 | 80.66 | 81.00 | 72.77 |
| RanPAC† | 88.73 | 90.04 | 90.74 | 70.80 | 73.37 | 78.80 | 62.34 | 62.08 | 62.28 | 88.42 | 87.57 | 88.68 | 78.65 |
| LoRanPAC† | 89.73 | 90.82 | 91.44 | 73.58 | 74.55 | 79.13 | 62.74 | 62.80 | 62.94 | 89.14 | 89.19 | 89.27 | 79.61 |
| EASE∗ | 84.43 | 86.48 | 88.16 | 73.53 | 77.02 | 77.55 | 58.26 | 61.69 | 62.28 | 80.66 | 81.68 | 81.13 | 76.07 |
| LoRanPAC∗ | 89.46 | 90.90 | 91.67 | 78.73 | 80.43 | 81.45 | 63.40 | 64.45 | 65.64 | 89.14 | 89.19 | 89.44 | 81.16 |
| (Part 2) | ObjectNet (B-0) | | | OmniBenchmark (B-0) | | | VTAB (B-10) | | | StanfordCars (B-16) | | | Avg. |
| LC ($\boldsymbol{X}_{1:T}$) | | 59.70 | | | 79.55 | | | 91.32 | | | 74.12 | | 76.17 |
| LC ($\boldsymbol{H}_{1:T}$) | | 59.96 | | | 80.02 | | | 91.17 | | | 73.65 | | 76.20 |
| | Inc-5 | Inc-10 | Inc-20 | Inc-5 | Inc-10 | Inc-20 | Inc-5 | Inc-10 | Inc-20 | Inc-5 | Inc-10 | Inc-20 | |
| L2P | 45.53 | 52.05 | 55.49 | 54.50 | 57.29 | 60.50 | 59.32 | 73.25 | 78.91 | 13.70 | 27.46 | 43.68 | 51.81 |
| DualPrompt | 47.56 | 53.68 | 55.64 | 56.14 | 59.18 | 62.39 | 64.10 | 77.78 | 83.75 | 11.38 | 18.84 | 27.89 | 51.53 |
| CodaPrompt | 46.61 | 54.44 | 59.17 | 60.00 | 64.98 | 68.25 | 68.77 | 76.81 | 86.32 | 7.96 | 11.29 | 30.74 | 52.95 |
| SimpleCIL | 51.66 | 51.66 | 51.66 | 70.19 | 70.19 | 70.19 | 82.53 | 82.53 | 82.53 | 35.46 | 35.46 | 35.46 | 59.96 |
| RanPAC | 58.77 | 57.66 | 57.69 | 77.63 | 77.63 | 77.46 | 91.15 | 91.58 | 91.58 | **58.03** | 71.40 | 71.40 | 73.50 |
| LoRanPAC | 60.83 | 60.86 | 60.77 | 79.50 | 79.60 | 79.70 | 92.46 | 92.55 | 92.56 | **74.21** | 74.39 | 74.39 | 76.82 |
| ADAM† | 52.16 | 53.94 | 55.97 | 70.54 | 70.53 | 70.38 | 82.55 | 82.55 | 82.55 | 35.61 | 35.61 | 35.61 | 60.67 |
| RanPAC† | 59.14 | 61.54 | 64.59 | 78.10 | 78.46 | 78.86 | 91.48 | 91.86 | 91.86 | **58.65** | 72.24 | 72.24 | 74.56 |
| LoRanPAC† | 61.78 | 63.56 | 66.48 | 80.07 | 80.28 | 80.45 | 92.55 | 92.53 | 92.60 | **74.87** | 74.89 | 75.13 | 77.93 |
| EASE∗ | 49.28 | 53.88 | 57.05 | 70.33 | 70.68 | 70.84 | 89.85 | 93.48 | 93.49 | 32.43 | 31.77 | 29.00 | 61.84 |
| LoRanPAC∗ | 61.57 | 63.40 | 66.29 | 80.02 | 80.42 | 80.82 | 92.68 | 92.71 | 92.67 | 75.91 | 75.71 | 75.96 | 78.18 |

StanfordCars and VTAB have 196 and 50 classes, respectively, we take $q_1 = 16$ and $q_1 = 10$ for them. We let $q_2$ vary in $\{5, 10, 20\}$, and also consider the more challenging case $q_2 = 1$.

**Metrics.** After learning task $t$ we evaluate the top-1 classification accuracy $\mathcal{A}_{i,t}$ for every $i = 1, \dots, t$. For a total of $T$ tasks, the *accuracy matrix* $\mathcal{A}$ is defined as a $T \times T$ upper triangular matrix with its $(i, t)$-th entry being $\mathcal{A}_{i,t}$. *Final accuracy* is defined as the average $\frac{1}{T} \sum_{i=1}^{T} \mathcal{A}_{i,T}$ of the last column of $\mathcal{A}$. *Total accuracy* is defined as the average $\frac{1}{T(T-1)} \sum_{1 \le i \le t \le T} \mathcal{A}_{i,t}$ of all upper triangular entries. Following common practices, we use total accuracy and final accuracy as our evaluation metrics.

## 6.2 EXPERIMENTAL RESULTS AND ANALYSIS

Table 1 contains the main results for $q_2 = 5, 10, 20$ on 8 different CIL datasets. First observe that L2P, DualPrompt, and CodaPrompt are unstable as their accuracy varies significantly in different datasets for different values of $q_2$. Second, SimpleCIL, ADAM, and EASE are unstable as their performance is largely compromised on StanfordCars. Then, RanPAC is unstable with respect to $q_2$ as it exhibits a large performance gap compared to LoRanPAC for $q_2 = 5$ on ImageNet-A and StanfordCars. Finally, we see LoRanPAC has more stable performance across datasets and for varying $q_2$.

**Why Does LoRanPAC Uniformly Outperform RanPAC?** The first reason is that LoRanPAC's high efficiency and scalability enable the use of a larger embedding dimension. Indeed, LoRanPAC uses $E = 10^5$, taking advantage of the scaling law (Fig. 6, Appendix K.5), while RanPAC uses its default choice $E = 10^4$. Note that this is a fair comparison since LoRanPAC's implementation is more scalable and more efficient than RanPAC's. Specifically, LoRanPAC has $O(E(k_{t-1} + m_t)^2)$

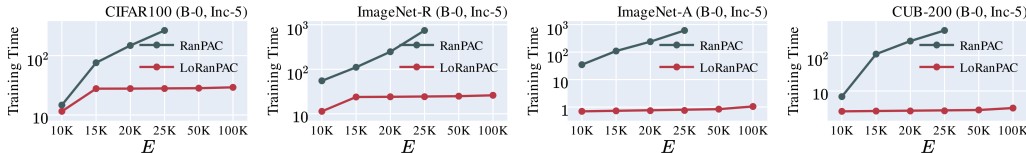

Figure 3: Training times (in minutes) for varying embedding dimensions $E$.

time complexity while RanPAC takes $O(E^3)$ time for each task $t$. An alternative way to make a fair comparison is to set the same embedding dimension $E$ for both methods, in which case LoRanPAC can be up to 1000 times faster than RanPAC (e.g., see $E = 25000$ in Fig. 3).

The second reason, as mentioned earlier, is that the cross-validation strategy of McDonnell et al. (2023) might fail to choose a suitable regularization $\lambda$ for RanPAC when the validation set is small. This is the case in ImageNet-A (B-0, Inc-5) and StanfordCars (B-16, Inc-5) of Table 1, where the validation sets are small and RanPAC's performance is severely degraded. A more careful analysis of these two failure cases shows that the accuracy matrices of RanPAC have multiple columns with nearly zero entries (cf. Fig. 15a and Fig. 17c), exposing RanPAC's instability.

Table 2: Final and total accuracy in CIL datasets with $q_2 = 1$ (Inc-1).

| | CIFAR100 | ImageNet-R | ImageNet-A | CUB | ObjectNet | OmniBenchmark | VTAB | StanfordCars |
|---|---|---|---|---|---|---|---|---|
| | | | | *Final Accuracy* | | | | |
| RanPAC | $86.99_{\pm0.06}$ | $70.12_{\pm0.39}$ | $36.6_{\pm25.35}$ | $55.15_{\pm37.14}$ | $57.14_{\pm0.24}$ | $77.9_{\pm0.04}$ | $91.47_{\pm0.3}$ | $35.56_{\pm24.75}$ |
| LoRanPAC | $88.19_{\pm0.05}$ | $73.66_{\pm0.07}$ | $62.76_{\pm0.16}$ | $89.19_{\pm0.06}$ | $60.82_{\pm0.15}$ | $79.3_{\pm0.06}$ | $92.51_{\pm0.05}$ | $74.32_{\pm0.11}$ |
| | | | | *Total Accuracy* | | | | |
| RanPAC | $90.46_{\pm0.73}$ | $69.1_{\pm0.37}$ | $44.23_{\pm0.46}$ | $74.67_{\pm2.87}$ | $62.37_{\pm2.1}$ | $85.23_{\pm0.56}$ | $74.67_{\pm3.07}$ | $56.27_{\pm0.78}$ |
| LoRanPAC | $92.18_{\pm0.56}$ | $78.87_{\pm0.34}$ | $70.08_{\pm0.86}$ | $92.89_{\pm0.59}$ | $70.54_{\pm1.94}$ | $86.51_{\pm0.59}$ | $96.41_{\pm0.31}$ | $81.18_{\pm0.68}$ |

**Inc-1: One Class at A Time.** In light of the above analysis, we consider the CIL setting, with one class given at each iteration (Inc-1). In this setting, a new task has much fewer training samples and CL methods need to cope with hundreds of tasks (classes) on certain datasets. Note that adapter-based methods such as EASE are infeasible for CIL with Inc-1 (cf. Appendix H.1). In this setting, the fragility of RanPAC with respect to the choice of $\lambda$ is amplified (see Table 2), and the method exhibits a significant performance drop compared to Table 1. In contrast, LoRanPAC's performance is stable, exhibiting high accuracy comparable to the cases of Inc-$\{5, 10, 20\}$ in Table 1. Accuracy matrices associated with Table 2 are plotted in Figs. 14 to 18 of Appendix K.10.3, where we present similar results for Inc-$\{1, 2, 4, 5\}$.

## 7    CONCLUSION

This work puts forward a simple method that bridges the gap between empirical performance and theoretical guarantees in continual learning with pre-trained models. By addressing the ill-conditioning of lifted features through continual SVD truncation, our approach achieves both stability and strong performance. Extensive experiments demonstrated that our method outperforms state-of-the-art methods across multiple datasets and can handle sequences with hundreds of tasks. Theoretically, we proved that our method maintains small training and test errors by appropriately truncating SVD factors. This work underscores the potential of combining empirical techniques with principled frameworks to develop robust and scalable continual learning systems, and will encourage follow-up works to achieve so as well.

ACKNOWLEDGMENTS

This work is supported by the project ULEARN "Unsupervised Lifelong Learning", funded by the Research Council of Norway (grant number 316080), and by NSF-Simons Research Collaborations on the Mathematical and Scientific Foundations of Deep Learning (NSF grant 2031985).

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

# A  OVERVIEW OF THE APPENDIX

- In Appendix B, we complie all the mathematical notations used throughout the paper.
- In Appendix C we describe the implementation details of LoRanPAC. There, we also discuss other potential implementations and our design choice.
- In Appendix D, we present auxiliary lemmas that are useful for proving our main theorems.
- In Appendix E, we prove the theorems displayed in the main paper (Theorems 1 and 2).
- In Appendix F we present results similar to Appendix E, with the difference that we now assume the noise is Gaussian, which gives slightly tighter error bounds.
- In Appendix G, we prove some extra theoretical results such as perturbation bounds on eigenvalues and eigenvectors (Theorem 6).
- In Appendix H we review related works on continual learning, focusing on CL methods with pretrained models and existing theoretical developments.
- In Appendix I we report the statistics of the datasets we use for experiments.
- In Appendix J we specify the experimental setup.
- In Appendix K we report extra experimental results, figures, and tables.

# B  NOTATIONS

Here in Table 3 we compile all the notations used in the paper.

Table 3: Notations

| | |
|---|---|
| $d$ | dimension of pre-trained features |
| $E$ | embedding dimension |
| $m_t$ | number of training samples for task $t$ |
| $M_t$ | $m_1 + \cdots + m_t$ |
| $c_t$ | total number of classes seen in the first $t$ tasks |
| $T$ | total number of tasks |
| $\mathcal{N}(0,1)$ | Gaussian distribution with mean 0 and variance 1 |
| $\lceil \cdot \rceil$ | the ceiling operator, which maps a number to the closest integer no smaller than it |
| $\boldsymbol{I}_E$ | $E \times E$ identity matrix |
| $\boldsymbol{X}_t$ | $d \times m_t$ matrix, whose columns are output features of pre-trained models |
| $\boldsymbol{P}$ | $E \times d$ random embedding matrix with $\mathcal{N}(0,1)$ entries |
| $\boldsymbol{H}_t$ | Random ReLU features $\text{relu}(\boldsymbol{P}\boldsymbol{X}_t)$ as defined in (1) |
| $\lambda$ | ridge regularization parameter in RanPAC |
| $\boldsymbol{B}_t$ | the matrix whose SVDs are truncated by Algorithm 4, defined in (3) |
| $k_t$ | the number of singular values and vectors preserved for the first $t$ tasks |
| $\tau_{k_t}(\cdot)$ | function that computes the best rank-$k_t$ approximation of a matrix |
| $\mu_k(\cdot)$ | the $k$-th largest eigenvalue of a symmetric matrix |
| $\boldsymbol{U}_{1:t}\boldsymbol{\Sigma}_{1:t}\boldsymbol{V}_{1:t}^\top$ | SVD of $\boldsymbol{H}_{1:t}$ |
| $\overline{\boldsymbol{U}}_{1:t}\overline{\boldsymbol{\Sigma}}_{1:t}\overline{\boldsymbol{V}}_{1:t}^\top$ | SVD of $\tau_{k_t}(\boldsymbol{H}_{1:t})$ |
| $\widetilde{\boldsymbol{U}}_{1:t}, \widetilde{\boldsymbol{\Sigma}}_{1:t}$ | SVD factors of $\boldsymbol{B}_t$ |
| $a_t$ | accumulative error defined in (6) |
| $\gamma_t$ | the eigengap between the present and past, defined in (5) |

## C   IMPLEMENTATION DETAILS FOR LORANPAC

In this section, we give full details of our algorithm for LoRanPAC. Note that Algorithm 1 of the main paper is a concise version of our approach, used to illustrate the methodology at high level.

In Appendix C.1, we introduce Algorithm 2, our implementation of the incremental SVD approach. Note that Algorithm 2 dates back at least to Bunch & Nielsen (1978) and has been applied to image processing, computer vision, and latent semantic indexing (Zha & Simon, 1999; Levey & Lindenbaum, 2000; Brand, 2002; Artac et al., 2002; Ross et al., 2008). Recently a link between continual learning and incremental SVD was built (Peng et al., 2023). However, it has not been applied to the context we consider here to the best of our knowledge, and suitable modifications are needed to incorporate incremental SVD for solving LoRanPAC satisfactorily. For example, we truncate the SVD factors in each continual update as shown Algorithm 2, and the outputs $(\widetilde{U}_{1:t}, \widetilde{\Sigma}_{1:t})$ of Algorithm 2 are not necessarily equal to the top-$k_t$ SVD factors of $H_{1:t}$. It is then our contribution to arm Algorithm 2 with theoretical guarantees (cf. Lemma 1 and Theorem 6).

In Appendix C.2, we introduce Algorithm 4, a continual learning method that stably solves LoRanPAC.

### C.1   INCREMENTAL TRUNCATED SVD

We first explain the design choice as suggested by (2): Should we maintain all SVD factors $\widetilde{U}_{1:t}, \widetilde{\Sigma}_{1:t}$, and $\widetilde{V}_{1:t}$, or should we just maintain the singular values $\widetilde{\Sigma}_{1:t}$ and and left singular vectors $\widetilde{U}_{1:t}$? In the main paper, we suggested taking the latter choice, as we empirically found continually updating all SVD factors $\widetilde{U}_{1:t}, \widetilde{\Sigma}_{1:t}$, and $\widetilde{V}_{1:t}$ lead to large test errors.

We now describe how to update the top $k_t$ SVD factors $\widetilde{U}_{1:t}, \widetilde{\Sigma}_{1:t}$ from the previous estimates $\widetilde{U}_{1:t-1}, \widetilde{\Sigma}_{1:t-1}$ and new data $H_t$. Let $Q_t R_t$ be the QR decomposition of $(I_E - \widetilde{U}_{1:t-1}\widetilde{U}_{1:t-1}^\top)H_t$. Then we have

$$\left[\widetilde{U}_{1:t-1}\widetilde{\Sigma}_{1:t-1},\ H_t\right] = \left[\widetilde{U}_{1:t-1},\ Q_t\right]\begin{bmatrix}\widetilde{\Sigma}_{1:t-1} & \widetilde{U}_{t-1}^\top H_t \\ 0 & R_t\end{bmatrix}.$$

Note that $[\widetilde{U}_{1:t-1},\ Q_t]$ is already orthogonal, we can do a truncated SVD on the smaller $(k_{t-1} + m_t) \times (k_{t-1} + m_t)$ matrix of the right-hand side. The full procedure is summarized below:

---

**Algorithm 2:** Incremental Truncated Singular Value Decomposition

---

1  Input: data matrix $H_t \in \mathbb{R}^{E \times m_t}$ of task $t$, desired output rank $k_t \leq \min\{E, M_t\}$
   ($M_t := m_1 + \cdots + m_t$), truncated SVD factors $\widetilde{U}_{1:t-1} \in \mathbb{R}^{E \times k_{t-1}}$ and $\widetilde{\Sigma}_{1:t-1} \in \mathbb{R}^{k_{t-1} \times k_{t-1}}$
   of the previous $t - 1$ tasks;

2  Compute the QR decomposition $Q_t R_t$ of $(I_E - \widetilde{U}_{1:t-1}\widetilde{U}_{1:t-1}^\top)H_t$;

3  Set $(\Sigma_{\text{tmp}}, U_{\text{tmp}})$ to the top-$k_t$ SVD components of                                     // Algorithm 3

4

$$\begin{bmatrix}\widetilde{\Sigma}_{1:t-1} & \widetilde{U}_{t-1}^\top H_t \\ 0 & R_t\end{bmatrix} \in \mathbb{R}^{(k_{t-1}+m_t)\times(k_{t-1}+m_t)}; \tag{12}$$

5  Set $\widetilde{\Sigma}_{1:t} \leftarrow \Sigma_{\text{tmp}}$ and $\widetilde{U}_{1:t} \leftarrow [\widetilde{U}_{t-1}\ Q_t]U_{\text{tmp}}$;

6  $\widetilde{U}_{1:t} \leftarrow$ The orthogonal factor of QR decomposition of $\widetilde{U}_{1:t}$;        // improve numerical stability

7  Output: $(\widetilde{U}_{1:t}, \widetilde{\Sigma}_{1:t})$;

---

*Remark* 4. Since $[\widetilde{U}_{1:t-1}\ Q_t]$ and $U_{\text{tmp}}$ are orthogonal, $\widetilde{U}_{1:t}$ is expected to be orthogonal as well. However, the multiplication $\widetilde{U}_{1:t} = [\widetilde{U}_{t-1}\ Q_t]U_{\text{tmp}}$ might lose orthogonality due to numerical errors, especially when $t$ gets large. This is fixed by an extra post-processing step that orthogonalizes $\widetilde{U}_{1:t}$.

**Memory Complexity Analysis.** The extra working memory of this approach is roughly:

- $O(Em_t + m_t^2)$, for the QR factors $Q_t R_t$;
- $O((k_{t-1} + m_t)^2)$, for the matrix in (12) and its SVD factors;

---

**Algorithm 3:** Truncated Singular Value Decomposition (TSVD)

1  `Input:` matrix $\boldsymbol{H} \in \mathbb{R}^{E \times m}$ and desired output rank $r \le \min\{E, m\}$;
2  $\text{tmp} \leftarrow \min\{E, m\}$;
3  Compute the SVD $\sigma_1 \boldsymbol{u}_1 \boldsymbol{v}_1^\top + \cdots + \sigma_{\text{tmp}} \boldsymbol{u}_{\text{tmp}} \boldsymbol{v}_{\text{tmp}}^\top$ of $\boldsymbol{H}$;
4  Set $\widetilde{\boldsymbol{\Sigma}} \leftarrow \text{diag}(\sigma_1, \ldots, \sigma_r), \widetilde{\boldsymbol{U}} \leftarrow [\boldsymbol{u}_1, \ldots, \boldsymbol{u}_r]$;
5  `Output:` $(\widetilde{\boldsymbol{U}}, \widetilde{\boldsymbol{\Sigma}})$;

---

Hence, for large $E$, this is less than the $O(E(k_{t-1} + m_t))$ memory used by the direct SVD method.

**Time Complexity Analysis.** The major cost is the SVD of (12), which takes $O((k_{t-1} + m_t)^3)$ time. While in principle the QR orthogonalization for the post-processing of $\widetilde{\boldsymbol{U}}_{1:t}$ takes $O(E(k_{t-1} + m_t)^2)$ time, it is significantly faster than SVD as the constants behind its $O(\cdot)$ is very small. Therefore, one would expect the SVD on the matrix of (12) in $O((k_{t-1} + m_t)^3)$ time should be much faster than the SVD on the matrix $[\widetilde{\boldsymbol{U}}_{1:t-1}\widetilde{\boldsymbol{\Sigma}}_{1:t-1}, \ \boldsymbol{H}_t]$, which needs $O(E(k_{t-1} + m_t)^2)$ time, where $E$ is far larger than $k_{t-1} + m_t$ (e.g., $E = 10^5$ and $k_{t-1} + m_t = 10^4$). This is true on a sequential machine, but their running time difference is not significant for highly parallel GPU implementations in our experience (e.g., computing the inner product between two $E$-dimensional vectors has similar running times to computing the inner product between $(k_{t-1} + m_t)$-dimensional vectors, due to parallelism). Hence, for a parallel implementation, the main advantage of doing SVD on the matrix in (12) is that it takes less working memory than SVD on $[\widetilde{\boldsymbol{U}}_{1:t-1}\widetilde{\boldsymbol{\Sigma}}_{1:t-1}, \ \boldsymbol{H}_t]$.

*Remark* 5. Algorithm 3 can, in fact, be implemented by randomized linear algebra techniques (Halko et al., 2011). Some of these techniques compute by design only the top $k$ SVD factors. Intuitively this could save time and memory if $k$ is very small. One such method is conveniently implemented in PyTorch as well (`torch.svd_lowrank`). However, in our rudimentary attempts at using randomized approaches, we found this PyTorch routine does not seem to be as efficient or accurate as our present implementation (we consistently set truncation percentage $\zeta$ to $25\%$). This observation aligns with the PyTorch document of `torch.svd_lowrank`: *In general, use the full-rank SVD implementation torch.linalg.svd() for dense matrices due to its 10-fold higher performance characteristics. The low-rank SVD will be useful for huge sparse matrices that torch.linalg.svd() cannot handle.* For the moment, we conclude that it needs deeper investigations to see whether randomized techniques are suitable for the continual learning contexts.

## C.2   CONTINUAL SOLVER FOR LORANPAC

The proposed algorithm is shown in Algorithm 4. Here are a few details that we have not yet mentioned in the main paper. First, note that Algorithm 4 formally updates $M_t$ ad $\boldsymbol{J}_t$ continually. At Line 8 of Algorithm 4 we compute the label-feature covariance matrix $\boldsymbol{J}_1 := \boldsymbol{Y}_1 \boldsymbol{H}_1^\top \in \mathbb{R}^{c_1 \times E}$, and then at lines 10 and 11 we update $\boldsymbol{J}_{t-1}$ into $\boldsymbol{J}_t$ via $\boldsymbol{J}_t \leftarrow \boldsymbol{J}_{t-1} + \boldsymbol{J}_{\text{tmp}}$. The attentive reader might find that $\boldsymbol{J}_{t-1}$ is of size $c_{t-1} \times E$ while $\boldsymbol{J}_{\text{tmp}}$ is of size $c_t \times E$. But it could be that $c_{t-1} < c_t$, so it might not make sense to add $\boldsymbol{J}_{t-1}$ and $\boldsymbol{J}_{\text{tmp}}$ as in Line 11. Note that we wrote Line 11 just for simplicity. The implementation would pad $c_t - c_{t-1}$ zero rows to $\boldsymbol{J}_{t-1}$ in a similar fashion to how we extend $\boldsymbol{Y}_{t-1}$ into $\boldsymbol{Y}_t$ when more classes are given, and this is what Line 11 should mean.

Second, we add an extra parameter $r_{\max}$, to control the *maximum allowable rank*, that is the maximum number of columns $\widetilde{\boldsymbol{U}}_{1:t}$ is allowed to have. The purpose is to control the time complexity of Algorithm 4 and allow it to run more efficiently on large datasets such as DomainNet (cf. Tables 7 and 8). We argue both the truncation percentage $\zeta$ and maximum allowable rank $r_{\max}$ are needed: With $\zeta$ alone, the method might run slowly or even exceed the memory for large datasets such as DomainNet (cf. Tables 7 and 8); with $r_{\max}$ alone, truncation is not activated before receiving $r_{\max}$ samples, and numerical instability if it arises, can not be prevented before truncation is in effect. Table 4 gives the values of $\zeta$ and $r_{\max}$ we use for each dataset.

| Dataset | Truncation Percentage $\zeta$ | Embedding Dimension $E$ | Maximum Allowable Rank $r_{\max}$ |
|---|---|---|---|
| CIFAR100 | 25% | $10^5$ | 10000 |
| ImageNet-R | 25% | $10^5$ | 10000 |
| ImageNet-A | 25% | $10^5$ | 10000 |
| CUB-200 | 25% | $10^5$ | 10000 |
| ObjectNet | 25% | $10^5$ | 20000 |
| OmniBenchmark | 25% | $10^5$ | 20000 |
| VTAB$^\dagger$ | 25% | $10^5$ | 10000 |
| StanfordCars | 25% | $10^5$ | 10000 |

Table 4: Hyperparameters we use for each dataset.

---

**Algorithm 4:** Continual Solver of LoRanPAC (concise version in Algorithm 1)

---

1   Input of Task t: Random ReLU features $\boldsymbol{H}_t \in \mathbb{R}^{E \times m_t}$, label matrix $\boldsymbol{Y}_t \in \mathbb{R}^{c_t \times m_t}$,
   truncation percentage $\zeta \in [0, 1]$, maximum allowable rank $r_{\max}$;

2   For $t \leftarrow 1, 2, \ldots$:

3     $M_t \leftarrow M_{t-1} + m_t$;             // update the total number of samples $M_t$

4     $k_t \leftarrow \min\{r_{\max}, (1 - \zeta)M_t\}$;       // perserve $k_t$ SVD factors for the first $t$ tasks

5     Form $\boldsymbol{B}_t$ as per (3);

6     $(\widehat{\boldsymbol{U}}_{1:t}, \widetilde{\boldsymbol{\Sigma}}_{1:t}) \leftarrow$ Top-$k_t$ SVD factors of $\boldsymbol{B}_t$;    // use Algorithm 3 if $t = 1$, or Algorithm 2 if $t > 1$

7     If $t = 1$:                      // Continual update of $\boldsymbol{J}_t := \boldsymbol{Y}_{1:t}\boldsymbol{H}_{1:t}^\top$

8       $\boldsymbol{J}_1 \leftarrow$ Output of Algorithm 5 run with inputs $\boldsymbol{H}_1, \boldsymbol{Y}_1$;    // label-feature covariance of task 1

9     Else:

10       $\boldsymbol{J}_{\text{tmp}} \leftarrow$ Output of Algorithm 5 run with inputs $\boldsymbol{H}_t, \boldsymbol{Y}_t$;    // label-feature covariance of task $t$

11       $\boldsymbol{J}_t \leftarrow \boldsymbol{J}_{t-1} + \boldsymbol{J}_{\text{tmp}}$;

12    Form the linear classifier $\widetilde{\boldsymbol{W}}_t := \boldsymbol{J}_t \left( \widetilde{\boldsymbol{U}}_{1:t}\widetilde{\boldsymbol{\Sigma}}_{1:t}^{-2}\widetilde{\boldsymbol{U}}_{1:t}^\top \right)$;         // cf. (2) and (4)

---

## D   AUXILIARY LEMMAS

The following lemma provides an explicit expression for the difference between the continually truncated factors $\widetilde{\boldsymbol{U}}_{1:t}\widetilde{\boldsymbol{\Sigma}}_{1:t}^2\widetilde{\boldsymbol{U}}_{1:t}^\top$ and covariance $\boldsymbol{H}_{1:t}\boldsymbol{H}_{1:t}^\top$.

**Lemma 1.** *Let $\boldsymbol{B}_t$ be the matrix whose SVDs are truncated by Algorithm 4, as defined in (3). We have $\overline{\boldsymbol{U}}_1 = \widetilde{\boldsymbol{U}}_1$ and $\overline{\boldsymbol{\Sigma}}_1 = \widetilde{\boldsymbol{\Sigma}}_1$. Moreover, we have*

$$\boldsymbol{H}_{1:t}\boldsymbol{H}_{1:t}^\top - \widetilde{\boldsymbol{U}}_{1:t}\widetilde{\boldsymbol{\Sigma}}_{1:t}^2\widetilde{\boldsymbol{U}}_{1:t}^\top = \sum_{i=1}^{t} \left( \boldsymbol{B}_i\boldsymbol{B}_i^\top - \tau_{k_i}\left(\boldsymbol{B}_i\boldsymbol{B}_i^\top\right) \right).$$

*Proof of Lemma 1.* It should be clear that $\overline{\boldsymbol{U}}_1 = \widetilde{\boldsymbol{U}}_1$ and $\overline{\boldsymbol{\Sigma}}_1 = \widetilde{\boldsymbol{\Sigma}}_1$. For every $i = 1, \ldots, t$ we have

$$\widetilde{\boldsymbol{U}}_{1:i}\widetilde{\boldsymbol{\Sigma}}_{1:i}^2\widetilde{\boldsymbol{U}}_{1:i}^\top = \tau_{k_i}\left(\boldsymbol{B}_i\boldsymbol{B}_i^\top\right).$$

and therefore

$$\widetilde{\boldsymbol{U}}_{1:i}\widetilde{\boldsymbol{\Sigma}}_{1:i}^2\widetilde{\boldsymbol{U}}_{1:i}^\top - \boldsymbol{B}_i\boldsymbol{B}_i^\top = \tau_{k_i}\left(\boldsymbol{B}_i\boldsymbol{B}_i^\top\right) - \boldsymbol{B}_i\boldsymbol{B}_i^\top.$$

---

**Algorithm 5:** Compute The Label-Feature Covariance Matrix

---

1  Input: matrix $\boldsymbol{H} = [\boldsymbol{h}_1, \ldots, \boldsymbol{h}_m] \in \mathbb{R}^{E \times m}$ and label matrix $\boldsymbol{Y} \in \mathbb{R}^{c \times m}$;
2  Convert $\boldsymbol{Y}$ into a vector of indices $\boldsymbol{y} = [y_1, \ldots, y_m]$ such that the $i$-th column of $\boldsymbol{Y}$ is the $y_i$-th standard basis vector (i.e., one-hot vector with 1 at position $y_i$);
3  Initialize $\boldsymbol{S} = [\boldsymbol{s}_1, \ldots, \boldsymbol{s}_c]$ to be the $E \times c$ zero matrix;
4  For $i = 1, \ldots, m$:                              // parallel implementation via `torch.Tensor.index_add_`
5      $\boldsymbol{S}_{y_i} \leftarrow \boldsymbol{S}_{y_i} + \boldsymbol{h}_i$;
6  Output: $\boldsymbol{S}^\top$;                              // $\boldsymbol{S}$ is equal to $\boldsymbol{H}\boldsymbol{Y}^\top$

---

A key observation is that summing the above equality over $i = 2, \ldots, t$ yields

$$\sum_{i=2}^{t} \left( \widetilde{\boldsymbol{U}}_{1:i} \widetilde{\boldsymbol{\Sigma}}_{1:i}^2 \widetilde{\boldsymbol{U}}_{1:i}^\top - \boldsymbol{B}_i \boldsymbol{B}_i^\top \right) = \sum_{i=2}^{t} \left( \tau_{k_i} \left( \boldsymbol{B}_i \boldsymbol{B}_i^\top \right) - \boldsymbol{B}_i \boldsymbol{B}_i^\top \right)$$

$$\Leftrightarrow \sum_{i=2}^{t} \left( \widetilde{\boldsymbol{U}}_{1:i} \widetilde{\boldsymbol{\Sigma}}_{1:i}^2 \widetilde{\boldsymbol{U}}_{1:i}^\top - \widetilde{\boldsymbol{U}}_{1:i-1} \widetilde{\boldsymbol{\Sigma}}_{1:i-1}^2 \widetilde{\boldsymbol{U}}_{1:i-1}^\top - \boldsymbol{H}_i \boldsymbol{H}_i^\top \right) = \sum_{i=2}^{t} \left( \tau_{k_i} \left( \boldsymbol{B}_i \boldsymbol{B}_i^\top \right) - \boldsymbol{B}_i \boldsymbol{B}_i^\top \right)$$

$$\Leftrightarrow \widetilde{\boldsymbol{U}}_{1:t} \widetilde{\boldsymbol{\Sigma}}_{1:t}^2 \widetilde{\boldsymbol{U}}_{1:t}^\top - \widetilde{\boldsymbol{U}}_1 \widetilde{\boldsymbol{\Sigma}}_1^2 \widetilde{\boldsymbol{U}}_1^\top - \boldsymbol{H}_{2:t} \boldsymbol{H}_{2:t}^\top = \sum_{i=2}^{t} \left( \tau_{k_i} \left( \boldsymbol{B}_i \boldsymbol{B}_i^\top \right) - \boldsymbol{B}_i \boldsymbol{B}_i^\top \right) \qquad (13)$$

$$\Leftrightarrow \widetilde{\boldsymbol{U}}_{1:t} \widetilde{\boldsymbol{\Sigma}}_{1:t}^2 \widetilde{\boldsymbol{U}}_{1:t}^\top - \boldsymbol{H}_{1:t} \boldsymbol{H}_{1:t}^\top = \overline{\boldsymbol{U}}_1 \overline{\boldsymbol{\Sigma}}_1^2 \overline{\boldsymbol{U}}_1^\top - \boldsymbol{H}_1 \boldsymbol{H}_1^\top + \sum_{i=2}^{t} \left( \tau_{k_i} \left( \boldsymbol{B}_i \boldsymbol{B}_i^\top \right) - \boldsymbol{B}_i \boldsymbol{B}_i^\top \right)$$

$$\Leftrightarrow \widetilde{\boldsymbol{U}}_{1:t} \widetilde{\boldsymbol{\Sigma}}_{1:t}^2 \widetilde{\boldsymbol{U}}_{1:t}^\top - \boldsymbol{H}_{1:t} \boldsymbol{H}_{1:t}^\top = \sum_{i=1}^{t} \left( \tau_{k_i} \left( \boldsymbol{B}_i \boldsymbol{B}_i^\top \right) - \boldsymbol{B}_i \boldsymbol{B}_i^\top \right).$$

The last equality also holds for $t = 1$. This finishes the proof.                              $\square$

The lemma below is a direct consequence of Von Neumann's trace inequality, and its proof is omitted.

**Lemma 2.** *Given two square matrices $A, B$ with $A$ positive semidefinite, we have*

$$\operatorname{tr}(AB) \leq \operatorname{tr}(A) \cdot \|B\|.$$

Lemma 3 presented below is elementary.

**Lemma 3.** *Assume $C$ is a positive semidefinite matrix. Then we have*

$$\operatorname{tr}(DACBD^\top) + \operatorname{tr}(DB^\top CA^\top D^\top) \leq \operatorname{tr}(DACA^\top D^\top) + \operatorname{tr}(DBCB^\top D^\top),$$

*where $A, B, C, D$ are matrices of compatible sizes. Therefore, it holds that*

$$\operatorname{tr}\left(D(A + B)C(A + B)D^\top\right) \leq 2\operatorname{tr}(DACA^\top D^\top) + 2\operatorname{tr}(DBCB^\top D^\top).$$

The following two lemmas provide upper bounds on several terms appearing naturally in our main results.

**Lemma 4.** *Let $\boldsymbol{B}_t$ be defined in (3), $\gamma_t$ in (5), and $a_t$ in (6). Define*

$$\boldsymbol{D}_t := \sum_{i=1}^{t} \left( \boldsymbol{B}_i \boldsymbol{B}_i^\top - \tau_{k_i} \left( \boldsymbol{B}_i \boldsymbol{B}_i^\top \right) \right). \qquad (14)$$

*We have*

$$\left\| \boldsymbol{D}_t \widetilde{\boldsymbol{U}}_{1:t} \widetilde{\boldsymbol{\Sigma}}_{1:t}^{-2} \widetilde{\boldsymbol{U}}_{1:t}^\top \right\| \leq \frac{t-1}{\gamma_t},$$

$$\left\| \widetilde{\boldsymbol{U}}_{1:t} \widetilde{\boldsymbol{\Sigma}}_{1:t}^{-2} \widetilde{\boldsymbol{U}}_{1:t}^\top \boldsymbol{D}_t \right\| \leq \frac{t-1}{\gamma_t},$$

$$\operatorname{tr}\left( \boldsymbol{D}_t \widetilde{\boldsymbol{U}}_{1:t} \widetilde{\boldsymbol{\Sigma}}_{1:t}^{-2} \widetilde{\boldsymbol{U}}_{1:t}^\top \right) \leq \frac{1}{\gamma_t} \min\left\{ M_{t-1} - k_{t-1}, (t-1)k_t \right\},$$

$$\operatorname{tr}\left( \boldsymbol{D}_t \widetilde{\boldsymbol{U}}_{1:t} \widetilde{\boldsymbol{\Sigma}}_{1:t}^{-4} \widetilde{\boldsymbol{U}}_{1:t}^\top \right) \leq \frac{1}{\mu_{k_t}\left( \boldsymbol{B}_t \boldsymbol{B}_t^\top \right)} \cdot \frac{1}{\gamma_t} \min\left\{ M_{t-1} - k_{t-1}, (t-1)k_t \right\}.$$

*Proof of Lemma 4.* It follows from definition that

$$\left(\boldsymbol{B}_i\boldsymbol{B}_i^\top - \tau_{k_i}\left(\boldsymbol{B}_i\boldsymbol{B}_i^\top\right)\right)\widetilde{\boldsymbol{U}}_{1:t} = 0,$$

hence $\boldsymbol{D}_t\widetilde{\boldsymbol{U}}_{1:t} = \boldsymbol{D}_{t-1}\widetilde{\boldsymbol{U}}_{1:t}$. This means

$$\left\|\boldsymbol{D}_t\widetilde{\boldsymbol{U}}_{1:t}\widetilde{\boldsymbol{\Sigma}}_{1:t}^{-2}\widetilde{\boldsymbol{U}}_{1:t}^\top\right\| = \left\|\boldsymbol{D}_{t-1}\widetilde{\boldsymbol{U}}_{1:t}\widetilde{\boldsymbol{\Sigma}}_{1:t}^{-2}\widetilde{\boldsymbol{U}}_{1:t}^\top\right\|$$

$$\leq \|\boldsymbol{D}_{t-1}\| \cdot \left\|\widetilde{\boldsymbol{U}}_{1:t}\widetilde{\boldsymbol{\Sigma}}_{1:t}^{-2}\widetilde{\boldsymbol{U}}_{1:t}^\top\right\|$$

$$= \|\boldsymbol{D}_{t-1}\| \cdot \frac{1}{\mu_{k_t}\left(\boldsymbol{B}_t\boldsymbol{B}_t^\top\right)},$$

where the last equality follows by definition. On the other hand, we have

$$\|\boldsymbol{D}_{t-1}\| \leq \sum_{i=1}^{t-1}\mu_{k_i+1}\left(\boldsymbol{B}_i\boldsymbol{B}_i^\top\right) \leq \frac{(t-1)}{\gamma_t}\mu_{k_t}\left(\boldsymbol{B}_t\boldsymbol{B}_t^\top\right).$$

Combining the above proves the first required equality. The second inequality follows similarly.

For the final trace inequality, we have ($k_0 := 0$)

$$\mathrm{tr}\left(\boldsymbol{D}_t\widetilde{\boldsymbol{U}}_{1:t}\widetilde{\boldsymbol{\Sigma}}_{1:t}^{-2}\widetilde{\boldsymbol{U}}_{1:t}^\top\right) = \mathrm{tr}\left(\boldsymbol{D}_{t-1}\widetilde{\boldsymbol{U}}_{1:t}\widetilde{\boldsymbol{\Sigma}}_{1:t}^{-2}\widetilde{\boldsymbol{U}}_{1:t}^\top\right)$$

$$\overset{(i)}{\leq} \mathrm{tr}(\boldsymbol{D}_{t-1}) \cdot \left\|\widetilde{\boldsymbol{U}}_{1:t}\widetilde{\boldsymbol{\Sigma}}_{1:t}^{-2}\widetilde{\boldsymbol{U}}_{1:t}^\top\right\|$$

$$= \left(\sum_{i=1}^{t-1}\sum_{j=k_i+1}^{m_i+k_{i-1}}\mu_j\left(\boldsymbol{B}_i\boldsymbol{B}_i^\top\right)\right) \cdot \frac{1}{\mu_{k_t}\left(\boldsymbol{B}_t\boldsymbol{B}_t^\top\right)}$$

$$\leq \frac{1}{\gamma_t}\sum_{i=1}^{t-1}(m_i + k_{i-1} - k_i)$$

$$= \frac{1}{\gamma_t}(M_{t-1} - k_{t-1}),$$

where (i) holds as $\boldsymbol{D}_{t-1}$ is positive semidefinite (cf. Lemma 2).

We can also bound $\mathrm{tr}\left(\boldsymbol{D}_t\widetilde{\boldsymbol{U}}_{1:t}\widetilde{\boldsymbol{\Sigma}}_{1:t}^{-2}\widetilde{\boldsymbol{U}}_{1:t}^\top\right)$ alternatively as follows:

$$\mathrm{tr}\left(\boldsymbol{D}_t\widetilde{\boldsymbol{U}}_{1:t}\widetilde{\boldsymbol{\Sigma}}_{1:t}^{-2}\widetilde{\boldsymbol{U}}_{1:t}^\top\right) = \mathrm{tr}\left(\boldsymbol{D}_{t-1}\widetilde{\boldsymbol{U}}_{1:t}\widetilde{\boldsymbol{\Sigma}}_{1:t}^{-2}\widetilde{\boldsymbol{U}}_{1:t}^\top\right)$$

$$\leq \|\boldsymbol{D}_{t-1}\| \cdot \mathrm{tr}(\widetilde{\boldsymbol{U}}_{1:t}\widetilde{\boldsymbol{\Sigma}}_{1:t}^{-2}\widetilde{\boldsymbol{U}}_{1:t}^\top)$$

$$\leq \frac{(t-1)k_t}{\gamma_t}$$

Combining the two bounds on $\mathrm{tr}\left(\boldsymbol{D}_t\widetilde{\boldsymbol{U}}_{1:t}\widetilde{\boldsymbol{\Sigma}}_{1:t}^{-2}\widetilde{\boldsymbol{U}}_{1:t}^\top\right)$ proves the third inequality. The fourth inequality follows similarly. □

**Lemma 5.** *Using the notations in Lemma 4, we have*

$$\mathrm{tr}\left(\boldsymbol{D}_t\widetilde{\boldsymbol{U}}_{1:t}\widetilde{\boldsymbol{\Sigma}}_{1:t}^{-2}\widetilde{\boldsymbol{U}}_{1:t}^\top\boldsymbol{D}_t\widetilde{\boldsymbol{U}}_{1:t}\widetilde{\boldsymbol{\Sigma}}_{1:t}^{-2}\widetilde{\boldsymbol{U}}_{1:t}^\top\right) \leq \frac{t-1}{\gamma_t^2}\min\left\{M_{t-1} - k_{t-1}, (t-1)k_t\right\},$$

$$\left\|\boldsymbol{D}_t\widetilde{\boldsymbol{U}}_{1:t}\widetilde{\boldsymbol{\Sigma}}_{1:t}^{-2}\widetilde{\boldsymbol{U}}_{1:t}^\top\boldsymbol{H}_{1:t}\boldsymbol{H}_{1:t}^\top\widetilde{\boldsymbol{U}}_{1:t}\widetilde{\boldsymbol{\Sigma}}_{1:t}^{-2}\widetilde{\boldsymbol{U}}_{1:t}^\top\boldsymbol{D}_t\right\| \leq a_{t-1} \cdot \left(\frac{(t-1)^2}{\gamma_t^2} + \frac{t-1}{\gamma_t}\right),$$

$$\left\|(\boldsymbol{I}_E - \widetilde{\boldsymbol{U}}_{1:t}\widetilde{\boldsymbol{U}}_{1:t}^\top)\boldsymbol{H}_{1:t}\boldsymbol{H}_{1:t}^\top(\boldsymbol{I}_E - \widetilde{\boldsymbol{U}}_{1:t}\widetilde{\boldsymbol{U}}_{1:t}^\top)\right\| \leq a_t,$$

$$\left\|\boldsymbol{H}_{1:t}^\top\widetilde{\boldsymbol{U}}_{1:t}\widetilde{\boldsymbol{\Sigma}}_{1:t}^{-2}\widetilde{\boldsymbol{U}}_{1:t}^\top\boldsymbol{H}_{1:t}\right\|_F^2 \leq \left(\frac{t-1}{\gamma_t^2} + \frac{2}{\gamma_t}\right)\min\left\{M_{t-1} - k_{t-1}, (t-1)k_t\right\} + k_t,$$

$$\mathrm{tr}\left(\widetilde{\boldsymbol{U}}_{1:t}\widetilde{\boldsymbol{\Sigma}}_{1:t}^{-2}\widetilde{\boldsymbol{U}}_{1:t}^\top\boldsymbol{H}_{1:t}\boldsymbol{H}_{1:t}^\top\widetilde{\boldsymbol{U}}_{1:t}\widetilde{\boldsymbol{\Sigma}}_{1:t}^{-2}\widetilde{\boldsymbol{U}}_{1:t}^\top\right) \leq \frac{1}{\mu_{k_t}\left(\boldsymbol{B}_t\boldsymbol{B}_t^\top\right)} \cdot \left(\frac{\min\left\{M_{t-1} - k_{t-1}, (t-1)k_t\right\}}{\gamma_t} + k_t\right).$$

*Proof of Lemma 5.* Since $\boldsymbol{D}_{t-1}$ is positive semidefinite, let $\boldsymbol{L}_{t-1}\boldsymbol{L}_{t-1}^\top$ be its Cholesky decomposition. Then we have

$$\operatorname{tr}\left(\boldsymbol{D}_t\widetilde{\boldsymbol{U}}_{1:t}\widetilde{\boldsymbol{\Sigma}}_{1:t}^{-2}\widetilde{\boldsymbol{U}}_{1:t}^\top\boldsymbol{D}_t\widetilde{\boldsymbol{U}}_{1:t}\widetilde{\boldsymbol{\Sigma}}_{1:t}^{-2}\widetilde{\boldsymbol{U}}_{1:t}^\top\right)$$

$$= \operatorname{tr}\left(\boldsymbol{D}_{t-1}\widetilde{\boldsymbol{U}}_{1:t}\widetilde{\boldsymbol{\Sigma}}_{1:t}^{-2}\widetilde{\boldsymbol{U}}_{1:t}^\top\boldsymbol{D}_{t-1}\widetilde{\boldsymbol{U}}_{1:t}\widetilde{\boldsymbol{\Sigma}}_{1:t}^{-2}\widetilde{\boldsymbol{U}}_{1:t}^\top\right)$$

$$= \operatorname{tr}\left(\boldsymbol{L}_{t-1}\boldsymbol{L}_{t-1}^\top\widetilde{\boldsymbol{U}}_{1:t}\widetilde{\boldsymbol{\Sigma}}_{1:t}^{-2}\widetilde{\boldsymbol{U}}_{1:t}^\top\boldsymbol{L}_{t-1}\boldsymbol{L}_{t-1}^\top\widetilde{\boldsymbol{U}}_{1:t}\widetilde{\boldsymbol{\Sigma}}_{1:t}^{-2}\widetilde{\boldsymbol{U}}_{1:t}^\top\right)$$

$$= \operatorname{tr}\left(\boldsymbol{L}_{t-1}^\top\widetilde{\boldsymbol{U}}_{1:t}\widetilde{\boldsymbol{\Sigma}}_{1:t}^{-2}\widetilde{\boldsymbol{U}}_{1:t}^\top\boldsymbol{L}_{t-1}\boldsymbol{L}_{t-1}^\top\widetilde{\boldsymbol{U}}_{1:t}\widetilde{\boldsymbol{\Sigma}}_{1:t}^{-2}\widetilde{\boldsymbol{U}}_{1:t}^\top\boldsymbol{L}_{t-1}\right)$$

$$\overset{(i)}{\leq} \operatorname{tr}\left(\boldsymbol{L}_{t-1}^\top\widetilde{\boldsymbol{U}}_{1:t}\widetilde{\boldsymbol{\Sigma}}_{1:t}^{-2}\widetilde{\boldsymbol{U}}_{1:t}^\top\boldsymbol{L}_{t-1}\right)\cdot\left\|\boldsymbol{L}_{t-1}^\top\widetilde{\boldsymbol{U}}_{1:t}\widetilde{\boldsymbol{\Sigma}}_{1:t}^{-2}\widetilde{\boldsymbol{U}}_{1:t}^\top\boldsymbol{L}_{t-1}\right\|$$

$$= \operatorname{tr}\left(\boldsymbol{D}_{t-1}\widetilde{\boldsymbol{U}}_{1:t}\widetilde{\boldsymbol{\Sigma}}_{1:t}^{-2}\widetilde{\boldsymbol{U}}_{1:t}^\top\right)\cdot\left\|\boldsymbol{L}_{t-1}^\top\widetilde{\boldsymbol{U}}_{1:t}\widetilde{\boldsymbol{\Sigma}}_{1:t}^{-2}\widetilde{\boldsymbol{U}}_{1:t}^\top\boldsymbol{L}_{t-1}\right\|$$

$$\overset{(ii)}{\leq} \frac{1}{\gamma_t}\min\left\{M_{t-1}-k_{t-1},(t-1)k_t\right\}\cdot\left\|\boldsymbol{L}_{t-1}^\top\widetilde{\boldsymbol{U}}_{1:t}\widetilde{\boldsymbol{\Sigma}}_{1:t}^{-2}\widetilde{\boldsymbol{U}}_{1:t}^\top\boldsymbol{L}_{t-1}\right\|.$$

In the above, (i) follows from Lemma 2, and (ii) follows from Lemma 4. Continuing with the above inequality, we have

$$\left\|\boldsymbol{L}_{t-1}^\top\widetilde{\boldsymbol{U}}_{1:t}\widetilde{\boldsymbol{\Sigma}}_{1:t}^{-2}\widetilde{\boldsymbol{U}}_{1:t}^\top\boldsymbol{L}_{t-1}\right\| \leq \|\boldsymbol{L}_{t-1}\|^2\cdot\frac{1}{\mu_{k_t}(\boldsymbol{B}_t\boldsymbol{B}_t^\top)}$$

$$= \|\boldsymbol{D}_{t-1}\|\cdot\frac{1}{\mu_{k_t}(\boldsymbol{B}_t\boldsymbol{B}_t^\top)}$$

$$\leq \frac{t-1}{\gamma_t}.$$

Recall the fact $\boldsymbol{D}_t\widetilde{\boldsymbol{U}}_{1:t} = \boldsymbol{D}_{t-1}\widetilde{\boldsymbol{U}}_{1:t}$. The second inequality in Lemma 5 can be proved as follows:

$$\left\|\boldsymbol{D}_t\widetilde{\boldsymbol{U}}_{1:t}\widetilde{\boldsymbol{\Sigma}}_{1:t}^{-2}\widetilde{\boldsymbol{U}}_{1:t}^\top\boldsymbol{H}_{1:t}\boldsymbol{H}_{1:t}^\top\widetilde{\boldsymbol{U}}_{1:t}\widetilde{\boldsymbol{\Sigma}}_{1:t}^{-2}\widetilde{\boldsymbol{U}}_{1:t}^\top\boldsymbol{D}_t\right\|$$

$$\overset{(i)}{=} \left\|\boldsymbol{D}_t\widetilde{\boldsymbol{U}}_{1:t}\widetilde{\boldsymbol{\Sigma}}_{1:t}^{-2}\widetilde{\boldsymbol{U}}_{1:t}^\top(\boldsymbol{D}_t+\widetilde{\boldsymbol{U}}_{1:t}\widetilde{\boldsymbol{\Sigma}}_{1:t}^2\widetilde{\boldsymbol{U}}_{1:t}^\top)\widetilde{\boldsymbol{U}}_{1:t}\widetilde{\boldsymbol{\Sigma}}_{1:t}^{-2}\widetilde{\boldsymbol{U}}_{1:t}^\top\boldsymbol{D}_t\right\|$$

$$= \left\|\boldsymbol{D}_t\widetilde{\boldsymbol{U}}_{1:t}\widetilde{\boldsymbol{\Sigma}}_{1:t}^{-2}\widetilde{\boldsymbol{U}}_{1:t}^\top\boldsymbol{D}_t\widetilde{\boldsymbol{U}}_{1:t}\widetilde{\boldsymbol{\Sigma}}_{1:t}^{-2}\widetilde{\boldsymbol{U}}_{1:t}^\top\boldsymbol{D}_t + \boldsymbol{D}_t\widetilde{\boldsymbol{U}}_{1:t}\widetilde{\boldsymbol{\Sigma}}_{1:t}^{-2}\widetilde{\boldsymbol{U}}_{1:t}^\top\boldsymbol{D}_t\right\|$$

$$\leq \|\boldsymbol{D}_{t-1}\|\cdot\left(\left\|\widetilde{\boldsymbol{U}}_{1:t}\widetilde{\boldsymbol{\Sigma}}_{1:t}^{-2}\widetilde{\boldsymbol{U}}_{1:t}^\top\boldsymbol{D}_t\widetilde{\boldsymbol{U}}_{1:t}\widetilde{\boldsymbol{\Sigma}}_{1:t}^{-2}\widetilde{\boldsymbol{U}}_{1:t}^\top\boldsymbol{D}_t\right\| + \left\|\widetilde{\boldsymbol{U}}_{1:t}\widetilde{\boldsymbol{\Sigma}}_{1:t}^{-2}\widetilde{\boldsymbol{U}}_{1:t}^\top\boldsymbol{D}_t\right\|\right)$$

$$\leq a_{t-1}\cdot\left(\frac{(t-1)^2}{\gamma_t^2}+\frac{t-1}{\gamma_t}\right).$$

In the above, (i) follows from Lemma 1.

The third inequality is proved as follows:

$$\left\|(\boldsymbol{I}_E-\widetilde{\boldsymbol{U}}_{1:t}\widetilde{\boldsymbol{U}}_{1:t}^\top)\boldsymbol{H}_{1:t}\boldsymbol{H}_{1:t}^\top(\boldsymbol{I}_E-\widetilde{\boldsymbol{U}}_{1:t}\widetilde{\boldsymbol{U}}_{1:t}^\top)\right\|$$

$$= \left\|(\boldsymbol{I}_E-\widetilde{\boldsymbol{U}}_{1:t}\widetilde{\boldsymbol{U}}_{1:t}^\top)(\boldsymbol{H}_{1:t}\boldsymbol{H}_{1:t}^\top-\widetilde{\boldsymbol{U}}_{1:t}\widetilde{\boldsymbol{\Sigma}}^2\widetilde{\boldsymbol{U}}_{1:t}^\top)(\boldsymbol{I}_E-\widetilde{\boldsymbol{U}}_{1:t}\widetilde{\boldsymbol{U}}_{1:t}^\top)\right\|$$

$$= \left\|(\boldsymbol{I}_E-\widetilde{\boldsymbol{U}}_{1:t}\widetilde{\boldsymbol{U}}_{1:t}^\top)\boldsymbol{D}_t(\boldsymbol{I}_E-\widetilde{\boldsymbol{U}}_{1:t}\widetilde{\boldsymbol{U}}_{1:t}^\top)\right\|$$

$$\leq \|\boldsymbol{D}_t\| = a_t.$$

We now prove the fourth inequality:

$$\left\|\boldsymbol{H}_{1:t}^\top\widetilde{\boldsymbol{U}}_{1:t}\widetilde{\boldsymbol{\Sigma}}_{1:t}^{-2}\widetilde{\boldsymbol{U}}_{1:t}^\top\boldsymbol{H}_{1:t}\right\|_F^2$$

$$= \operatorname{tr}\left(\boldsymbol{H}_{1:t}\boldsymbol{H}_{1:t}^\top\widetilde{\boldsymbol{U}}_{1:t}\widetilde{\boldsymbol{\Sigma}}_{1:t}^{-2}\widetilde{\boldsymbol{U}}_{1:t}^\top\boldsymbol{H}_{1:t}\boldsymbol{H}_{1:t}^\top\widetilde{\boldsymbol{U}}_{1:t}\widetilde{\boldsymbol{\Sigma}}_{1:t}^{-2}\widetilde{\boldsymbol{U}}_{1:t}^\top\right)$$

$$\overset{(i)}{=} \operatorname{tr}\left((\boldsymbol{D}_t\widetilde{\boldsymbol{U}}_{1:t}\widetilde{\boldsymbol{\Sigma}}_{1:t}^{-2}\widetilde{\boldsymbol{U}}_{1:t}^\top + \widetilde{\boldsymbol{U}}_{1:t}\widetilde{\boldsymbol{U}}_{1:t}^\top)(\boldsymbol{D}_t\widetilde{\boldsymbol{U}}_{1:t}\widetilde{\boldsymbol{\Sigma}}_{1:t}^{-2}\widetilde{\boldsymbol{U}}_{1:t}^\top + \widetilde{\boldsymbol{U}}_{1:t}\widetilde{\boldsymbol{U}}_{1:t}^\top)\right)$$

$$= \operatorname{tr}\left(\boldsymbol{D}_t\widetilde{\boldsymbol{U}}_{1:t}\widetilde{\boldsymbol{\Sigma}}_{1:t}^{-2}\widetilde{\boldsymbol{U}}_{1:t}^\top\boldsymbol{D}_t\widetilde{\boldsymbol{U}}_{1:t}\widetilde{\boldsymbol{\Sigma}}_{1:t}^{-2}\widetilde{\boldsymbol{U}}_{1:t}^\top + 2\boldsymbol{D}_t\widetilde{\boldsymbol{U}}_{1:t}\widetilde{\boldsymbol{\Sigma}}_{1:t}^{-2}\widetilde{\boldsymbol{U}}_{1:t}^\top\right) + k_t$$

$$\overset{(ii)}{\leq} \left(\frac{t-1}{\gamma_t^2}+\frac{2}{\gamma_t}\right)\min\left\{M_{t-1}-k_{t-1},(t-1)k_t\right\} + k_t.$$

In the above, (i) follows from Lemma 1, and (ii) follows from Lemma 4 and the first inequality we just proved for Lemma 5.

The fifth inequality can be proved as follows:

$$\operatorname{tr}\left(\widetilde{U}_{1:t}\widetilde{\Sigma}_{1:t}^{-2}\widetilde{U}_{1:t}^{\top}H_{1:t}H_{1:t}^{\top}\widetilde{U}_{1:t}\widetilde{\Sigma}_{1:t}^{-2}\widetilde{U}_{1:t}^{\top}\right)$$

$$\overset{(i)}{=}\operatorname{tr}\left(\widetilde{U}_{1:t}\widetilde{\Sigma}_{1:t}^{-2}\widetilde{U}_{1:t}^{\top}(D_t\widetilde{U}_{1:t}\widetilde{\Sigma}_{1:t}^{-2}\widetilde{U}_{1:t}^{\top}+\widetilde{U}_{1:t}\widetilde{U}_{1:t}^{\top})\right)$$

$$=\operatorname{tr}\left(D_t\widetilde{U}_{1:t}\widetilde{\Sigma}_{1:t}^{-4}\widetilde{U}_{1:t}^{\top}\right)+\operatorname{tr}\left(\widetilde{\Sigma}_{1:t}^{-2}\right)$$

$$\overset{(ii)}{\leq}\frac{1}{\mu_{k_t}\left(B_tB_t^{\top}\right)}\cdot\frac{1}{\gamma_t}\min\left\{M_{t-1}-k_{t-1},(t-1)k_t\right\}+\frac{k_t}{\mu_{k_t}\left(B_tB_t^{\top}\right)}.$$

Here, (i) holds as a result of Lemma 1 and (ii) follows from Lemma 4. $\qquad\square$

**Lemma 6.** *Using the notations in Lemma 4, we have for any $W$ that*

$$\left\|W(H_{1:t}H_{1:t}^{\top}\widetilde{U}_{1:t}\widetilde{\Sigma}_{1:t}^{-2}\widetilde{U}_{1:t}^{\top}-I_{M_t})H_{1:t}\right\|_{\mathrm{F}}^{2}\leq 2\cdot\|W\|_{\mathrm{F}}^{2}\left(a_t+\frac{a_{t-1}(t-1)}{\gamma_t}+\frac{a_{t-1}(t-1)^2}{\gamma_t^2}\right).$$

*Proof.* We have

$$\left\|W(H_{1:t}H_{1:t}^{\top}\widetilde{U}_{1:t}\widetilde{\Sigma}_{1:t}^{-2}\widetilde{U}_{1:t}^{\top}-I_{M_t})H_{1:t}\right\|_{\mathrm{F}}^{2}$$

$$=\operatorname{tr}\left(W(H_{1:t}H_{1:t}^{\top}\widetilde{U}_{1:t}\widetilde{\Sigma}_{1:t}^{-2}\widetilde{U}_{1:t}^{\top}-I_{M_t})H_{1:t}H_{1:t}^{\top}(\widetilde{U}_{1:t}\widetilde{\Sigma}_{1:t}^{-2}\widetilde{U}_{1:t}^{\top}H_{1:t}H_{1:t}^{\top}-I_{M_t})(W)^{\top}\right)$$

$$\overset{(i)}{\leq}\operatorname{tr}\left(W(D_t\widetilde{U}_{1:t}\widetilde{\Sigma}_{1:t}^{-2}\widetilde{U}_{1:t}^{\top}+\widetilde{U}_{1:t}\widetilde{U}_{1:t}^{\top}-I_E)H_{1:t}H_{1:t}^{\top}(\widetilde{U}_{1:t}\widetilde{\Sigma}_{1:t}^{-2}\widetilde{U}_{1:t}^{\top}D_t+\widetilde{U}_{1:t}\widetilde{U}_{1:t}^{\top}-I_E)(W)^{\top}\right)$$

$$\overset{(ii)}{\leq}2\operatorname{tr}\left(W(\widetilde{U}_{1:t}\widetilde{U}_{1:t}^{\top}-I_E)H_{1:t}H_{1:t}^{\top}(\widetilde{U}_{1:t}\widetilde{U}_{1:t}^{\top}-I_E)(W)^{\top}\right)$$

$$\quad+2\operatorname{tr}\left(WD_t\widetilde{U}_{1:t}\widetilde{\Sigma}_{1:t}^{-2}\widetilde{U}_{1:t}^{\top}H_{1:t}H_{1:t}^{\top}\widetilde{U}_{1:t}\widetilde{\Sigma}_{1:t}^{-2}\widetilde{U}_{1:t}^{\top}D_t(W)^{\top}\right)$$

$$\overset{(iii)}{\leq}2\cdot\|W\|_{\mathrm{F}}^{2}\cdot\left\|(I_E-\widetilde{U}_{1:t}\widetilde{U}_{1:t}^{\top})H_{1:t}H_{1:t}^{\top}(I_E-\widetilde{U}_{1:t}\widetilde{U}_{1:t}^{\top})\right\|$$

$$\quad+2\cdot\|W\|_{\mathrm{F}}^{2}\cdot\left\|D_t\widetilde{U}_{1:t}\widetilde{\Sigma}_{1:t}^{-2}\widetilde{U}_{1:t}^{\top}H_{1:t}H_{1:t}^{\top}\widetilde{U}_{1:t}\widetilde{\Sigma}_{1:t}^{-2}\widetilde{U}_{1:t}^{\top}D_t\right\|$$

$$\overset{(iv)}{\leq}2\cdot\|W\|_{\mathrm{F}}^{2}\cdot a_t+2\cdot\|W\|_{\mathrm{F}}^{2}\cdot a_{t-1}\cdot\left(\frac{(t-1)^2}{\gamma_t^2}+\frac{t-1}{\gamma_t}\right)$$

$$=2\cdot\|W\|_{\mathrm{F}}^{2}\left(a_t+\frac{a_{t-1}(t-1)}{\gamma_t}+\frac{a_{t-1}(t-1)^2}{\gamma_t^2}\right).$$

In the above, (i) follows from Lemma 1, (ii) from Lemma 3, (iii) from Lemma 2, and (iv) from Lemma 5. The proof is complete. $\qquad\square$

# E PROOFS OF THEOREM 1 AND THEOREM 2

*Proof of Theorem 1.* Let $I_{M_t}$ be the $M_t\times M_t$ identity matrix. The training loss can be written as

$$\left\|\widetilde{W}_tH_{1:t}-Y_{1:t}\right\|_{\mathrm{F}}^{2}$$

$$=\left\|Y_{1:t}H_{1:t}^{\top}\widetilde{U}_{1:t}\widetilde{\Sigma}_{1:t}^{-2}\widetilde{U}_{1:t}^{\top}H_{1:t}-Y_{1:t}\right\|_{\mathrm{F}}^{2}$$

$$=\left\|Y_{1:t}(H_{1:t}^{\top}\widetilde{U}_{1:t}\widetilde{\Sigma}_{1:t}^{-2}\widetilde{U}_{1:t}^{\top}H_{1:t}-I_{M_t})\right\|_{\mathrm{F}}^{2}$$

$$=\left\|(W_t^*H_{1:t}+\mathcal{E}_{1:t})(H_{1:t}^{\top}\widetilde{U}_{1:t}\widetilde{\Sigma}_{1:t}^{-2}\widetilde{U}_{1:t}^{\top}H_{1:t}-I_{M_t})\right\|_{\mathrm{F}}^{2}$$

$$\leq 2\cdot\left\|W_t^*(H_{1:t}H_{1:t}^{\top}\widetilde{U}_{1:t}\widetilde{\Sigma}_{1:t}^{-2}\widetilde{U}_{1:t}^{\top}-I_{M_t})H_{1:t}\right\|_{\mathrm{F}}^{2}+2\cdot\left\|\mathcal{E}_{1:t}(H_{1:t}^{\top}\widetilde{U}_{1:t}\widetilde{\Sigma}_{1:t}^{-2}\widetilde{U}_{1:t}^{\top}H_{1:t}-I_{M_t})\right\|_{\mathrm{F}}^{2}.$$

We can now bound the first term by Lemma 6 as follows:

$$\left\|W_t^*(H_{1:t}H_{1:t}^{\top}\widetilde{U}_{1:t}\widetilde{\Sigma}_{1:t}^{-2}\widetilde{U}_{1:t}^{\top}-I_{M_t})H_{1:t}\right\|_{\mathrm{F}}^{2}\leq 2\cdot\|W_t^*\|_{\mathrm{F}}^{2}\left(a_t+\frac{a_{t-1}(t-1)}{\gamma_t}+\frac{a_{t-1}(t-1)^2}{\gamma_t^2}\right).$$

The second term is bounded above as follows:

$$2 \cdot \left\| \mathcal{E}_{1:t}(\boldsymbol{H}_{1:t}^\top \widetilde{\boldsymbol{U}}_{1:t} \widetilde{\boldsymbol{\Sigma}}_{1:t}^{-2} \widetilde{\boldsymbol{U}}_{1:t}^\top \boldsymbol{H}_{1:t} - \boldsymbol{I}_{M_t}) \right\|_{\mathrm{F}}^2$$

$$\leq 2 \cdot \|\mathcal{E}_{1:t}\|^2 \cdot \left\| (\boldsymbol{H}_{1:t}^\top \widetilde{\boldsymbol{U}}_{1:t} \widetilde{\boldsymbol{\Sigma}}_{1:t}^{-2} \widetilde{\boldsymbol{U}}_{1:t}^\top \boldsymbol{H}_{1:t} - \boldsymbol{I}_{M_t}) \right\|_{\mathrm{F}}^2$$

$$\leq 2 \cdot \|\mathcal{E}_{1:t}\|^2 \cdot \left( M_t - k_t + \frac{t-1}{\gamma_t^2} \min\{M_{t-1} - k_{t-1}, (t-1)k_t\} \right),$$

where the first inequality follows from Lemma 2 and the last inequality from Proposition 1. $\qquad\square$

*Proof of Theorem 2.* Define $\boldsymbol{D}_t := \sum_{i=1}^t \left( \boldsymbol{B}_i \boldsymbol{B}_i^\top - \tau_{k_i}(\boldsymbol{B}_i \boldsymbol{B}_i^\top) \right)$. Note that $\boldsymbol{D}_t$ is a symmetric and positive semi-definite matrix.

Note that we have

$$\mathbb{E}_{\boldsymbol{h}} \left[ \left\| \widetilde{\boldsymbol{W}}_t \boldsymbol{h} - \boldsymbol{y} \right\|^2 \right] = \mathbb{E}_{\boldsymbol{h}} \left[ \left\| \widetilde{\boldsymbol{W}}_t \boldsymbol{h} - \boldsymbol{W}_t^* \boldsymbol{h} - \boldsymbol{\epsilon} \right\|^2 \right]$$

$$= 2 \cdot \mathbb{E}_{\boldsymbol{h}} \left[ \left\| \widetilde{\boldsymbol{W}}_t \boldsymbol{h} - \boldsymbol{W}_t^* \boldsymbol{h} \right\|^2 \right] + 2 \cdot \|\boldsymbol{\epsilon}\|^2,$$

so we next focus on bounding $\mathbb{E}_{\boldsymbol{h}} \left[ \left\| \widetilde{\boldsymbol{W}}_t \boldsymbol{h} - \boldsymbol{W}_t^* \boldsymbol{h} \right\|^2 \right]$. With the $E \times E$ identity matrix $\boldsymbol{I}_E$ and $\boldsymbol{\Lambda} := \mathbb{E}[\boldsymbol{h}\boldsymbol{h}^\top]$, we have

$$\mathbb{E}_{\boldsymbol{h}} \left[ \|\widetilde{\boldsymbol{W}}_t \boldsymbol{h} - \boldsymbol{W}_t^* \boldsymbol{h}\|^2 \right]$$

$$= \mathbb{E}_{\boldsymbol{h}} \left[ \|\boldsymbol{Y}_{1:t} \boldsymbol{H}_{1:t}^\top \widetilde{\boldsymbol{U}}_{1:t} \widetilde{\boldsymbol{\Sigma}}_{1:t}^{-2} \widetilde{\boldsymbol{U}}_{1:t}^\top \boldsymbol{h} - \boldsymbol{W}_t^* \boldsymbol{h}\|^2 \right]$$

$$= \mathbb{E}_{\boldsymbol{h}} \left[ \|(\boldsymbol{W}_t^* \boldsymbol{H}_{1:t} + \mathcal{E}_{1:t}) \boldsymbol{H}_{1:t}^\top \widetilde{\boldsymbol{U}}_{1:t} \widetilde{\boldsymbol{\Sigma}}_{1:t}^{-2} \widetilde{\boldsymbol{U}}_{1:t}^\top \boldsymbol{h} - \boldsymbol{W}_t^* \boldsymbol{h}\|^2 \right]$$

$$= \mathbb{E}_{\boldsymbol{h}} \left[ \|(\boldsymbol{W}_t^* \boldsymbol{H}_{1:t} + \mathcal{E}_{1:t}) \boldsymbol{H}_{1:t}^\top \widetilde{\boldsymbol{U}}_{1:t} \widetilde{\boldsymbol{\Sigma}}_{1:t}^{-2} \widetilde{\boldsymbol{U}}_{1:t}^\top \boldsymbol{h} - \boldsymbol{W}_t^* \boldsymbol{h}\|^2 \right]$$

$$\leq 2 \cdot \mathbb{E}_{\boldsymbol{h}} \left[ \|\boldsymbol{W}_t^* \boldsymbol{H}_{1:t} \boldsymbol{H}_{1:t}^\top \widetilde{\boldsymbol{U}}_{1:t} \widetilde{\boldsymbol{\Sigma}}_{1:t}^{-2} \widetilde{\boldsymbol{U}}_{1:t}^\top \boldsymbol{h} - \boldsymbol{W}_t^* \boldsymbol{h}\|^2 \right] + 2 \cdot \mathbb{E}_{\boldsymbol{h}} \left[ \|\mathcal{E}_{1:t} \boldsymbol{H}_{1:t}^\top \widetilde{\boldsymbol{U}}_{1:t} \widetilde{\boldsymbol{\Sigma}}_{1:t}^{-2} \widetilde{\boldsymbol{U}}_{1:t}^\top \boldsymbol{h}\|^2 \right]$$

$$\leq 2 \cdot \mathbb{E}_{\boldsymbol{h}} \left[ \|\boldsymbol{W}_t^* (\boldsymbol{H}_{1:t} \boldsymbol{H}_{1:t}^\top \widetilde{\boldsymbol{U}}_{1:t} \widetilde{\boldsymbol{\Sigma}}_{1:t}^{-2} \widetilde{\boldsymbol{U}}_{1:t}^\top - \boldsymbol{I}_E) \boldsymbol{h}\|^2 \right] + 2 \cdot \|\mathcal{E}_{1:t}\|^2 \cdot \mathbb{E}_{\boldsymbol{h}} \left[ \|\boldsymbol{H}_{1:t}^\top \widetilde{\boldsymbol{U}}_{1:t} \widetilde{\boldsymbol{\Sigma}}_{1:t}^{-2} \widetilde{\boldsymbol{U}}_{1:t}^\top \boldsymbol{h}\|^2 \right]$$

The term $\mathbb{E}_{\boldsymbol{h}} \|\boldsymbol{H}_{1:t}^\top \widetilde{\boldsymbol{U}}_{1:t} \widetilde{\boldsymbol{\Sigma}}_{1:t}^{-2} \widetilde{\boldsymbol{U}}_{1:t}^\top \boldsymbol{h}\|^2$ can be bounded above as follows:

$$\mathbb{E}_{\boldsymbol{h}} \|\boldsymbol{H}_{1:t}^\top \widetilde{\boldsymbol{U}}_{1:t} \widetilde{\boldsymbol{\Sigma}}_{1:t}^{-2} \widetilde{\boldsymbol{U}}_{1:t}^\top \boldsymbol{h}\|^2$$

$$= \mathrm{tr} \left( \boldsymbol{H}_{1:t}^\top \widetilde{\boldsymbol{U}}_{1:t} \widetilde{\boldsymbol{\Sigma}}_{1:t}^{-2} \widetilde{\boldsymbol{U}}_{1:t}^\top \boldsymbol{\Lambda} \widetilde{\boldsymbol{U}}_{1:t} \widetilde{\boldsymbol{\Sigma}}_{1:t}^{-2} \widetilde{\boldsymbol{U}}_{1:t}^\top \boldsymbol{H}_{1:t} \right)$$

$$= \mathrm{tr} \left( \left( \boldsymbol{\Lambda} - \frac{1}{M_t} \boldsymbol{H}_{1:t} \boldsymbol{H}_{1:t}^\top + \frac{1}{M_t} \boldsymbol{H}_{1:t} \boldsymbol{H}_{1:t}^\top \right) \widetilde{\boldsymbol{U}}_{1:t} \widetilde{\boldsymbol{\Sigma}}_{1:t}^{-2} \widetilde{\boldsymbol{U}}_{1:t}^\top \boldsymbol{H}_{1:t} \boldsymbol{H}_{1:t}^\top \widetilde{\boldsymbol{U}}_{1:t} \widetilde{\boldsymbol{\Sigma}}_{1:t}^{-2} \widetilde{\boldsymbol{U}}_{1:t}^\top \right)$$

$$\overset{(i)}{\leq} \left\| \boldsymbol{\Lambda} - \frac{1}{M_t} \boldsymbol{H}_{1:t} \boldsymbol{H}_{1:t}^\top \right\| \cdot \mathrm{tr} \left( \widetilde{\boldsymbol{U}}_{1:t} \widetilde{\boldsymbol{\Sigma}}_{1:t}^{-2} \widetilde{\boldsymbol{U}}_{1:t}^\top \boldsymbol{H}_{1:t} \boldsymbol{H}_{1:t}^\top \widetilde{\boldsymbol{U}}_{1:t} \widetilde{\boldsymbol{\Sigma}}_{1:t}^{-2} \widetilde{\boldsymbol{U}}_{1:t}^\top \right)$$

$$\quad + \frac{1}{M_t} \mathrm{tr} \left( \boldsymbol{H}_{1:t} \boldsymbol{H}_{1:t}^\top \widetilde{\boldsymbol{U}}_{1:t} \widetilde{\boldsymbol{\Sigma}}_{1:t}^{-2} \widetilde{\boldsymbol{U}}_{1:t}^\top \boldsymbol{H}_{1:t} \boldsymbol{H}_{1:t}^\top \widetilde{\boldsymbol{U}}_{1:t} \widetilde{\boldsymbol{\Sigma}}_{1:t}^{-2} \widetilde{\boldsymbol{U}}_{1:t}^\top \right)$$

$$\overset{(ii)}{\leq} \left\| \boldsymbol{\Lambda} - \frac{1}{M_t} \boldsymbol{H}_{1:t} \boldsymbol{H}_{1:t}^\top \right\| \cdot \frac{\left( \frac{1}{\gamma_t} \min\{M_{t-1} - k_{t-1}, (t-1)k_t\} + k_t \right)}{\mu_{k_t}(\boldsymbol{B}_t \boldsymbol{B}_t^\top)}$$

$$\quad + \frac{1}{M_t \gamma_t} \left( \frac{t-1}{\gamma_t^2 M_t} + \frac{2}{\gamma_t M_t} \right) \cdot \min\{M_{t-1} - k_{t-1}, (t-1)k_t\} + \frac{k_t}{M_t}$$

$$=: \mathbb{V}_t$$

Here, (i) follows from Lemma 2 and (ii) follows from Lemma 5.

The term $\mathbb{E}_{\boldsymbol{h}}\left[\|\boldsymbol{W}_t^*(\boldsymbol{H}_{1:t}\boldsymbol{H}_{1:t}^\top\widetilde{\boldsymbol{U}}_{1:t}\widetilde{\boldsymbol{\Sigma}}_{1:t}^{-2}\widetilde{\boldsymbol{U}}_{1:t}^\top - \boldsymbol{I}_E)\boldsymbol{h}\|^2\right]$ satisfies:

$$\mathbb{E}_{\boldsymbol{h}}\left[\|\boldsymbol{W}_t^*(\boldsymbol{H}_{1:t}\boldsymbol{H}_{1:t}^\top\widetilde{\boldsymbol{U}}_{1:t}\widetilde{\boldsymbol{\Sigma}}_{1:t}^{-2}\widetilde{\boldsymbol{U}}_{1:t}^\top - \boldsymbol{I}_E)\boldsymbol{h}\|^2\right]$$

$$= \mathrm{tr}\left(\boldsymbol{W}_t^*(\boldsymbol{H}_{1:t}\boldsymbol{H}_{1:t}^\top\widetilde{\boldsymbol{U}}_{1:t}\widetilde{\boldsymbol{\Sigma}}_{1:t}^{-2}\widetilde{\boldsymbol{U}}_{1:t}^\top - \boldsymbol{I}_E)\boldsymbol{\Lambda}(\widetilde{\boldsymbol{U}}_{1:t}\widetilde{\boldsymbol{\Sigma}}_{1:t}^{-2}\widetilde{\boldsymbol{U}}_{1:t}^\top\boldsymbol{H}_{1:t}\boldsymbol{H}_{1:t}^\top - \boldsymbol{I}_E)(\boldsymbol{W}_t^*)^\top\right)$$

$$\overset{(i)}{=} \mathrm{tr}\left(\boldsymbol{W}_t^*(\boldsymbol{D}_t\widetilde{\boldsymbol{U}}_{1:t}\widetilde{\boldsymbol{\Sigma}}_{1:t}^{-2}\widetilde{\boldsymbol{U}}_{1:t}^\top + \widetilde{\boldsymbol{U}}_{1:t}\widetilde{\boldsymbol{U}}_{1:t}^\top - \boldsymbol{I}_E)\boldsymbol{\Lambda}(\widetilde{\boldsymbol{U}}_{1:t}\widetilde{\boldsymbol{\Sigma}}_{1:t}^{-2}\widetilde{\boldsymbol{U}}_{1:t}^\top\boldsymbol{D}_t + \widetilde{\boldsymbol{U}}_{1:t}\widetilde{\boldsymbol{U}}_{1:t}^\top - \boldsymbol{I}_E)(\boldsymbol{W}_t^*)^\top\right)$$

$$\overset{(ii)}{\le} 2\,\mathrm{tr}\left(\boldsymbol{W}_t^*(\widetilde{\boldsymbol{U}}_{1:t}\widetilde{\boldsymbol{U}}_{1:t}^\top - \boldsymbol{I}_E)\boldsymbol{\Lambda}(\widetilde{\boldsymbol{U}}_{1:t}\widetilde{\boldsymbol{U}}_{1:t}^\top - \boldsymbol{I}_E)(\boldsymbol{W}_t^*)^\top\right)$$

$$\quad + 2\,\mathrm{tr}\left(\boldsymbol{W}_t^*\boldsymbol{D}_t\widetilde{\boldsymbol{U}}_{1:t}\widetilde{\boldsymbol{\Sigma}}_{1:t}^{-2}\widetilde{\boldsymbol{U}}_{1:t}^\top\boldsymbol{\Lambda}\widetilde{\boldsymbol{U}}_{1:t}\widetilde{\boldsymbol{\Sigma}}_{1:t}^{-2}\widetilde{\boldsymbol{U}}_{1:t}^\top\boldsymbol{D}_t(\boldsymbol{W}_t^*)^\top\right)$$

$$\overset{(iii)}{\le} 2\cdot\|\boldsymbol{W}_t^*(\boldsymbol{I}_E - \widetilde{\boldsymbol{U}}_{1:t}\widetilde{\boldsymbol{U}}_{1:t}^\top)\|_{\mathrm{F}}^2 \cdot \|(\boldsymbol{I}_E - \widetilde{\boldsymbol{U}}_{1:t}\widetilde{\boldsymbol{U}}_{1:t}^\top)\boldsymbol{\Lambda}(\boldsymbol{I}_E - \widetilde{\boldsymbol{U}}_{1:t}\widetilde{\boldsymbol{U}}_{1:t}^\top)\|$$

$$\quad + 2\cdot\|\boldsymbol{W}_t^*\|_{\mathrm{F}}^2 \cdot \|\boldsymbol{D}_t\widetilde{\boldsymbol{U}}_{1:t}\widetilde{\boldsymbol{\Sigma}}_{1:t}^{-2}\widetilde{\boldsymbol{U}}_{1:t}^\top\boldsymbol{\Lambda}\widetilde{\boldsymbol{U}}_{1:t}\widetilde{\boldsymbol{\Sigma}}_{1:t}^{-2}\widetilde{\boldsymbol{U}}_{1:t}^\top\boldsymbol{D}_t\|$$

where the above three steps, (i), (ii), and (iii), follow from Lemma 1, Lemma 3, and Lemma 2 respectively. To bound $\mathbb{B}_{t1} := \left\|(\boldsymbol{I}_E - \widetilde{\boldsymbol{U}}_{1:t}\widetilde{\boldsymbol{U}}_{1:t}^\top)\boldsymbol{\Lambda}(\boldsymbol{I}_E - \widetilde{\boldsymbol{U}}_{1:t}\widetilde{\boldsymbol{U}}_{1:t}^\top)\right\|$, we have

$$\mathbb{B}_{t1} = \left\|(\boldsymbol{I}_E - \widetilde{\boldsymbol{U}}_{1:t}\widetilde{\boldsymbol{U}}_{1:t}^\top)\Big(\boldsymbol{\Lambda} - \frac{1}{M_t}\widetilde{\boldsymbol{U}}_{1:t}\widetilde{\boldsymbol{\Sigma}}_{1:t}^2\widetilde{\boldsymbol{U}}_{1:t}^\top\Big)(\boldsymbol{I}_E - \widetilde{\boldsymbol{U}}_{1:t}\widetilde{\boldsymbol{U}}_{1:t}^\top)\right\|$$

$$\le \left\|\boldsymbol{\Lambda} - \frac{1}{M_t}\widetilde{\boldsymbol{U}}_{1:t}\widetilde{\boldsymbol{\Sigma}}_{1:t}^2\widetilde{\boldsymbol{U}}_{1:t}^\top\right\|$$

$$\le \left\|\boldsymbol{\Lambda} - \frac{1}{M_t}\boldsymbol{H}_{1:t}\boldsymbol{H}_{1:t}^\top\right\| + \frac{1}{M_t}\cdot\left\|\boldsymbol{H}_{1:t}\boldsymbol{H}_{1:t}^\top - \widetilde{\boldsymbol{U}}_{1:t}\widetilde{\boldsymbol{\Sigma}}_{1:t}^2\widetilde{\boldsymbol{U}}_{1:t}^\top\right\|$$

$$= \left\|\boldsymbol{\Lambda} - \frac{1}{M_t}\boldsymbol{H}_{1:t}\boldsymbol{H}_{1:t}^\top\right\| + \frac{1}{M_t}\cdot\|\boldsymbol{D}_t\|$$

$$= \left\|\boldsymbol{\Lambda} - \frac{1}{M_t}\boldsymbol{H}_{1:t}\boldsymbol{H}_{1:t}^\top\right\| + \frac{a_t}{M_t}.$$

To bound $\mathbb{B}_{t2} := \left\|\boldsymbol{D}_t\widetilde{\boldsymbol{U}}_{1:t}\widetilde{\boldsymbol{\Sigma}}_{1:t}^{-2}\widetilde{\boldsymbol{U}}_{1:t}^\top\boldsymbol{\Lambda}\widetilde{\boldsymbol{U}}_{1:t}\widetilde{\boldsymbol{\Sigma}}_{1:t}^{-2}\widetilde{\boldsymbol{U}}_{1:t}^\top\boldsymbol{D}_t\right\|$, we have

$$\mathbb{B}_{t2} = \left\|\boldsymbol{D}_t\widetilde{\boldsymbol{U}}_{1:t}\widetilde{\boldsymbol{\Sigma}}_{1:t}^{-2}\widetilde{\boldsymbol{U}}_{1:t}^\top\boldsymbol{\Lambda}\widetilde{\boldsymbol{U}}_{1:t}\widetilde{\boldsymbol{\Sigma}}_{1:t}^{-2}\widetilde{\boldsymbol{U}}_{1:t}^\top\boldsymbol{D}_t\right\|$$

$$= \left\|\boldsymbol{D}_t\widetilde{\boldsymbol{U}}_{1:t}\widetilde{\boldsymbol{\Sigma}}_{1:t}^{-2}\widetilde{\boldsymbol{U}}_{1:t}^\top\Big(\boldsymbol{\Lambda} - \frac{1}{M_t}\boldsymbol{H}_{1:t}\boldsymbol{H}_{1:t}^\top + \frac{1}{M_t}\boldsymbol{H}_{1:t}\boldsymbol{H}_{1:t}^\top\Big)\widetilde{\boldsymbol{U}}_{1:t}\widetilde{\boldsymbol{\Sigma}}_{1:t}^{-2}\widetilde{\boldsymbol{U}}_{1:t}^\top\boldsymbol{D}_t\right\|$$

$$\le \left\|\boldsymbol{\Lambda} - \frac{1}{M_t}\boldsymbol{H}_{1:t}\boldsymbol{H}_{1:t}^\top\right\| \cdot \left\|\widetilde{\boldsymbol{U}}_{1:t}\widetilde{\boldsymbol{\Sigma}}_{1:t}^{-2}\widetilde{\boldsymbol{U}}_{1:t}^\top\boldsymbol{D}_t\right\|^2$$

$$\quad + \frac{1}{M_t}\left\|\boldsymbol{D}_t\widetilde{\boldsymbol{U}}_{1:t}\widetilde{\boldsymbol{\Sigma}}_{1:t}^{-2}\widetilde{\boldsymbol{U}}_{1:t}^\top\boldsymbol{H}_{1:t}\boldsymbol{H}_{1:t}^\top\widetilde{\boldsymbol{U}}_{1:t}\widetilde{\boldsymbol{\Sigma}}_{1:t}^{-2}\widetilde{\boldsymbol{U}}_{1:t}^\top\boldsymbol{D}_t\right\|$$

$$\le \left\|\boldsymbol{\Lambda} - \frac{1}{M_t}\boldsymbol{H}_{1:t}\boldsymbol{H}_{1:t}^\top\right\| \cdot \frac{(t-1)^2}{\gamma_t^2} + a_{t-1}\cdot\left(\frac{(t-1)^2}{\gamma_t^2} + \frac{t-1}{\gamma_t}\right),$$

where the last step follows from Lemma 5. Putting together, we have obtained

$$\frac{\mathbb{E}_{\boldsymbol{h}}\left[\|\boldsymbol{W}_t^*(\boldsymbol{H}_{1:t}\boldsymbol{H}_{1:t}^\top\widetilde{\boldsymbol{U}}_{1:t}\widetilde{\boldsymbol{\Sigma}}_{1:t}^{-2}\widetilde{\boldsymbol{U}}_{1:t}^\top - \boldsymbol{I}_E)\boldsymbol{h}\|^2\right]}{2\cdot\|\boldsymbol{W}_t^*\|_{\mathrm{F}}^2}$$

$$\le \mathbb{B}_{t1} + \mathbb{B}_{t2}$$

$$\le \left\|\boldsymbol{\Lambda} - \frac{1}{M_t}\boldsymbol{H}_{1:t}\boldsymbol{H}_{1:t}^\top\right\|\cdot\left(1 + \frac{(t-1)^2}{\gamma_t^2}\right) + \frac{a_{t-1}}{M_t}\cdot\left(\frac{(t-1)^2}{\gamma_t^2} + \frac{t-1}{\gamma_t}\right) + \frac{a_t}{M_t} =: \mathbb{B}_t.$$

Combining the above finishes the proof. $\qquad\square$

# F  THEORETICAL GUARANTEES UNDER GAUSSIAN ASSUMPTIONS

In this section, we prove slightly tighter results than Theorems 1 and 2 presented in the main paper. The key idea is to make certain Gaussian assumptions on noise. Specifically, we assume both the

training noise $\mathcal{E}_{1:t}$ and test noise $\epsilon$ have i.i.d. $\mathcal{N}(0, \nu^2)$ entries. With these, we present and prove Theorems 3 to 5 below.

**Theorem 3.** *On top of the settings of Theorem 1, furthermore assume $\mathcal{E}_{1:t}$ consists of i.i.d. $\mathcal{N}(0, \nu^2)$ entries. Then the output $\widetilde{\boldsymbol{W}}_t = \boldsymbol{Y}_{1:t} \boldsymbol{H}_{1:t}^\top \widetilde{\boldsymbol{U}}_{1:t} \widetilde{\boldsymbol{\Sigma}}_{1:t}^{-2} \widetilde{\boldsymbol{U}}_{1:t}^\top$ of our method (4) satisfies*

$$\frac{1}{M_t} \mathbb{E}_{\mathcal{E}_{1:t}} \left\| \widetilde{\boldsymbol{W}}_t \boldsymbol{H}_{1:t} - \boldsymbol{Y}_{1:t} \right\|_{\mathrm{F}}^2 \leq 2 \cdot \|\boldsymbol{W}_t^*\|_{\mathrm{F}}^2 \left( \frac{a_t}{M_t} + \frac{a_{t-1}(t-1)}{\gamma_t M_t} + \frac{a_{t-1}(t-1)^2}{\gamma_t^2 M_t} \right) \tag{15}$$
$$+ c_t \nu^2 \left( \frac{(M_t - k_t)}{M_t} + \frac{(t-1) \min\{M_{t-1} - k_{t-1}, (t-1)k_t\}}{\gamma_t^2 M_t} \right).$$

*Proof of Theorem 3.* Let $\boldsymbol{I}_{M_t}$ be the $M_t \times M_t$ identity matrix. The training loss can be written as

$$\mathbb{E}_{\mathcal{E}_{1:t}} \left\| \widetilde{\boldsymbol{W}}_t \boldsymbol{H}_{1:t} - \boldsymbol{Y}_{1:t} \right\|_{\mathrm{F}}^2$$
$$= \mathbb{E}_{\mathcal{E}_{1:t}} \left\| \boldsymbol{Y}_{1:t} \boldsymbol{H}_{1:t}^\top \widetilde{\boldsymbol{U}}_{1:t} \widetilde{\boldsymbol{\Sigma}}_{1:t}^{-2} \widetilde{\boldsymbol{U}}_{1:t}^\top \boldsymbol{H}_{1:t} - \boldsymbol{Y}_{1:t} \right\|_{\mathrm{F}}^2$$
$$= \mathbb{E}_{\mathcal{E}_{1:t}} \left\| \boldsymbol{Y}_{1:t} (\boldsymbol{H}_{1:t}^\top \widetilde{\boldsymbol{U}}_{1:t} \widetilde{\boldsymbol{\Sigma}}_{1:t}^{-2} \widetilde{\boldsymbol{U}}_{1:t}^\top \boldsymbol{H}_{1:t} - \boldsymbol{I}_{M_t}) \right\|_{\mathrm{F}}^2$$
$$= \mathbb{E}_{\mathcal{E}_{1:t}} \left\| (\boldsymbol{W}_t^* \boldsymbol{H}_{1:t} + \mathcal{E}_{1:t}) (\boldsymbol{H}_{1:t}^\top \widetilde{\boldsymbol{U}}_{1:t} \widetilde{\boldsymbol{\Sigma}}_{1:t}^{-2} \widetilde{\boldsymbol{U}}_{1:t}^\top \boldsymbol{H}_{1:t} - \boldsymbol{I}_{M_t}) \right\|_{\mathrm{F}}^2$$
$$= \left\| \boldsymbol{W}_t^* (\boldsymbol{H}_{1:t} \boldsymbol{H}_{1:t}^\top \widetilde{\boldsymbol{U}}_{1:t} \widetilde{\boldsymbol{\Sigma}}_{1:t}^{-2} \widetilde{\boldsymbol{U}}_{1:t}^\top - \boldsymbol{I}_{M_t}) \boldsymbol{H}_{1:t} \right\|_{\mathrm{F}}^2 + \mathbb{E}_{\mathcal{E}_{1:t}} \left\| \mathcal{E}_{1:t} (\boldsymbol{H}_{1:t}^\top \widetilde{\boldsymbol{U}}_{1:t} \widetilde{\boldsymbol{\Sigma}}_{1:t}^{-2} \widetilde{\boldsymbol{U}}_{1:t}^\top \boldsymbol{H}_{1:t} - \boldsymbol{I}_{M_t}) \right\|_{\mathrm{F}}^2.$$

We can now bound the first term by Lemma 6 as follows:

$$\left\| \boldsymbol{W}_t^* (\boldsymbol{H}_{1:t} \boldsymbol{H}_{1:t}^\top \widetilde{\boldsymbol{U}}_{1:t} \widetilde{\boldsymbol{\Sigma}}_{1:t}^{-2} \widetilde{\boldsymbol{U}}_{1:t}^\top - \boldsymbol{I}_{M_t}) \boldsymbol{H}_{1:t} \right\|_{\mathrm{F}}^2 \leq 2 \cdot \|\boldsymbol{W}_t^*\|_{\mathrm{F}}^2 \left( a_t + \frac{a_{t-1}(t-1)}{\gamma_t} + \frac{a_{t-1}(t-1)^2}{\gamma_t^2} \right).$$

The second term can be bounded above as follows:

$$\mathbb{E}_{\mathcal{E}} \left\| \mathcal{E}_{1:t} (\boldsymbol{H}_{1:t}^\top \widetilde{\boldsymbol{U}}_{1:t} \widetilde{\boldsymbol{\Sigma}}_{1:t}^{-2} \widetilde{\boldsymbol{U}}_{1:t}^\top \boldsymbol{H}_{1:t} - \boldsymbol{I}_{M_t}) \right\|_{\mathrm{F}}^2$$
$$= \mathbb{E}_{\mathcal{E}} \operatorname{tr} \left( (\boldsymbol{H}_{1:t}^\top \widetilde{\boldsymbol{U}}_{1:t} \widetilde{\boldsymbol{\Sigma}}_{1:t}^{-2} \widetilde{\boldsymbol{U}}_{1:t}^\top \boldsymbol{H}_{1:t} - \boldsymbol{I}_{M_t}) \mathcal{E}_{1:t}^\top \mathcal{E}_{1:t} (\boldsymbol{H}_{1:t}^\top \widetilde{\boldsymbol{U}}_{1:t} \widetilde{\boldsymbol{\Sigma}}_{1:t}^{-2} \widetilde{\boldsymbol{U}}_{1:t}^\top \boldsymbol{H}_{1:t} - \boldsymbol{I}_{M_t}) \right)$$
$$= c_t \nu^2 \cdot \left\| \boldsymbol{H}_{1:t}^\top \widetilde{\boldsymbol{U}}_{1:t} \widetilde{\boldsymbol{\Sigma}}_{1:t}^{-2} \widetilde{\boldsymbol{U}}_{1:t}^\top \boldsymbol{H}_{1:t} - \boldsymbol{I}_{M_t} \right\|_{\mathrm{F}}^2$$
$$\leq c_t \nu^2 \cdot (M_t - k_t) + c_t \nu^2 \cdot \frac{t-1}{\gamma_t^2} \min\{M_{t-1} - k_{t-1}, (t-1)k_t\}.$$

The last inequality follows from Proposition 1. Combining the above finishes the proof. □

While in Theorem 3 bounds the average training MSE loss $\frac{1}{M_t} \mathbb{E}_{\mathcal{E}_{1:t}} \left\| \widetilde{\boldsymbol{W}}_t \boldsymbol{H}_{1:t} - \boldsymbol{Y}_{1:t} \right\|_{\mathrm{F}}^2$, an alternative is to give a bound on $\frac{1}{M_t} \mathbb{E}_{\mathcal{E}_{1:t}} \left\| \widetilde{\boldsymbol{W}}_t \boldsymbol{H}_{1:t} - \boldsymbol{W}_t^* \boldsymbol{H}_{1:t} \right\|_{\mathrm{F}}^2$. The latter term evaluates the difference between the prediction of $\widetilde{\boldsymbol{W}}_t$ and the ground-truth $\boldsymbol{W}_t^*$ on training data $\boldsymbol{H}_{1:t}$. The difference between the two terms is that $\boldsymbol{Y}_{1:t} = \boldsymbol{W}_t^* \boldsymbol{H}_{1:t} + \mathcal{E}_{1:t}$ is contaminated by noise. We bound $\frac{1}{M_t} \mathbb{E}_{\mathcal{E}_{1:t}} \left\| \widetilde{\boldsymbol{W}}_t \boldsymbol{H}_{1:t} - \boldsymbol{W}_t^* \boldsymbol{H}_{1:t} \right\|_{\mathrm{F}}^2$ in the next result.

**Theorem 4.** *On top of the settings of Theorem 1, furthermore assume $\mathcal{E}_{1:t}$ consists of i.i.d. $\mathcal{N}(0, \nu^2)$ entries. Then the output $\widetilde{\boldsymbol{W}}_t = \boldsymbol{Y}_{1:t} \boldsymbol{H}_{1:t}^\top \widetilde{\boldsymbol{U}}_{1:t} \widetilde{\boldsymbol{\Sigma}}_{1:t}^{-2} \widetilde{\boldsymbol{U}}_{1:t}^\top$ of our method (4) satisfies*

$$\frac{1}{M_t} \mathbb{E}_{\mathcal{E}_{1:t}} \left\| \widetilde{\boldsymbol{W}}_t \boldsymbol{H}_{1:t} - \boldsymbol{W}_t^* \boldsymbol{H}_{1:t} \right\|_{\mathrm{F}}^2 \leq 2 \cdot \|\boldsymbol{W}_t^*\|_{\mathrm{F}}^2 \left( \frac{a_t}{M_t} + \frac{a_{t-1}(t-1)}{\gamma_t M_t} + \frac{a_{t-1}(t-1)^2}{\gamma_t^2 M_t} \right)$$
$$+ c_t \nu^2 \cdot \left( \frac{k_t}{M_t} + \left( \frac{t-1}{\gamma_t^2 M_t} + \frac{2}{\gamma_t M_t} \right) \min\{M_{t-1} - k_{t-1}, (t-1)k_t\} \right).$$

*Proof of Theorem 4.* We have

$$\mathbb{E}_{\mathcal{E}_{1:t}} \left\| \widetilde{\boldsymbol{W}}_t \boldsymbol{H}_{1:t} - \boldsymbol{W}_t^* \boldsymbol{H}_{1:t} \right\|_{\mathrm{F}}^2$$
$$= \mathbb{E}_{\mathcal{E}_{1:t}} \left\| \boldsymbol{Y}_{1:t} \boldsymbol{H}_{1:t}^\top \widetilde{\boldsymbol{U}}_{1:t} \widetilde{\boldsymbol{\Sigma}}_{1:t}^{-2} \widetilde{\boldsymbol{U}}_{1:t}^\top \boldsymbol{H}_{1:t} - \boldsymbol{W}_t^* \boldsymbol{H}_{1:t} \right\|_{\mathrm{F}}^2$$
$$= \mathbb{E}_{\mathcal{E}_{1:t}} \left\| \boldsymbol{W}_t^* \boldsymbol{H}_{1:t} (\boldsymbol{H}_{1:t}^\top \widetilde{\boldsymbol{U}}_{1:t} \widetilde{\boldsymbol{\Sigma}}_{1:t}^{-2} \widetilde{\boldsymbol{U}}_{1:t}^\top \boldsymbol{H}_{1:t} - \boldsymbol{I}_{M_t}) + \mathcal{E}_{1:t} \boldsymbol{H}_{1:t}^\top \widetilde{\boldsymbol{U}}_{1:t} \widetilde{\boldsymbol{\Sigma}}_{1:t}^{-2} \widetilde{\boldsymbol{U}}_{1:t}^\top \boldsymbol{H}_{1:t} \right\|_{\mathrm{F}}^2$$
$$= \left\| \boldsymbol{W}_t^* (\boldsymbol{H}_{1:t} \boldsymbol{H}_{1:t}^\top \widetilde{\boldsymbol{U}}_{1:t} \widetilde{\boldsymbol{\Sigma}}_{1:t}^{-2} \widetilde{\boldsymbol{U}}_{1:t}^\top - \boldsymbol{I}_{M_t}) \boldsymbol{H}_{1:t} \right\|_{\mathrm{F}}^2 + \mathbb{E}_{\mathcal{E}_{1:t}} \left\| \mathcal{E}_{1:t} \boldsymbol{H}_{1:t}^\top \widetilde{\boldsymbol{U}}_{1:t} \widetilde{\boldsymbol{\Sigma}}_{1:t}^{-2} \widetilde{\boldsymbol{U}}_{1:t}^\top \boldsymbol{H}_{1:t} \right\|_{\mathrm{F}}^2.$$

The first term is identical to that of Theorem 3, and it remains to bound the second term:

$$\mathbb{E}_{\mathcal{E}_{1:t}}\big\|\mathcal{E}_{1:t}\boldsymbol{H}_{1:t}^{\top}\widetilde{\boldsymbol{U}}_{1:t}\widetilde{\boldsymbol{\Sigma}}_{1:t}^{-2}\widetilde{\boldsymbol{U}}_{1:t}^{\top}\boldsymbol{H}_{1:t}\big\|_{\mathrm{F}}^{2}$$

$$= \mathbb{E}_{\mathcal{E}}\,\mathrm{tr}\left(\boldsymbol{H}_{1:t}^{\top}\widetilde{\boldsymbol{U}}_{1:t}\widetilde{\boldsymbol{\Sigma}}_{1:t}^{-2}\widetilde{\boldsymbol{U}}_{1:t}^{\top}\boldsymbol{H}_{1:t}\mathcal{E}_{1:t}^{\top}\mathcal{E}_{1:t}\boldsymbol{H}_{1:t}^{\top}\widetilde{\boldsymbol{U}}_{1:t}\widetilde{\boldsymbol{\Sigma}}_{1:t}^{-2}\widetilde{\boldsymbol{U}}_{1:t}^{\top}\boldsymbol{H}_{1:t}\right)$$

$$= c_t\nu^2 \cdot \left\|\boldsymbol{H}_{1:t}^{\top}\widetilde{\boldsymbol{U}}_{1:t}\widetilde{\boldsymbol{\Sigma}}_{1:t}^{-2}\widetilde{\boldsymbol{U}}_{1:t}^{\top}\boldsymbol{H}_{1:t}\right\|_{\mathrm{F}}^{2}$$

$$\leq c_t\nu^2 \cdot \left(\frac{t-1}{\gamma_t^2}+\frac{2}{\gamma_t}\right)\min\left\{M_{t-1}-k_{t-1},(t-1)k_t\right\}+c_t\nu^2\cdot k_t.$$

The last inequality follows from Lemma 5. $\qquad\square$

**Theorem 5.** *On top of the settings of Theorem 2, furthermore assume both $\mathcal{E}_{1:t}$ and $\boldsymbol{\epsilon}$ consists of i.i.d. $\mathcal{N}(0,\nu^2)$ entries. Then the output $\widetilde{\boldsymbol{W}}_t = \boldsymbol{Y}_{1:t}\boldsymbol{H}_{1:t}^{\top}\widetilde{\boldsymbol{U}}_{1:t}\widetilde{\boldsymbol{\Sigma}}_{1:t}^{-2}\widetilde{\boldsymbol{U}}_{1:t}^{\top}$ of our method (4) satisfies*

$$\mathbb{E}_{\mathcal{E}_{1:t},\boldsymbol{h},\boldsymbol{\epsilon}}\big\|\widetilde{\boldsymbol{W}}_t\boldsymbol{h}-\boldsymbol{y}\big\|^2 \leq 2\cdot\|\boldsymbol{W}_t^*\|_{\mathrm{F}}^2\cdot\mathbb{B}_t+c_t\nu^2\cdot\mathbb{V}_t+c_t\nu^2. \tag{16}$$

*where $\mathbb{B}_t$ and $\mathbb{V}_t$ are defined in (10) and also shown below:*

$$\mathbb{B}_t = \left\|\boldsymbol{\Lambda}-\frac{1}{M_t}\boldsymbol{H}_{1:t}\boldsymbol{H}_{1:t}^{\top}\right\|\left(1+\frac{(t-1)^2}{\gamma_t^2}\right)+\left(\frac{a_t}{M_t}+\frac{a_{t-1}(t-1)}{\gamma_t M_t}+\frac{a_{t-1}(t-1)^2}{\gamma_t^2 M_t}\right)$$

$$\mathbb{V}_t = \left\|\boldsymbol{\Lambda}-\frac{1}{M_t}\boldsymbol{H}_{1:t}\boldsymbol{H}_{1:t}^{\top}\right\|\cdot\frac{\left(\frac{1}{\gamma_t}\min\left\{M_{t-1}-k_{t-1},(t-1)k_t\right\}+k_t\right)}{\mu_{k_t}(\boldsymbol{B}_t\boldsymbol{B}_t^{\top})}$$

$$+\frac{k_t}{M_t}+\left(\frac{t-1}{\gamma_t^2 M_t}+\frac{2}{\gamma_t M_t}\right)\cdot\min\left\{M_{t-1}-k_{t-1},(t-1)k_t\right\}.$$

*Proof of Theorem 5.* Recall the definition of $\boldsymbol{D}_t$ in (14). Note that $\boldsymbol{D}_t$ is a symmetric and positive semi-definite matrix.

Note that for any $\boldsymbol{W}\in\mathbb{R}^{c_t\times E}$ we have

$$\mathbb{E}_{\mathcal{E}_{1:t},\boldsymbol{h},\boldsymbol{\epsilon}}\left[\|\boldsymbol{W}\boldsymbol{h}-\boldsymbol{y}\|^2\right] = \mathbb{E}_{\mathcal{E}_{1:t},\boldsymbol{h},\boldsymbol{\epsilon}}\left[\|\boldsymbol{W}\boldsymbol{h}-\boldsymbol{W}_t^*\boldsymbol{h}-\boldsymbol{\epsilon}\|^2\right]$$

$$= \mathbb{E}_{\mathcal{E}_{1:t},\boldsymbol{h}}\left[\|\boldsymbol{W}\boldsymbol{h}-\boldsymbol{W}_t^*\boldsymbol{h}\|^2\right]+c_t\nu^2.$$

Denote by $\boldsymbol{I}_E$ the $E\times E$ identity matrix. With $\widetilde{\boldsymbol{W}}_t = \boldsymbol{Y}_{1:t}\boldsymbol{H}_{1:t}^{\top}\widetilde{\boldsymbol{U}}_{1:t}\widetilde{\boldsymbol{\Sigma}}_{1:t}^{-2}\widetilde{\boldsymbol{U}}_{1:t}^{\top}$ and $\boldsymbol{Y}_{1:t} = \boldsymbol{W}_t^*\boldsymbol{H}_{1:t}+\mathcal{E}_{1:t}$ we obtain

$$\mathbb{E}_{\mathcal{E}_{1:t},\boldsymbol{h}}\left[\|\widetilde{\boldsymbol{W}}_t\boldsymbol{h}-\boldsymbol{W}_t^*\boldsymbol{h}\|^2\right]$$

$$= \mathbb{E}_{\mathcal{E}_{1:t},\boldsymbol{h}}\left[\|\boldsymbol{Y}_{1:t}\boldsymbol{H}_{1:t}^{\top}\widetilde{\boldsymbol{U}}_{1:t}\widetilde{\boldsymbol{\Sigma}}_{1:t}^{-2}\widetilde{\boldsymbol{U}}_{1:t}^{\top}\boldsymbol{h}-\boldsymbol{W}_t^*\boldsymbol{h}\|^2\right]$$

$$= \mathbb{E}_{\mathcal{E}_{1:t},\boldsymbol{h}}\left[\|(\boldsymbol{W}_t^*\boldsymbol{H}_{1:t}+\mathcal{E}_{1:t})\boldsymbol{H}_{1:t}^{\top}\widetilde{\boldsymbol{U}}_{1:t}\widetilde{\boldsymbol{\Sigma}}_{1:t}^{-2}\widetilde{\boldsymbol{U}}_{1:t}^{\top}\boldsymbol{h}-\boldsymbol{W}_t^*\boldsymbol{h}\|^2\right]$$

$$= \mathbb{E}_{\mathcal{E}_{1:t},\boldsymbol{h}}\left[\|(\boldsymbol{W}_t^*\boldsymbol{H}_{1:t}+\mathcal{E}_{1:t})\boldsymbol{H}_{1:t}^{\top}\widetilde{\boldsymbol{U}}_{1:t}\widetilde{\boldsymbol{\Sigma}}_{1:t}^{-2}\widetilde{\boldsymbol{U}}_{1:t}^{\top}\boldsymbol{h}-\boldsymbol{W}_t^*\boldsymbol{h}\|^2\right]$$

$$= \mathbb{E}_{\boldsymbol{h}}\left[\|\boldsymbol{W}_t^*\boldsymbol{H}_{1:t}\boldsymbol{H}_{1:t}^{\top}\widetilde{\boldsymbol{U}}_{1:t}\widetilde{\boldsymbol{\Sigma}}_{1:t}^{-2}\widetilde{\boldsymbol{U}}_{1:t}^{\top}\boldsymbol{h}-\boldsymbol{W}_t^*\boldsymbol{h}\|^2\right]+\mathbb{E}_{\mathcal{E}_{1:t},\boldsymbol{h}}\left[\|\mathcal{E}_{1:t}\boldsymbol{H}_{1:t}^{\top}\widetilde{\boldsymbol{U}}_{1:t}\widetilde{\boldsymbol{\Sigma}}_{1:t}^{-2}\widetilde{\boldsymbol{U}}_{1:t}^{\top}\boldsymbol{h}\|^2\right]$$

$$= \mathbb{E}_{\boldsymbol{h}}\left[\|\boldsymbol{W}_t^*(\boldsymbol{H}_{1:t}\boldsymbol{H}_{1:t}^{\top}\widetilde{\boldsymbol{U}}_{1:t}\widetilde{\boldsymbol{\Sigma}}_{1:t}^{-2}\widetilde{\boldsymbol{U}}_{1:t}^{\top}-\boldsymbol{I}_E)\boldsymbol{h}\|^2\right]+c_t\nu^2\mathbb{E}_{\boldsymbol{h}}\left[\|\boldsymbol{H}_{1:t}^{\top}\widetilde{\boldsymbol{U}}_{1:t}\widetilde{\boldsymbol{\Sigma}}_{1:t}^{-2}\widetilde{\boldsymbol{U}}_{1:t}^{\top}\boldsymbol{h}\|^2\right]$$

The rest of the proof is identical to that of Theorem 2. $\qquad\square$

## G   ADDITIONAL THEORETICAL RESULTS

Given the weight $\widetilde{\boldsymbol{W}}_t := \boldsymbol{Y}_{1:t}\boldsymbol{H}_{1:t}^{\top}\widetilde{\boldsymbol{U}}_{1:t}\widetilde{\boldsymbol{\Sigma}}_{1:t}^{-2}\widetilde{\boldsymbol{U}}_{1:t}^{\top}$ computed by our continual implementation in Section 3, here we aim to derive upper bounds on the training MSE losses $\frac{1}{M_t}\big\|\widetilde{\boldsymbol{W}}_t\boldsymbol{H}_{1:t}-\boldsymbol{Y}_{1:t}\big\|_{\mathrm{F}}^2$ without the linear model assumption $\boldsymbol{Y}_{1:t} = \boldsymbol{W}_t^*\boldsymbol{H}_{1:t}+\mathcal{E}_{1:t}$ as used in the main paper.

First observe that
$$\big\|\widetilde{\boldsymbol{W}}_t \boldsymbol{H}_{1:t} - \boldsymbol{Y}_{1:t}\big\|_{\mathrm{F}}^2 = \big\|\boldsymbol{Y}_{1:t}(\boldsymbol{H}_{1:t}^\top \widetilde{\boldsymbol{U}}_{1:t} \widetilde{\boldsymbol{\Sigma}}_{1:t}^{-2} \widetilde{\boldsymbol{U}}_{1:t}^\top \boldsymbol{H}_{1:t} - \boldsymbol{I}_{M_t})\big\|_{\mathrm{F}}^2,$$
where we recall $\boldsymbol{I}_{M_t}$ is the $M_t \times M_t$ identity matrix. This motivates us to give a bound on $\|(\boldsymbol{H}_{1:t}^\top \widetilde{\boldsymbol{U}}_{1:t} \widetilde{\boldsymbol{\Sigma}}_{1:t}^{-2} \widetilde{\boldsymbol{U}}_{1:t}^\top \boldsymbol{H}_{1:t} - \boldsymbol{I}_{M_t})\|_{\mathrm{F}}^2$:

**Proposition 1.** *It holds for every $t \geq 1$ that ($M_0 := 0, k_0 := 0$)*
$$\left\|\boldsymbol{H}_{1:t}^\top \widetilde{\boldsymbol{U}}_{1:t} \widetilde{\boldsymbol{\Sigma}}_{1:t}^{-2} \widetilde{\boldsymbol{U}}_{1:t}^\top \boldsymbol{H}_{1:t} - \boldsymbol{I}_{M_t}\right\|_{\mathrm{F}}^2 \leq M_t - k_t + \frac{t-1}{\gamma_t^2} \min\left\{M_{t-1} - k_{t-1}, (t-1)k_t\right\}.$$

*Remark* 6. The term $M_t - k_t$ is inevitable as we truncate $M_t - k_t$ eigenvalues. Indeed, $M_t - k_t$ is precisely equal to $\|\boldsymbol{H}_{1:t}^\top \overline{\boldsymbol{U}}_{1:t} \overline{\boldsymbol{\Sigma}}_{1:t}^{-2} \overline{\boldsymbol{U}}_{1:t}^\top \boldsymbol{H}_{1:t} - \boldsymbol{I}_{M_t}\|_{\mathrm{F}}^2$, and it is the minimum of a rank-$k_t$ approximation problem:
$$M_t - k_t = \min_{\boldsymbol{L} \in \mathbb{R}^{M_t \times k_t}} \|\boldsymbol{L}\boldsymbol{L}^\top - \boldsymbol{I}_{M_t}\|_{\mathrm{F}}^2.$$
The term $\frac{t-1}{\gamma_t^2} \min\{M_{t-1} - k_{t-1}, (t-1)k_t\}$ arises as we solve LoRanPAC continually rather than offline. With $\gamma_t = 1$, this term is upper bounded by $(t-1)(M_{t-1} - k_{t-1})$. With $\gamma_t = 10^{10}$ (as discussed in the main paper), this term is negligible for even hundreds of tasks.

*Proof of Proposition 1.* From Lemma 1 it follows that
$$\boldsymbol{H}_{1:t}\boldsymbol{H}_{1:t}^\top \widetilde{\boldsymbol{U}}_{1:t} \widetilde{\boldsymbol{\Sigma}}_{1:t}^{-2} \widetilde{\boldsymbol{U}}_{1:t}^\top = \widetilde{\boldsymbol{U}}_{1:t} \widetilde{\boldsymbol{U}}_{1:t}^\top + \boldsymbol{D}_t \widetilde{\boldsymbol{U}}_{1:t} \widetilde{\boldsymbol{\Sigma}}_{1:t}^{-2} \widetilde{\boldsymbol{U}}_{1:t}^\top, \tag{17}$$
where we recall $\boldsymbol{D}_t$ is defined as $\boldsymbol{D}_t = \sum_{i=1}^t \left(\boldsymbol{B}_i \boldsymbol{B}_i^\top - \tau_{k_i}(\boldsymbol{B}_i\boldsymbol{B}_i^\top)\right)$ in (14). Then we have
$$\left\|\boldsymbol{H}_{1:t}^\top \widetilde{\boldsymbol{U}}_{1:t} \widetilde{\boldsymbol{\Sigma}}_{1:t}^{-2} \widetilde{\boldsymbol{U}}_{1:t}^\top \boldsymbol{H}_{1:t} - \boldsymbol{I}_{M_t}\right\|_{\mathrm{F}}^2$$
$$= \mathrm{tr}\left(\boldsymbol{H}_{1:t}^\top \widetilde{\boldsymbol{U}}_{1:t} \widetilde{\boldsymbol{\Sigma}}_{1:t}^{-2} \widetilde{\boldsymbol{U}}_{1:t}^\top \boldsymbol{H}_{1:t} \boldsymbol{H}_{1:t}^\top \widetilde{\boldsymbol{U}}_{1:t} \widetilde{\boldsymbol{\Sigma}}_{1:t}^{-2} \widetilde{\boldsymbol{U}}_{1:t}^\top \boldsymbol{H}_{1:t} - 2\boldsymbol{H}_{1:t}^\top \widetilde{\boldsymbol{U}}_{1:t} \widetilde{\boldsymbol{\Sigma}}_{1:t}^{-2} \widetilde{\boldsymbol{U}}_{1:t}^\top \boldsymbol{H}_{1:t} + \boldsymbol{I}_{M_t}\right)$$
$$= \mathrm{tr}\left(\boldsymbol{H}_{1:t}\boldsymbol{H}_{1:t}^\top \widetilde{\boldsymbol{U}}_{1:t} \widetilde{\boldsymbol{\Sigma}}_{1:t}^{-2} \widetilde{\boldsymbol{U}}_{1:t}^\top \boldsymbol{H}_{1:t}\boldsymbol{H}_{1:t}^\top \widetilde{\boldsymbol{U}}_{1:t} \widetilde{\boldsymbol{\Sigma}}_{1:t}^{-2} \widetilde{\boldsymbol{U}}_{1:t}^\top - 2\boldsymbol{H}_{1:t}\boldsymbol{H}_{1:t}^\top \widetilde{\boldsymbol{U}}_{1:t} \widetilde{\boldsymbol{\Sigma}}_{1:t}^{-2} \widetilde{\boldsymbol{U}}_{1:t}^\top\right) + M_t$$
$$\overset{(17)}{=} \mathrm{tr}\left(\left(\widetilde{\boldsymbol{U}}_{1:t} \widetilde{\boldsymbol{U}}_{1:t}^\top + \boldsymbol{D}_t \widetilde{\boldsymbol{U}}_{1:t} \widetilde{\boldsymbol{\Sigma}}_{1:t}^{-2} \widetilde{\boldsymbol{U}}_{1:t}^\top\right)\left(\widetilde{\boldsymbol{U}}_{1:t} \widetilde{\boldsymbol{U}}_{1:t}^\top + \boldsymbol{D}_t \widetilde{\boldsymbol{U}}_{1:t} \widetilde{\boldsymbol{\Sigma}}_{1:t}^{-2} \widetilde{\boldsymbol{U}}_{1:t}^\top\right)\right) + M_t$$
$$\qquad - 2\left(\widetilde{\boldsymbol{U}}_{1:t} \widetilde{\boldsymbol{U}}_{1:t}^\top + \boldsymbol{D}_t \widetilde{\boldsymbol{U}}_{1:t} \widetilde{\boldsymbol{\Sigma}}_{1:t}^{-2} \widetilde{\boldsymbol{U}}_{1:t}^\top\right)$$
$$= \mathrm{tr}\left(\boldsymbol{D}_t \widetilde{\boldsymbol{U}}_{1:t} \widetilde{\boldsymbol{\Sigma}}_{1:t}^{-2} \widetilde{\boldsymbol{U}}_{1:t}^\top \boldsymbol{D}_t \widetilde{\boldsymbol{U}}_{1:t} \widetilde{\boldsymbol{\Sigma}}_{1:t}^{-2} \widetilde{\boldsymbol{U}}_{1:t}^\top - \widetilde{\boldsymbol{U}}_{1:t} \widetilde{\boldsymbol{U}}_{1:t}^\top\right) + M_t$$
$$= \mathrm{tr}\left(\boldsymbol{D}_t \widetilde{\boldsymbol{U}}_{1:t} \widetilde{\boldsymbol{\Sigma}}_{1:t}^{-2} \widetilde{\boldsymbol{U}}_{1:t}^\top \boldsymbol{D}_t \widetilde{\boldsymbol{U}}_{1:t} \widetilde{\boldsymbol{\Sigma}}_{1:t}^{-2} \widetilde{\boldsymbol{U}}_{1:t}^\top\right) + M_t - k_t$$
$$\overset{(i)}{\leq} \frac{t-1}{\gamma_t^2} \min\left\{M_{t-1} - k_{t-1}, (t-1)k_t\right\} + M_t - k_t$$
where (i) is due to Lemma 5. $\qquad\qquad\square$

A simple corollary of Proposition 1 now follows:

**Corollary 1.** *The output $\widetilde{\boldsymbol{W}}_t$ of Algorithm 4 satisfies*
$$\frac{1}{M_t}\big\|\widetilde{\boldsymbol{W}}_t \boldsymbol{H}_{1:t} - \boldsymbol{Y}_{1:t}\big\|_{\mathrm{F}}^2 \leq \frac{\|\boldsymbol{Y}_{1:t}^\top \boldsymbol{Y}_{1:t}\|}{M_t}\left(M_t - k_t + \frac{t-1}{\gamma_t^2} \min\left\{M_{t-1} - k_{t-1}, (t-1)k_t\right\}\right).$$

*Remark* 7. In classification, the columns of $\boldsymbol{Y}_{1:t}$ are one-hot vectors. Hence, up to permutation, $\boldsymbol{Y}_{1:t}^\top \boldsymbol{Y}_{1:t} \in \mathbb{R}^{M_t \times M_t}$ is a block diagonal matrix with $c_t$ block, where the $i$-th diagonal block is a $n_i \times n_i$ matrix of all ones $\mathbf{1}_{n_i}$; here $n_i$ is the number of labels in class $i$. In other words, there exists a permutation matrix $\boldsymbol{\Pi}$ such that
$$\boldsymbol{Y}_{1:t}^\top \boldsymbol{Y}_{1:t} = \boldsymbol{\Pi}\,\mathrm{diag}(\mathbf{1}_{n_1}, \mathbf{1}_{n_2}, \ldots, \mathbf{1}_{n_{c_t}})\boldsymbol{\Pi}^\top.$$
Since the maximum eigenvalue of $\mathbf{1}_{m_i}$ is $m_i$, we know
$$\|\boldsymbol{Y}_{1:t}^\top \boldsymbol{Y}_{1:t}\| = \max_{i=1,\ldots,c_t}\{n_i\}.$$
Substitute this into Corollary 1 and we obtain
$$\frac{1}{M_t}\big\|\widetilde{\boldsymbol{W}}_t \boldsymbol{H}_{1:t} - \boldsymbol{Y}_{1:t}\big\|_{\mathrm{F}}^2 \leq \frac{\max_{i=1,\ldots,c_t}\{n_i\}}{M_t}\left(M_t - k_t + \frac{t-1}{\gamma_t^2} \min\left\{M_{t-1} - k_{t-1}, (t-1)k_t\right\}\right).$$

It is also of interest to bound the distances between the SVD factors computed online and offline, namely the distances between $\widetilde{\boldsymbol{\Sigma}}_{1:t}, \overline{\boldsymbol{\Sigma}}_{1:t}$ and between $\widetilde{\boldsymbol{U}}_{1:t}, \overline{\boldsymbol{U}}_{1:t}$. We do so in the next result.

**Theorem 6.** *Let $a_t$ be defined as in (6). For $t \geq 1$ define*

$$\mathrm{gap}_t := \mu_{k_t}\left(\boldsymbol{H}_{1:t}\boldsymbol{H}_{1:t}^\top\right) - \mu_{k_t+1}\left(\boldsymbol{H}_{1:t}\boldsymbol{H}_{1:t}^\top\right). \tag{18}$$

*Then it always holds that*

$$\left\|\overline{\boldsymbol{\Sigma}}_{1:t}^2 - \widetilde{\boldsymbol{\Sigma}}_{1:t}^2\right\|_\infty \leq a_{t-1}. \tag{19}$$

*Moreover, if $a_{t-1} < \left(1 - 1/\sqrt{2}\right)\mathrm{gap}_t$, then for any $t \geq 1$ we have*

$$\min_{\boldsymbol{O} \in \mathcal{O}(k)} \left\|\overline{\boldsymbol{U}}_{1:t} - \widetilde{\boldsymbol{U}}_{1:t}\boldsymbol{O}\right\|_{\mathrm{F}} \leq \left\|\overline{\boldsymbol{U}}_{1:t}\overline{\boldsymbol{U}}_{1:t}^\top - \widetilde{\boldsymbol{U}}_{1:t}\widetilde{\boldsymbol{U}}_{1:t}^\top\right\| \leq \frac{\sqrt{2}a_{t-1}}{\mathrm{gap}_t}, \tag{20}$$

*where $\mathcal{O}(k)$ be the set of $k \times k$ orthogonal matrices, defined as*

$$\mathcal{O}(k) := \{\boldsymbol{O} \in \mathbb{R}^{k \times k} : \boldsymbol{O}^\top\boldsymbol{O} = \boldsymbol{O}\boldsymbol{O}^\top = \boldsymbol{I}_k\}.$$

*Proof of Theorem 6.* It is clear that $\overline{\boldsymbol{U}}_1 = \widetilde{\boldsymbol{U}}_1$ and $\overline{\boldsymbol{\Sigma}}_1 = \widetilde{\boldsymbol{\Sigma}}_1$. We now consider the case $t \geq 2$. Note that $\widetilde{\boldsymbol{U}}_{1:t}\widetilde{\boldsymbol{\Sigma}}_{1:t}\widetilde{\boldsymbol{U}}_{1:t}^\top$ is the eigen decomposition of $\tau_{k_t}(\widetilde{\boldsymbol{U}}_{1:t-1}\widetilde{\boldsymbol{\Sigma}}_{1:t-1}^2\widetilde{\boldsymbol{U}}_{1:t-1}^\top + \boldsymbol{H}_t\boldsymbol{H}_t^\top)$, and $\overline{\boldsymbol{U}}_{1:t}\overline{\boldsymbol{\Sigma}}_{1:t}\overline{\boldsymbol{U}}_{1:t}^\top$ is the eigen decomposition of $\tau_{k_t}(\boldsymbol{H}_{1:t}\boldsymbol{H}_{1:t}^\top)$. We can compute

$$\boldsymbol{H}_{1:t}\boldsymbol{H}_{1:t}^\top - \left(\widetilde{\boldsymbol{U}}_{1:t-1}\widetilde{\boldsymbol{\Sigma}}_{1:t-1}^2\widetilde{\boldsymbol{U}}_{1:t-1}^\top + \boldsymbol{H}_t\boldsymbol{H}_t^\top\right) = \boldsymbol{H}_{1:t-1}\boldsymbol{H}_{1:t-1}^\top - \widetilde{\boldsymbol{U}}_{1:t-1}\widetilde{\boldsymbol{\Sigma}}_{1:t-1}^2\widetilde{\boldsymbol{U}}_{1:t-1}^\top$$

$$\overset{(i)}{=} \sum_{i=1}^{t-1}\left(\boldsymbol{B}_i\boldsymbol{B}_i^\top - \tau_{k_i}\left(\boldsymbol{B}_i\boldsymbol{B}_i^\top\right)\right) =: \boldsymbol{D}_t,$$

where (i) follows from Lemma 1. We can therefore apply Weyl's inequality to obtain

$$\left\|\overline{\boldsymbol{\Sigma}}_{1:t}^2 - \widetilde{\boldsymbol{\Sigma}}_{1:t}^2\right\|_\infty \leq \|\boldsymbol{D}_t\| = a_{t-1}.$$

This proves (19). On the other hand, (20) follows from the Davis-Kahan theorem (Davis & Kahan, 1970), or more precisely, from Corollary 2.8 of Chen et al. (2021). □

## G.1 RELATION BETWEEN THE OFFLINE AND ONLINE SOLUTIONS

Recall the definitions of the offline solution $\overline{\boldsymbol{W}}_t$ in (2) and the output $\widetilde{\boldsymbol{W}}_t$ of LoRanPAC in (4):

$$\overline{\boldsymbol{W}}_t = \boldsymbol{Y}_{1:t}\boldsymbol{H}_{1:t}^\top\overline{\boldsymbol{U}}_{1:t}\overline{\boldsymbol{\Sigma}}_{1:t}^{-2}\overline{\boldsymbol{U}}_{1:t}^\top, \quad \widetilde{\boldsymbol{W}}_t = \boldsymbol{Y}_{1:t}\boldsymbol{H}_{1:t}^\top\widetilde{\boldsymbol{U}}_{1:t}\widetilde{\boldsymbol{\Sigma}}_{1:t}^{-2}\widetilde{\boldsymbol{U}}_{1:t}^\top.$$

Here we aim to bound the distance $\|\overline{\boldsymbol{W}}_t - \widetilde{\boldsymbol{W}}_t\|_{\mathrm{F}}$. We consider the model $\boldsymbol{Y}_{1:t} = \boldsymbol{W}_t^*\boldsymbol{H}_{1:t}$; this is the $\boldsymbol{Y}_{1:t} = \boldsymbol{W}_t^*\boldsymbol{H}_{1:t} + \mathcal{E}_{1:t}$ with $\mathcal{E}_{1:t}$. Here we make this assumption for simplicity, and the result here can be extended to the case with noise.

Recall the definition of $\mathrm{gap}_t$ in (18):

$$\mathrm{gap}_t := \mu_{k_t}\left(\boldsymbol{H}_{1:t}\boldsymbol{H}_{1:t}^\top\right) - \mu_{k_t+1}\left(\boldsymbol{H}_{1:t}\boldsymbol{H}_{1:t}^\top\right).$$

Based on Theorem 6, we prove the following result.

**Theorem 7.** *Let $a_t$ be defined as in (6) and $\mathrm{gap}_t$ as in (18). Assume $a_{t-1} < \left(1 - 1/\sqrt{2}\right)\mathrm{gap}_t$. Suppose $\boldsymbol{Y}_{1:t} = \boldsymbol{W}_t^*\boldsymbol{H}_{1:t}$. Then we have*

$$\left\|\overline{\boldsymbol{W}}_t - \widetilde{\boldsymbol{W}}_t\right\|_{\mathrm{F}} \leq \|\boldsymbol{W}_t^*\|_{\mathrm{F}} \cdot \left(\frac{\sqrt{2}a_{t-1}}{\mathrm{gap}_t} + \frac{t-1}{\gamma_t}\right). \tag{21}$$

*Proof.* Note that

$$\boldsymbol{H}_{1:t}\boldsymbol{H}_{1:t}^\top\overline{\boldsymbol{U}}_{1:t}\overline{\boldsymbol{\Sigma}}_{1:t}^{-2}\overline{\boldsymbol{U}}_{1:t}^\top = \overline{\boldsymbol{U}}_{1:t}\overline{\boldsymbol{U}}_{1:t}^\top$$

$$\boldsymbol{H}_{1:t}\boldsymbol{H}_{1:t}^\top\widetilde{\boldsymbol{U}}_{1:t}\widetilde{\boldsymbol{\Sigma}}_{1:t}^{-2}\widetilde{\boldsymbol{U}}_{1:t}^\top = \widetilde{\boldsymbol{U}}_{1:t}\widetilde{\boldsymbol{U}}_{1:t}^\top + \boldsymbol{D}_t\widetilde{\boldsymbol{U}}_{1:t}\widetilde{\boldsymbol{\Sigma}}_{1:t}^{-2}\widetilde{\boldsymbol{U}}_{1:t}^\top,$$

where $\boldsymbol{D}_t$ is defined in (14) and the second equality follows from Lemma 1. So we have

$$\left\|\overline{\boldsymbol{W}}_t - \widetilde{\boldsymbol{W}}_t\right\|_{\mathrm{F}} = \left\|\boldsymbol{W}_t^* \boldsymbol{H}_{1:t}\boldsymbol{H}_{1:t}^\top \overline{\boldsymbol{U}}_{1:t}\overline{\boldsymbol{\Sigma}}_{1:t}^{-2}\overline{\boldsymbol{U}}_{1:t}^\top - \boldsymbol{W}_t^* \boldsymbol{H}_{1:t}\boldsymbol{H}_{1:t}^\top \widetilde{\boldsymbol{U}}_{1:t}\widetilde{\boldsymbol{\Sigma}}_{1:t}^{-2}\widetilde{\boldsymbol{U}}_{1:t}^\top\right\|_{\mathrm{F}} \tag{22}$$

$$\leq \|\boldsymbol{W}_t^*\|_{\mathrm{F}} \cdot \left\|\boldsymbol{H}_{1:t}\boldsymbol{H}_{1:t}^\top \overline{\boldsymbol{U}}_{1:t}\overline{\boldsymbol{\Sigma}}_{1:t}^{-2}\overline{\boldsymbol{U}}_{1:t}^\top - \boldsymbol{H}_{1:t}\boldsymbol{H}_{1:t}^\top \widetilde{\boldsymbol{U}}_{1:t}\widetilde{\boldsymbol{\Sigma}}_{1:t}^{-2}\widetilde{\boldsymbol{U}}_{1:t}^\top\right\| \tag{23}$$

$$= \|\boldsymbol{W}_t^*\|_{\mathrm{F}} \cdot \left\|\overline{\boldsymbol{U}}_{1:t}\overline{\boldsymbol{U}}_{1:t}^\top - \widetilde{\boldsymbol{U}}_{1:t}\widetilde{\boldsymbol{U}}_{1:t}^\top - \boldsymbol{D}_t\widetilde{\boldsymbol{U}}_{1:t}\widetilde{\boldsymbol{\Sigma}}_{1:t}^{-2}\widetilde{\boldsymbol{U}}_{1:t}^\top\right\| \tag{24}$$

$$\leq \|\boldsymbol{W}_t^*\|_{\mathrm{F}} \cdot \left(\left\|\overline{\boldsymbol{U}}_{1:t}\overline{\boldsymbol{U}}_{1:t}^\top - \widetilde{\boldsymbol{U}}_{1:t}\widetilde{\boldsymbol{U}}_{1:t}^\top\right\| + \left\|\boldsymbol{D}_t\widetilde{\boldsymbol{U}}_{1:t}\widetilde{\boldsymbol{\Sigma}}_{1:t}^{-2}\widetilde{\boldsymbol{U}}_{1:t}^\top\right\|\right) \tag{25}$$

$$\leq \|\boldsymbol{W}_t^*\|_{\mathrm{F}} \cdot \left(\left\|\overline{\boldsymbol{U}}_{1:t}\overline{\boldsymbol{U}}_{1:t}^\top - \widetilde{\boldsymbol{U}}_{1:t}\widetilde{\boldsymbol{U}}_{1:t}^\top\right\| + \left\|\boldsymbol{D}_t\widetilde{\boldsymbol{U}}_{1:t}\widetilde{\boldsymbol{\Sigma}}_{1:t}^{-2}\widetilde{\boldsymbol{U}}_{1:t}^\top\right\|\right) \tag{26}$$

$$\leq \|\boldsymbol{W}_t^*\|_{\mathrm{F}} \cdot \left(\frac{\sqrt{2}a_{t-1}}{\mathrm{gap}_t} + \frac{t-1}{\gamma_t}\right) \tag{27}$$

where the last inequality is due to Theorem 6 and Lemma 4. The proof is complete. $\qquad\square$

## H  REVIEW OF RELATED WORKS

In Appendix H.1 we review related work on CL. Recent surveys on CL include Parisi et al. (2019); van de Ven et al. (2022); Shaheen et al. (2022); Zhou et al. (2024b); Wang et al. (2024); Shi et al. (2024). See also the GitHub repo of Liu (2024) for an extensive list of CL papers.

In Appendix H.2 we review related work on random feature models.

### H.1  MORE RELATED WORK ON CONTINUAL LEARNING

Many CL methods have been proposed without explicitizing the use of pre-trained models (Ruvolo & Eaton, 2013; Kirkpatrick et al., 2017; Rebuffi et al., 2017; Lopez-Paz & Ranzato, 2017; Zeng et al., 2019; Chaudhry et al., 2019; Yan et al., 2021; Saha et al., 2021; Douillard et al., 2022; Elenter et al., 2023). An easy way to boost their performance is to adapt them for the context of pre-trained models. There are two natural approaches to do so. One approach is to use the pre-trained model as initialization and run these CL algorithms to fine-tune the pre-trained model; see, e.g., Li et al. (2024). The other approach is to train a shallow network with the output features of the pre-trained model and either of these CL algorithms. We do not explore these directions here. In what follows, we review existing CL methods explicitly designed for leveraging pre-trained models, and we review theoretical developments for CL as well.

**Prior Work on CL with Pre-trained Models.** The availability of pre-trained models has motivated new insights into designing CL methods. CL methods such designed can be roughly divided into two categories. In one category, the pre-trained model is completely frozen, and their output features are used as inputs for a tailored CL method. A straightforward method in this category, known as *SimpleCIL* (Zhou et al., 2023) or *Nearest Mean Classifier* (NMC) (Mensink et al., 2013; Rebuffi et al., 2017; Panos et al., 2023; Janson et al., 2022), is to classify a test image based on the (cosine) distances of its feature to class means of the training features. While this method is stable, hyperparameter-free, and can handle long task sequences, to the best of our knowledge, it does not have theoretical guarantees. Of course, RanPAC and LoRanPAC also fall into this category.[1] Other methods in the category include Ahrens et al. (2024); Prabhu et al. (2024). Both methods make certain modifications on top of RanPAC:

- The method of Ahrens et al. (2024) replaces the random ReLU features with the concatenation of the output features of intermediate layers. We identify that this generalizes the idea of Pao & Takefuji (1992)[2].

---

[1]Note that RanPAC might use first-session adaptation, which modifies the output features of the pre-trained model, so one might not consider RanPAC as completely freezing the pre-trained model. However, such strategy of first-session adaptation is applied only before the first task, and is not used during continual learning of tasks. In other words, the model after first-session adaptation is completely frozen, and we might just view it as our pre-trained model.

- The method of Prabhu et al. (2024) replaces the random ReLU features with random Fourier features (which were used by Rahimi & Recht (2007) for learning kernel machines), and the ridge regression solver of RanPAC with *linear discriminant analysis* (LDA) (Hayes & Kanan, 2020). Note that LDA optimizes an objective that is in general different from the MSE training loss, for which Prabhu et al. (2024) have not provided theoretical guarantees.

Similarly to RanPAC, the methods of Ahrens et al. (2024); Prabhu et al. (2024) need $O(E^3)$ time per task to invert the (regularized) $E \times E$ covariance matrix. Prabhu et al. (2024) uses $E = 25000$ in their experiments, which might constitute the current computational limit of performing the inversion (cf. Figs. 3 and 5). Also, both methods lack theoretical guarantees. In contrast, the running time of our LoRanPAC method depends only linearly on $E$ and can handle $E \geq 10^5$ with stable performance and theoretical guarantees.

In the other category of methods, the weights of the pre-trained models remain fixed, but the output features of the pre-trained models are changed. The catch is that these methods either change the input or change the network architecture. Such change could be applied layer-wise, therefore, in order to describe the idea, it is the simplest to assume the pre-trained model $f$ is a single-layer network.

- If we keep both the input and architecture fixed, then the network would take an input $\boldsymbol{X}$ and output $f(\boldsymbol{X})$.

- A popular way to change the input is to stack some trainable parameters $\boldsymbol{Z}$ with input $\boldsymbol{X}$, where $\boldsymbol{Z}$ and $\boldsymbol{X}$ have the same number of columns. The network outputs $f([\boldsymbol{X}; \boldsymbol{Z}])$. Here, it is implicitly assumed that $f$ can take input matrices with different number of rows (i.e., different number of tokens). For instance, $f$ can be a single-layer vision transformer. In this case $\boldsymbol{Z}$ is often called (visual) *prompts* (Jia et al., 2022), and CL methods using this strategy are often called *prompt-based methods*; see, e.g., (Wang et al., 2022b;c;a; Smith et al., 2023; Wang et al., 2023; Jung et al., 2023; Tang et al., 2023; Gao et al., 2024b; Roy et al., 2024; Kim et al., 2024).

- A popular way to change the architecture is to replace the input-output map $\boldsymbol{X} \mapsto f(\boldsymbol{X})$ with $\boldsymbol{X} \mapsto f(\boldsymbol{X}) + g(\boldsymbol{X})$, where $g$ is some simple shallow network parametrized by the extra trainable parameters. For instance, $g$ could be a simple two-layer linear network of the form $g(\boldsymbol{X}) = \boldsymbol{ABX}$ or $g(\boldsymbol{X}) = \boldsymbol{A}\operatorname{relu}(\boldsymbol{BX})$, where $\boldsymbol{A}, \boldsymbol{B}$ are trainable. In these case, $\boldsymbol{A}, \boldsymbol{B}$ are called *adapters* (Houlsby et al., 2019; Hu et al., 2022; Chen et al., 2022), and CL methods using this strategy are often called *adapter-based methods* (Zhou et al., 2023; 2024a; Liang & Li, 2024; Tan et al., 2024; Gao et al., 2024a).

Clearly, prompt-based and adapter-based methods can both be viewed as *expansion-based methods* that enlarge the capacity of a network in order to learn new tasks (Rusu et al., 2016; Yoon et al., 2018; Li et al., 2019; Ramesh & Chaudhari, 2022).

Despite their popularity, both prompt-based and adapter-based methods need to solve highly non-convex training problems, for which deriving informative theoretical guarantees is a significant challenge. Their lack of theoretical guarantees makes them prone to unexpected failures. For example, prompt-based methods such as *L2P* (Wang et al., 2022b), *DualPrompt* (Wang et al., 2022c), *CodaPrompt* (Smith et al., 2023), have their performance highly sensitive to the choice of hyperparameters and therefore to the pre-trained model in use (Wang et al., 2023), dataset, and problem setting (cf. Table 1); indeed, a small perturbation in learning rates might change the accuracy drastically (Zhang et al., 2023). While they are often equipped with dataset-specific hyperparameters released by authors (cf. Appendix J), their instability still emerges when applied to a long sequence of tasks. This is because new prompts or adapters are often needed to maintain high accuracy on new tasks (Zhou et al., 2024a), but doing so eventually becomes infeasible. Indeed, to train on the 100 tasks of the CIFAR100 dataset in the CIL setting with one class given at a time (B-0, Inc-1), running the adapter-based method of Zhou et al. (2024a), called *EASE*, with default hyperparameters, would create more than 117M parameters for its growing number of adapters, while the pre-trained ViTs in use have less than 87M parameters.

In Table 5 we summarize the conceptual differences of our approach from prior works.

**Prior Work on CL Theory.** Theoretical developments on CL have been chasing the current CL practice, with a majority of the theory CL papers limiting themselves to the linear, two-layer, or kernel setting (Doan et al., 2021; Heckel, 2022; Evron et al., 2022; Peng & Risteski, 2022; Lin et al., 2023;

Table 5: Conceptual comparison to prior work.

|  | Theoretical Guarantees? | Stable? | Can Handle Long Task Sequences? |
|---|---|---|---|
| L2P, Dual Prompt, CodaPrompt | None | No | No |
| EASE | None | No | No |
| SimpleCIL | None | Yes | Yes |
| RanPAC | None | No | No |
| LoRanPAC (Ours) | Theorems 1 and 2 | Yes | Yes |

Swartworth et al., 2023; Goldfarb et al., 2024; Zhao et al., 2024; Ding et al., 2024). While these works cover various theoretical aspects (e.g., generalization bounds, sample complexity, and convergence rates), there has arguably been a huge gap between their simplified settings and deep networks that state-of-the-art CL methods use. On the other hand, we have seen that deep pre-trained models in cascade with shallow trainable networks can provide competitive performance, thus it now makes sense to revise and extend these theoretical contributions within this cascaded architecture, thereby providing meaningful guarantees for learning the shallow networks. We believe this viewpoint would greatly reduce the gap between the theory and practice in the current CL literature.

## H.2    RANDOM VECTOR FUNCTIONAL LINK NETWORK AND RANDOM FEATURE MODELS

Here we review related works on random feature models. Recall our model is a two-layer network of the form

$$\boldsymbol{X} \mapsto \boldsymbol{W} \cdot \mathrm{relu}(\boldsymbol{P}\boldsymbol{X}) \tag{28}$$

where $\boldsymbol{P}$ is randomly generated and fixed, and $\boldsymbol{W}$ consists of trainable parameters.

**Random Vector Functional Link Network.** In independent efforts, Schmidt et al. (1992) and Pao & Takefuji (1992) considered models of form (1). Schmidt et al. (1992) used the sigmoid activation function $\xi \mapsto \frac{1}{1+\exp(-\xi)}$, while Pao & Takefuji (1992) specified an arbitrary activation function as inspired by Hornik et al. (1989) and stack the features $\boldsymbol{X}$ and $\boldsymbol{H}$ together (see, e.g., Section 2.1 of Malik et al. (2023)).[2] The model Pao & Takefuji (1992) proposed has been known as *random vector functional link* (RVFL), and the model of Schmidt et al. (1992) is referred to, according to a recent review (Malik et al., 2023), as *Schmidt neural network* (SNN).

Models of form (28) are best combined with MSE losses, as training $\boldsymbol{W}$ with $\boldsymbol{P}$ fixed amounts to solving a least-squares problem, which admits a closed-form solution as shown by Schmidt et al. (1992) and even earlier by Webb & Lowe (1990).

The model Schmidt et al. (1992) and Pao & Takefuji (1992) advocated was proposed, again, by Huang et al. (2004; 2006) under the name *extreme learning machine* (ELM). While Huang et al. (2004) claimed ELMs to be a *new learning scheme* in the paper title, it was criticized by (Wang & Wan, 2008; Authors) that ELMs are ideas stolen from the last century (Schmidt et al., 1992; Pao & Takefuji, 1992), which Huang et al. (2004) were aware of yet did not cite. Despite the criticism, and perhaps because of its "fancy" name, ELMs had once been popular and attracted many follow-up variants. We shall not review these variants here.

The model we considered, therefore, follows in spirit the framework put forth by Schmidt et al. (1992); Pao & Takefuji (1992). Crucially, our approach is a modern instantiation of their framework (cf. Table 6), where we consider larger-scale problems with ill-conditioned data, online solvers with GPU implementations. But does it make sense to use the random ReLU features $\boldsymbol{H}_t := \mathrm{relu}(\boldsymbol{P}\boldsymbol{X}_t)$ rather than the pre-trained features $\boldsymbol{X}_t$ for regression? Would the transformation $\boldsymbol{P} \in \mathbb{R}^{E \times d}$ even harm the pre-trained knowledge? McDonnell et al. (2023) empirically verified that having the first layer $\boldsymbol{P}$ is beneficial to performance as long as $E \geq d$, and the accuracy tends to be higher for larger $E$ (see Table A5 of Appendix F.6 of McDonnell et al. (2023)). While RanPAC is limited to $E \approx 10^4$, our approach is inherently more scalable, allowing us to take $E = 10^5$. The pursuit in higher embedding dimension $E$ brings us into the *over-parameterized* territory, where the corresponding MSE objective has infinitely many solutions, and this is different from the classic works on RVFLs or SNNs that largely focus on the case where there are just a few hundred neurons (e.g., $E \approx 100$).

---

[2]Hence, the method of Ahrens et al. (2024) can be viewed as a modern variant of Pao & Takefuji (1992) as Ahrens et al. (2024) stacks the output features of multiple intermediate layers for regression.

Table 6: Comparison between our approach and classic methods for extreme learning machines.

|  | Problem Scale | Solver | Data | Compute Platform |
|---|---|---|---|---|
| Schmidt et al. (1992); Pao & Takefuji (1992) | small | offline | well-conditioned | CPU |
| LoRanPAC (Ours) | large | online | ill-conditioned | GPU |

**Random Feature Models.** Model (28) is also studied under the name *random feature model* (RFM) with the origin of RFMs often attributed to Rahimi & Recht (2007). The RFM is considered to be a simple proxy model for understanding deep networks, and hence it has recently been popular (Belkin et al., 2018; 2019; Bartlett et al., 2020; Hastie et al., 2022; Mei & Montanari, 2022; Tsigler & Bartlett, 2023). Some of these works make statistical assumptions on $X_t$ and address technical challenges in analyzing the nonlinear map $W \cdot \mathrm{relu}(PX_t)$.

Alternatively, one could conduct analysis conditioned on $H_t := \mathrm{relu}(PX_t)$, which would be more manageable as it reduces to linear models. For example, the work of Xu & Hsu (2019); Huang et al. (2022); Bach (2024); Green & Romanov (2024) truncates the SVD factors of the features before applying least-squares. This is similar to ours, with an important difference that they apply LoRanPAC only once, while we apply it continually. Also, their results make statistical (e.g., Gaussian) assumptions on $H_{1:t}$, which could violate our context that $H_{1:t}$ is generated via $H_t := \mathrm{relu}(PX_t)$. In contrast, our results in the main paper, namely Theorems 1 and 2, make no assumptions on $H_{1:t}$ and are therefore applicable to random feature models and to the pre-trained models bridged with a random feature model.

# I  DATASET DETAILS

For convenience and completeness, we collect some details about the datasets in Tables 7 and 8.

Table 7: Datasets used for class-incremental learning experiments. $^{\dagger}$: ObjectNet, OmniBenchmark, and VTAB contain a large number of classes, and we use a subset of these datasets delivered by Zhou et al. (2023); see their Table 4 and also Table A2 of McDonnell et al. (2023).

| Dataset Name | Origin | Training Set Size | Test Set Size | # of Classes | Link |
|---|---|---|---|---|---|
| CIFAR100 | (Krizhevsky et al., 2009) | 50,000 | 10,000 | 100 | Here |
| ImageNet-R | (Hendrycks et al., 2021a) | 24,000 | 6,000 | 200 | Here |
| ImageNet-A | (Hendrycks et al., 2021b) | 5,981 | 5,985 | 200 | Here |
| CUB-200 | (Wah et al., 2011) | 9,430 | 2,358 | 200 | Here |
| ObjectNet$^{\dagger}$ | (Barbu et al., 2019) | 26,509 | 6,628 | 200 | Here |
| OmniBenchmark$^{\dagger}$ | (Zhang et al., 2022) | 89,697 | 5,985 | 300 | Here |
| VTAB$^{\dagger}$ | (Zhai et al., 2019) | 1,796 | 8,619 | 50 | Here |
| StanfordCars | (Krause et al., 2013) | 8,144 | 8,041 | 196 | Here |

Table 8: Datasets used for domain-incremental learning. See Table A3 of McDonnell et al. (2023) for even more details.

| Dataset Name | Origin | Training Set Size | Test Set Size | # of Classes | Link |
|---|---|---|---|---|---|
| CORe50 | (Lomonaco & Maltoni, 2017) | 119,894 | 44,972 | 50 | Here |
| CDDB-Hard | (Li et al., 2023) | 16,068 | 5,353 | 2 | Here |
| DomainNet | (Peng et al., 2019) | 409,832 | 176,743 | 345 | Here |

# J  EXPERIMENTAL SETUP DETAILS

The details of how we run each of the methods are specified as follows. For L2P, DualPrompt, CodaPrompt, we use the hyperparameters available in the PILOT repo (Sun et al., 2023) for the CIFAR100 and ImageNet-R datasets. Since no official hyperparameters are released for other datasets, we simply use their respective hyperparameters of CIFAR100 for other datasets. One might notice that these methods have large accuracy drops on other datasets, suggesting that they are sensitive to hyperparameters and the good and dataset-specific hyperparameters, if they exist, are to be found for these methods to perform well. This is an inherent drawback as they involve minimizing a highly non-convex training objective. On the other hand, many other methods, including SimpleCIL, RanPAC, and ours, are almost parameter-free. Specifically, the only hyperparameter of our approach is the truncation threshold and its role is clearly explained in the main paper.

For joint linear classifiers, that is LC ($\boldsymbol{X}_{1:T}$) or LC ($\boldsymbol{H}_{1:T}$), we train for 20 epochs using the cross-entropy loss, batch size 48, weight decay 0.0005, and SGD with the cosine annealing schedule. We run LC ($\boldsymbol{X}_{1:T}$) and LC ($\boldsymbol{H}_{1:T}$) with different initial learning rates $\{0.001, 0.005, 0.01, 0.02, 0.03\}$, and take report the maximum accuracy (Table 9). Note that LC ($\boldsymbol{X}_{1:T}$) and LC ($\boldsymbol{H}_{1:T}$) are trained using the cross-entropy loss, not the MSE loss. The reason is that the features $\boldsymbol{H}_{1:t}$ are highly ill-conditioned (Fig. 1), which makes SGD converge very slowly with the MSE loss. Comparing this to (11), we conclude that the MSE loss in our setting is useful when the objective is minimized via robust numerical computation techniques (e.g., our LoRanPAC implementation in Appendix C) instead of SGD.

We also consider the idea of *first-session adaptation*. This idea introduces a few hyperparameters such as the learning rate and schedule. We run experiments with two sets of hyperparameters, given respectively by RanPAC and EASE. We attach the symbol $\dagger$ to the method name when we use the hyperparameters of RanPAC (e.g., ADAM$^{\dagger}$, LoRanPAC$^{\dagger}$, RanPAC$^{\dagger}$). We attach the symbol $*$ when we use the hyperparamters of EASE (e.g., LoRanPAC$^{*}$, EASE$^{*}$).

For RanPAC, the official hyperparameters given by McDonnell et al. (2023) vary for different datasets. We try to unify the setup by keeping using the hyperparameters most frequently used by McDonnell

Table 9: Accuracy of training joint linear classifiers given all data with different initial learning rates $\{0.001, 0.005, 0.01, 0.02, 0.03\}$ and the corresponding maximum accuracy. LC ($\boldsymbol{X}_{1:T}$) trains a linear classifier using all pre-trained features $\boldsymbol{X}_{1:T}$ and LC ($\boldsymbol{H}_{1:T}$) uses all embedded features.

| | LC ($\boldsymbol{X}_{1:T}$) | | | | | | LC ($\boldsymbol{H}_{1:T}$) | | | | | |
|---|---|---|---|---|---|---|---|---|---|---|---|---|
| | 0.001 | 0.005 | 0.01 | 0.02 | 0.03 | Max | 0.001 | 0.005 | 0.01 | 0.02 | 0.03 | Max |
| *ViTs pre-trained on ImageNet-1K* (`vit_base_patch16_224`): | | | | | | | | | | | | |
| CIFAR100 | 86.39 | 87.56 | 87.47 | 87.09 | 86.99 | 87.56 | 87.76 | 86.68 | 86.36 | 86.75 | 86.46 | 87.76 |
| ImageNet-R | 70.25 | 72.42 | 72.22 | 72.02 | 71.52 | 72.42 | 73.00 | 71.08 | 71.18 | 70.70 | 71.15 | 73.00 |
| ImageNet-A | 54.64 | 58.85 | 58.46 | 57.67 | 56.55 | 58.85 | 59.25 | 56.16 | 56.09 | 56.48 | 56.42 | 59.25 |
| CUB-200 | 82.32 | 87.62 | 88.59 | 88.63 | 88.76 | 88.76 | 88.72 | 88.13 | 88.08 | 88.17 | 87.83 | 88.72 |
| ObjectNet | 57.53 | 59.70 | 59.22 | 58.68 | 58.43 | 59.70 | 59.96 | 56.14 | 55.63 | 55.48 | 55.34 | 59.96 |
| Omnibenchmark | 76.11 | 79.00 | 79.55 | 79.43 | 79.50 | 79.55 | 80.02 | 79.62 | 79.57 | 79.62 | 79.43 | 80.02 |
| VTAB | 86.40 | 90.89 | 91.32 | 91.23 | 90.89 | 91.32 | 91.17 | 89.86 | 89.99 | 90.16 | 89.99 | 91.17 |
| StanfordCars | 39.56 | 62.54 | 69.29 | 72.86 | 74.12 | 74.12 | 72.43 | 73.65 | 73.54 | 72.81 | 72.49 | 73.65 |
| *ViTs pre-trained on ImageNet-21K* (`vit_base_patch16_224_in21k`): | | | | | | | | | | | | |
| CIFAR100 | 86.15 | 86.78 | 86.33 | 85.86 | 85.17 | 86.78 | 85.80 | 85.16 | 85.05 | 85.31 | 85.4 | 85.80 |
| ImageNet-R | 67.22 | 68.63 | 67.12 | 65.90 | 65.28 | 68.63 | 68.65 | 68.00 | 68.17 | 68.23 | 67.78 | 68.65 |
| ImageNet-A | 46.68 | 51.42 | 50.03 | 49.24 | 48.58 | 51.42 | 51.15 | 50.16 | 49.44 | 48.98 | 49.64 | 51.15 |
| CUB-200 | 85.58 | 88.89 | 89.06 | 88.72 | 88.21 | 89.06 | 89.31 | 88.17 | 88.63 | 88.46 | 88.51 | 89.31 |
| ObjectNet | 58.01 | 58.39 | 57.50 | 55.87 | 54.42 | 58.39 | 56.93 | 53.33 | 54.44 | 54.47 | 54.54 | 56.93 |
| Omnibenchmark | 78.95 | 79.67 | 79.73 | 79.45 | 79.33 | 79.73 | 79.62 | 79.26 | 79.11 | 78.91 | 79.05 | 79.62 |
| VTAB | 87.71 | 90.78 | 90.97 | 90.71 | 90.11 | 90.97 | 90.78 | 91.03 | 90.42 | 90.44 | 90.27 | 91.03 |
| StanfordCars | 44.80 | 64.22 | 68.06 | 68.92 | 68.71 | 68.92 | 69.38 | 67.64 | 67.65 | 67.79 | 67.39 | 69.38 |

et al. (2023). Specifically, we set the embedding dimension $E$ to 10000 for RanPAC; note the exception that McDonnell et al. (2023) run the CDDB experiments with $E = 5000$, even though their Table A5 showed that larger $E$ in general leads to higher accuracy on CIFAR100. The main hyperparameters of RanPAC used for first-session adaptation are as follows:

```
{"tuned_epoch":20,
 "init_lr":0.01,
 "batch_size":48,
 "weight_decay":0.0005}
```

We run ADAM[†], LoRanPAC[†], RanPAC[†] where first-session adaptation uses these hyperparameters consistently for all datasets.

The EASE approach of Zhou et al. (2024a) performs fine-tuning, not just for the first session, but for every session, in an interesting way. The hyperparameters in their released code vary for different sessions and different datasets, and we refer the reader to the official GitHub repo of EASE for details. We also run LoRanPAC* with the hyperparameters of EASE for first-session adaptation. Note that since EASE does not show experiments on StanfordCars, or does not release hyperparameters on this dataset, we run EASE with its CIFAR100 hyperparameters for StanfordCars; see also Table 10 of Appendix K.1 where we tune the initial learning rates of EASE on StanfordCars, showing that the accuracy is still low.

Finally, we note that in all tables, some approaches are marked in gray; they are not directly comparable to our approach as the methodology can be very different and it is in fact possible to combine one with another for even better performance. On the other hand, RanPAC is the most related to our method, hence we highlight the comparison with the purple background.

## K  EXTRA EXPERIMENTS, FIGURES, AND TABLES

### K.1  PERFORMANCE ON STANFORDCARS

It is observed in Table 1 that many methods exhibit significant performance drops on StanfordCars. Are these methods inherently unable to handle this dataset, or is it our taking inappropriate hyperparameters that lead to poor performance? Note that the authors of these works did not test their

Table 10: Accuracy of EASE with different initial learning rates $\{0.001, 0.005, 0.01, 0.02, 0.03\}$ on StanfordCars.

| | 0.001 | | | 0.005 | | | 0.01 | | | 0.02 | | | 0.03 | |
|---|---|---|---|---|---|---|---|---|---|---|---|---|---|---|
| Inc-5 | Inc-10 | Inc-20 | Inc-5 | Inc-10 | Inc-20 | Inc-5 | Inc-10 | Inc-20 | Inc-5 | Inc-10 | Inc-20 | Inc-5 | Inc-10 | Inc-20 |
| 33.42 | 32.27 | 19.38 | 33.48 | 32.33 | 19.46 | 33.39 | 32.65 | 20.06 | 32.58 | 32.11 | 24.96 | 32.41 | 31.77 | 29.76 |

Table 11: Final accuracy of different methods using ViTs pre-trained on ImageNet-21K (`vit_base_patch16_224_in21k`). Compare this with Table 1 of the main paper.

| (Part 1) | CIFAR100 (B-0) | | | ImageNet-R (B-0) | | | ImageNet-A (B-0) | | | CUB-200 (B-0) | | | Avg. |
|---|---|---|---|---|---|---|---|---|---|---|---|---|---|
| LC ($\boldsymbol{X}_{1:T}$) | 86.78 | | | 68.63 | | | 51.42 | | | 89.06 | | | 73.97 |
| LC ($\boldsymbol{H}_{1:T}$) | 85.80 | | | 68.65 | | | 51.15 | | | 89.31 | | | 73.73 |
| | Inc-5 | Inc-10 | Inc-20 | Inc-5 | Inc-10 | Inc-20 | Inc-5 | Inc-10 | Inc-20 | Inc-5 | Inc-10 | Inc-20 | |
| RanPAC | 87.58 | 87.65 | 87.75 | 68.18 | 70.03 | 70.13 | **37.59** | 52.01 | 52.73 | 89.48 | 89.65 | 89.44 | 73.52 |
| LoRanPAC | 88.62 | 88.63 | 88.67 | 70.85 | 70.93 | 70.72 | **54.71** | 54.71 | 55.10 | 90.33 | 90.33 | 90.46 | 76.17 |
| EASE* | 85.85 | 87.67 | 89.47 | 70.27 | 74.53 | 75.88 | 43.05 | 47.53 | 54.51 | 86.77 | 86.81 | 85.50 | 73.99 |
| RanPAC* | 90.26 | 91.39 | 91.97 | 75.47 | 76.10 | 77.33 | **47.60** | 58.00 | 62.74 | 83.21 | 89.57 | 89.69 | 77.78 |
| LoRanPAC* | 90.55 | 91.88 | 92.39 | 76.48 | 76.82 | 77.25 | **56.95** | 58.85 | 62.74 | 90.63 | 90.71 | 90.67 | 79.66 |
| (Part 2) | ObjectNet (B-0) | | | OmniBenchmark (B-0) | | | VTAB (B-10) | | | StanfordCars (B-16) | | | Avg. |
| LC ($\boldsymbol{X}_{1:T}$) | 58.39 | | | 79.73 | | | 90.97 | | | 68.92 | | | 74.50 |
| LC ($\boldsymbol{H}_{1:T}$) | 56.93 | | | 79.62 | | | 91.03 | | | 69.38 | | | 74.24 |
| | Inc-5 | Inc-10 | Inc-20 | Inc-5 | Inc-10 | Inc-20 | Inc-5 | Inc-10 | Inc-20 | Inc-5 | Inc-10 | Inc-20 | |
| RanPAC | 59.14 | 59.23 | 59.29 | 78.38 | 78.40 | 78.06 | 92.37 | 92.84 | 92.84 | **43.60** | 65.78 | 65.78 | 72.14 |
| LoRanPAC | 61.35 | 61.36 | 61.33 | 80.12 | 80.03 | 80.13 | 92.90 | 92.76 | 92.92 | **68.96** | 68.76 | 68.95 | 75.80 |
| EASE* | 54.98 | 57.50 | 60.35 | 72.88 | 73.50 | 73.87 | 88.24 | 93.46 | 93.43 | 35.43 | 34.62 | 37.32 | 64.63 |
| RanPAC* | 64.56 | 66.55 | 66.05 | 78.55 | 79.15 | 79.23 | 92.77 | 92.84 | 93.72 | **2.31** | 69.11 | 69.11 | 71.16 |
| LoRanPAC* | 66.76 | 66.67 | 66.93 | 80.55 | 80.82 | 81.55 | 93.85 | 93.79 | 93.76 | **71.46** | 71.58 | 71.71 | 78.29 |

methods on StanfordCars, nor they released the corresponding hyperparameters. To rule out the case of hyperparameter misspecification, we take the EASE* method of Zhou et al. (2024a) for example, and we run it with different initial learning rates $\{0.001, 0.005, 0.01, 0.02, 0.03\}$ for the first task (this hyperparameter is called `init_lr` in the JSON file of the code repo of Sun et al. (2023); all other hyperparameters are set to the corresponding hyperparamters Zhou et al. (2024a) gave for CIFAR100. The results are in Table 10. It shows that different initial learning rates for EASE do not improve the performance on StanfordCars too much, suggesting that StanfordCars is perhaps inherently difficult for these types of methods.

## K.2 EXPERIMENTS WITH VIT FEATURES PRE-TRAINED ON IMAGENET-21K

Note that by default we use ViTs pre-trained on ImageNet-1K (`vit_base_patch16_224`). Here in Table 11 we show experiments with ViTs pre-trained on ImageNet-21K. Comparing this with Table 1, we obtain a similar conclusion that LoRanPAC is more stable than and outperforms RanPAC.

## K.3 EXPERIMENTS THAT SHOW LORANPAC STABILIZES THE TRAINING LOSSES

In Fig. 2 of the main paper we showed that LoRanPAC has stable test accuracy by truncating the SVD factors, compared to the nearly zero test accuracy of Min-Norm. In Fig. 4, we show that the truncation also stabilizes the average training MSE losses $\frac{1}{M_t}\|\boldsymbol{W}\boldsymbol{H}_{1:t} - \boldsymbol{Y}_{1:t}\|_{\mathrm{F}}^2$, which eliminates the numerical issues and furthermore enables generalization in test scenarios.

## K.4 EXPERIMENTS ON DOMAIN-INCREMENTAL LEARNING (DIL)

Here we consider *domain-incremental learning* (DIL), where each task has images of all objects collected from different sources or domains, e.g., objects in the images of task 1 could be hand-written sketches of cars, and images of task 2 could be colored cars.

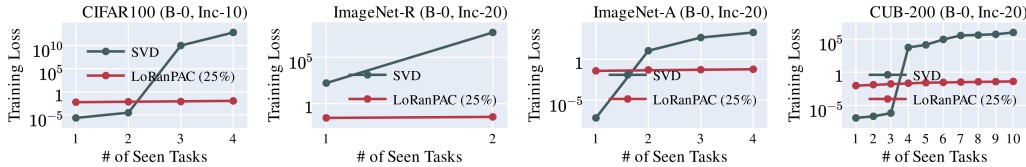

Figure 4: The average training MSE loss $\frac{1}{M_t}\|\boldsymbol{W}\boldsymbol{H}_{1:t} - \boldsymbol{Y}_{1:t}\|_F^2$ of the incremental SVD solution to Min-Norm explodes when eigenvalues of order $10^{-5}$ emerge (Fig. 1b). LoRanPAC (25%) truncates 25% minimum singular values and implements LoRanPAC online, stabilizing Min-Norm.

Table 12: Final accuracies of RanPAC and LoRanPAC with pre-trained ViTs for domain incremental learning.

|  | CORe50 | CDDB-Hard | DomainNet | Avg. |
| --- | --- | --- | --- | --- |
| RanPAC | 94.98 | 75.14 | 64.20 | 78.11 |
| LoRanPAC | 96.06 | 79.21 | 67.18 | 80.82 |

We follow the work of McDonnell et al. (2023) to run domain-incremental learning experiments on 3 datasets, CORe50 (Lomonaco & Maltoni, 2017), CDDB-Hard (Li et al., 2023), and DomainNet (Peng et al., 2019). The corresponding experimental results are shown in Table 12, from which we observe a similar phenomenon: LoRanPAC is more stable and has higher accuracy.

## K.5 SCALING LAWS OF THE EMBEDDING DIMENSION

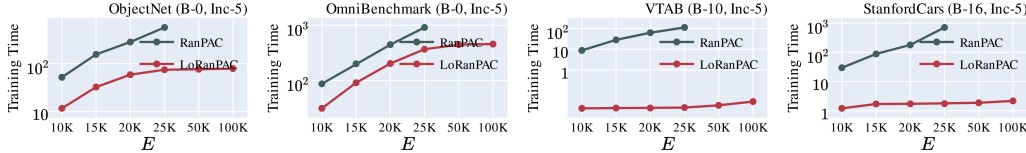

Figure 5: Training times (in minutes) of LoRanPAC and RanPAC for different embedding dimensions $E$. See also Fig. 3 in the main paper for similar results on the other four datasets.

Fig. 6 exhibits a scaling law where the accuracy of LoRanPAC grows with the embedding dimension $E$. A similar phenomenon is also shown for RanPAC in Table A5 of Appendix F.6 of McDonnell et al. (2023). However, the experiments of McDonnell et al. (2023) are limited to $E = 15000$ as RanPAC is not scalable (cf. Figs. 3 and 5), and also limited to only the CIFAR100 dataset. Here, since our LoRanPAC implementation is more stable and scalable, we can run it with $E$ as large as $10^5$, therefore visualizing the scaling phenomenon for eight different datasets in Fig. 6.

It is clearly tempting to scale the embedding dimension even more, but doing so brings up two issues. First, a large $E$ entails a large memory use, and will therefore eventually be infeasible. Second, we have not found any significant performance gain with even larger $E$ (e.g., $E = 150000$ or $E = 200000$), which is why we stopped at dimension $E = 10^5$. Note that enlarging $E$ amounts to increasing the width of the corresponding layer. It was suggested that increasing the width is beneficial, theoretically (Peng et al., 2023) and empirically (Mirzadeh et al., 2022). On the other hand, Fig. 6 suggests the benefit of increasing the width empirically diminishes, e.g., the accuracies for $E = 50K$ and $E = 100K$ are comparable on ImageNet-A, CUB-200, VTAB, and StanfordCars. This is empirically corroborated by Guha & Lakshman (2024) and theoretically confirmed by Hu et al. (2024).

## K.6 RUNNING TIMES

In Fig. 3 we compare the running times of several methods in addition to RanPAC (see Figs. 3 and 5). We see that LoRanPAC is faster than prompt-based methods (e.g., L2P, CodaPrompt) and adapter-based methods (e.g., EASE) on CIFAR100, ImageNet-R, ImageNet-A, and CUB.

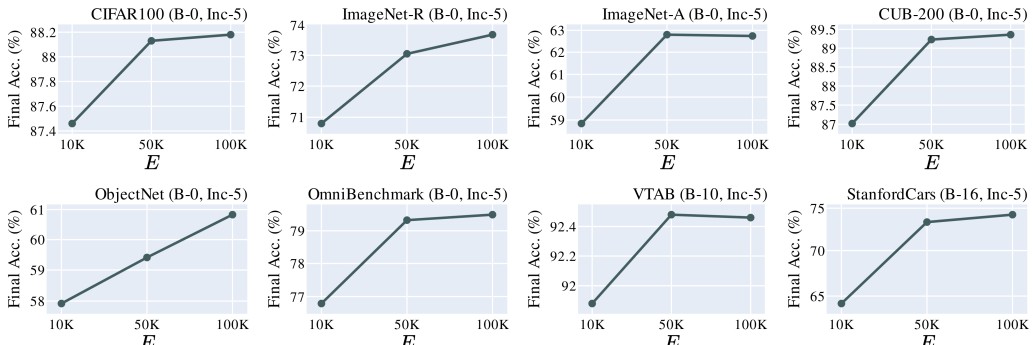

Figure 6: Final accuracy of LoRanPAC for varying embedding dimensions $E$. See also Table 15 in comparison to RanPAC.

Table 13: Running times (in minutes) of various methods on CIL datasets with $q_2 = 5$ (Inc-5). This is the same setting as Fig. 3.

| | CIFAR100 | ImageNet-R | ImageNet-A | CUB |
|---|---|---|---|---|
| L2P | 52.09 | 72.31 | 26.96 | 17.62 |
| CodaPrompt | 174.9 | 246.94 | 27.28 | 39.91 |
| EASE | 139.74 | 128.35 | 54.0 | 66.23 |
| LoRanPAC ($E = 100$K) | 31.24 | 27.0 | 1.29 | 3.73 |

### K.7 MULTIPLE RUNS OF THE EXPERIMENTS

In Table 14 we report results under the the same setting of Table 1. We report the mean and standard deviations over 3 random seeds.

Table 14: Final accuracy with pre-trained ViTs under the same setting of Table 1. Reported results are mean and standard deviations over 3 random seeds. See also Table 1.

| (Part 1) | CIFAR100 (B-0) | | | ImageNet-R (B-0) | | | ImageNet-A (B-0) | | | CUB-200 (B-0) | | |
|---|---|---|---|---|---|---|---|---|---|---|---|---|
| | Inc-5 | Inc-10 | Inc-20 | Inc-5 | Inc-10 | Inc-20 | Inc-5 | Inc-10 | Inc-20 | Inc-5 | Inc-10 | Inc-20 |
| L2P | $78.22_{\pm1.49}$ | $82.33_{\pm1.14}$ | $83.61_{\pm0.24}$ | $68.36_{\pm0.31}$ | $72.01_{\pm0.37}$ | $73.74_{\pm0.16}$ | $44.5_{\pm0.18}$ | $49.31_{\pm0.65}$ | $50.87_{\pm0.35}$ | $52.6_{\pm0.76}$ | $59.46_{\pm0.55}$ | $67.47_{\pm1.03}$ |
| DualPrompt | $80.39_{\pm0.59}$ | $83.92_{\pm0.12}$ | $85.58_{\pm0.7}$ | $67.67_{\pm0.4}$ | $71.58_{\pm0.25}$ | $72.6_{\pm0.16}$ | $49.24_{\pm1.1}$ | $53.59_{\pm0.19}$ | $56.46_{\pm0.28}$ | $53.04_{\pm1.32}$ | $61.96_{\pm2.04}$ | $68.39_{\pm1.25}$ |
| CodaPrompt | $82.53_{\pm0.51}$ | $85.85_{\pm0.43}$ | $87.62_{\pm0.2}$ | $67.74_{\pm0.33}$ | $72.31_{\pm0.37}$ | $74.67_{\pm0.33}$ | $33.42_{\pm0.56}$ | $48.56_{\pm0.76}$ | $58.29_{\pm0.93}$ | $38.48_{\pm0.29}$ | $58.61_{\pm0.87}$ | $70.67_{\pm1.38}$ |
| SimpleCIL | $80.48_{\pm0.0}$ | $80.48_{\pm0.0}$ | $80.48_{\pm0.0}$ | $63.47_{\pm0.0}$ | $63.47_{\pm0.0}$ | $63.47_{\pm0.0}$ | $58.72_{\pm0.0}$ | $58.72_{\pm0.0}$ | $58.72_{\pm0.0}$ | $80.45_{\pm0.0}$ | $80.45_{\pm0.0}$ | $80.45_{\pm0.0}$ |
| RanPAC | $86.93_{\pm0.15}$ | $87.06_{\pm0.04}$ | $87.02_{\pm0.06}$ | $71.95_{\pm0.13}$ | $71.85_{\pm0.06}$ | $72.29_{\pm0.23}$ | $\mathbf{60.11}_{\pm3.03}$ | $61.07_{\pm1.4}$ | $61.71_{\pm0.32}$ | $88.07_{\pm0.33}$ | $87.53_{\pm0.33}$ | $87.83_{\pm0.54}$ |
| LoRanPAC | $88.21_{\pm0.02}$ | $88.23_{\pm0.03}$ | $88.24_{\pm0.04}$ | $73.62_{\pm0.1}$ | $73.6_{\pm0.13}$ | $73.67_{\pm0.02}$ | $\mathbf{62.63}_{\pm0.08}$ | $62.96_{\pm0.19}$ | $62.89_{\pm0.23}$ | $89.2_{\pm0.11}$ | $89.16_{\pm0.16}$ | $89.14_{\pm0.18}$ |
| ADAM† | $83.44_{\pm0.22}$ | $84.81_{\pm0.2}$ | $86.03_{\pm0.13}$ | $63.8_{\pm0.11}$ | $64.94_{\pm0.06}$ | $70.82_{\pm0.51}$ | $58.72_{\pm0.0}$ | $58.7_{\pm0.03}$ | $58.86_{\pm0.19}$ | $80.48_{\pm0.02}$ | $80.69_{\pm0.02}$ | $80.99_{\pm0.12}$ |
| RanPAC† | $88.76_{\pm0.12}$ | $90.14_{\pm0.11}$ | $90.62_{\pm0.16}$ | $71.81_{\pm0.83}$ | $73.43_{\pm0.06}$ | $78.07_{\pm0.59}$ | $\mathbf{21.26}_{\pm29.0}$ | $61.71_{\pm0.32}$ | $62.06_{\pm0.17}$ | $87.91_{\pm0.43}$ | $87.56_{\pm0.19}$ | $88.31_{\pm0.33}$ |
| LoRanPAC† | $89.62_{\pm0.08}$ | $90.85_{\pm0.12}$ | $91.54_{\pm0.07}$ | $73.76_{\pm0.19}$ | $74.58_{\pm0.04}$ | $78.79_{\pm0.58}$ | $62.56_{\pm0.4}$ | $62.76_{\pm0.38}$ | $62.98_{\pm0.32}$ | $89.17_{\pm0.02}$ | $89.24_{\pm0.04}$ | $89.36_{\pm0.07}$ |
| EASE* | $84.68_{\pm0.4}$ | $86.87_{\pm0.25}$ | $88.46_{\pm0.24}$ | $73.44_{\pm0.26}$ | $76.2_{\pm0.69}$ | $77.54_{\pm0.18}$ | $59.67_{\pm1.05}$ | $59.56_{\pm1.7}$ | $62.41_{\pm0.24}$ | $80.56_{\pm0.09}$ | $81.07_{\pm0.45}$ | $80.92_{\pm0.56}$ |
| LoRanPAC* | $89.97_{\pm0.44}$ | $90.89_{\pm0.11}$ | $91.58_{\pm0.06}$ | $78.42_{\pm0.26}$ | $80.07_{\pm0.35}$ | $81.48_{\pm0.13}$ | $63.18_{\pm0.17}$ | $64.12_{\pm0.34}$ | $65.81_{\pm0.2}$ | $89.1_{\pm0.02}$ | $89.24_{\pm0.11}$ | $89.38_{\pm0.05}$ |

| (Part 2) | ObjectNet (B-0) | | | OmniBenchmark (B-0) | | | VTAB (B-10) | | | StanfordCars (B-16) | | |
|---|---|---|---|---|---|---|---|---|---|---|---|---|
| | Inc-5 | Inc-10 | Inc-20 | Inc-5 | Inc-10 | Inc-20 | Inc-5 | Inc-10 | Inc-20 | Inc-5 | Inc-10 | Inc-20 |
| L2P | $46.63_{\pm0.73}$ | $52.3_{\pm0.9}$ | $55.51_{\pm0.65}$ | $53.74_{\pm0.59}$ | $57.32_{\pm0.2}$ | $60.9_{\pm1.13}$ | $59.03_{\pm5.12}$ | $72.32_{\pm2.15}$ | $76.66_{\pm1.67}$ | $14.49_{\pm0.77}$ | $27.14_{\pm0.79}$ | $41.35_{\pm2.04}$ |
| DualPrompt | $47.71_{\pm0.4}$ | $52.83_{\pm0.67}$ | $56.03_{\pm0.54}$ | $56.56_{\pm0.48}$ | $59.45_{\pm0.24}$ | $62.4_{\pm0.3}$ | $64.67_{\pm4.34}$ | $75.98_{\pm3.22}$ | $79.58_{\pm3.09}$ | $11.91_{\pm0.46}$ | $18.92_{\pm0.76}$ | $29.04_{\pm3.23}$ |
| CodaPrompt | $46.7_{\pm0.37}$ | $54.32_{\pm0.79}$ | $59.26_{\pm0.39}$ | $59.33_{\pm0.67}$ | $64.87_{\pm0.65}$ | $68.37_{\pm0.45}$ | $62.39_{\pm0.58}$ | $74.93_{\pm0.6}$ | $85.24_{\pm0.52}$ | $7.89_{\pm0.17}$ | $11.63_{\pm0.35}$ | $29.46_{\pm1.43}$ |
| SimpleCIL | $51.66_{\pm0.0}$ | $51.66_{\pm0.0}$ | $51.66_{\pm0.0}$ | $70.19_{\pm0.0}$ | $70.19_{\pm0.0}$ | $70.19_{\pm0.0}$ | $82.53_{\pm0.0}$ | $82.53_{\pm0.0}$ | $82.53_{\pm0.0}$ | $35.46_{\pm0.0}$ | $35.46_{\pm0.0}$ | $35.46_{\pm0.0}$ |
| RanPAC | $58.15_{\pm0.17}$ | $58.17_{\pm0.33}$ | $58.21_{\pm0.43}$ | $77.74_{\pm0.09}$ | $77.48_{\pm0.1}$ | $77.45_{\pm0.2}$ | $91.53_{\pm0.14}$ | $91.77_{\pm0.08}$ | $91.77_{\pm0.13}$ | $\mathbf{55.53}_{\pm3.65}$ | $\mathbf{71.55}_{\pm0.11}$ | $\mathbf{71.54}_{\pm0.12}$ |
| LoRanPAC | $60.71_{\pm0.09}$ | $60.67_{\pm0.16}$ | $60.61_{\pm0.12}$ | $79.63_{\pm0.09}$ | $79.65_{\pm0.04}$ | $79.76_{\pm0.04}$ | $92.47_{\pm0.01}$ | $92.56_{\pm0.06}$ | $92.53_{\pm0.04}$ | $\mathbf{74.23}_{\pm0.11}$ | $\mathbf{74.35}_{\pm0.13}$ | $\mathbf{74.28}_{\pm0.09}$ |
| ADAM† | $52.2_{\pm0.15}$ | $53.78_{\pm0.59}$ | $55.9_{\pm0.09}$ | $70.53_{\pm0.04}$ | $70.43_{\pm0.14}$ | $70.25_{\pm0.18}$ | $82.58_{\pm0.05}$ | $82.58_{\pm0.05}$ | $82.58_{\pm0.05}$ | $35.56_{\pm0.04}$ | $35.56_{\pm0.04}$ | $35.56_{\pm0.04}$ |
| RanPAC† | $59.33_{\pm0.42}$ | $61.3_{\pm0.6}$ | $64.37_{\pm0.18}$ | $78.29_{\pm0.08}$ | $78.2_{\pm0.23}$ | $78.39_{\pm0.37}$ | $91.76_{\pm0.11}$ | $91.88_{\pm0.07}$ | $91.98_{\pm0.1}$ | $44.02_{\pm31.09}$ | $72.32_{\pm0.09}$ | $72.32_{\pm0.09}$ |
| LoRanPAC† | $61.43_{\pm0.17}$ | $63.29_{\pm0.74}$ | $66.24_{\pm0.38}$ | $79.85_{\pm0.16}$ | $80.19_{\pm0.24}$ | $80.08_{\pm0.2}$ | $92.57_{\pm0.03}$ | $92.54_{\pm0.02}$ | $92.59_{\pm0.09}$ | $74.67_{\pm0.18}$ | $74.66_{\pm0.21}$ | $74.67_{\pm0.24}$ |
| EASE* | $50.2_{\pm0.4}$ | $53.92_{\pm0.53}$ | $56.78_{\pm0.31}$ | $70.4_{\pm0.05}$ | $70.67_{\pm0.22}$ | $70.81_{\pm0.18}$ | $89.86_{\pm0.24}$ | $93.56_{\pm0.12}$ | $93.67_{\pm0.15}$ | $32.46_{\pm0.73}$ | $31.64_{\pm1.03}$ | $29.79_{\pm1.91}$ |
| LoRanPAC* | $63.67_{\pm0.6}$ | $66.52_{\pm0.79}$ | $66.96_{\pm0.27}$ | $80.07_{\pm0.37}$ | $80.34_{\pm0.17}$ | $80.41_{\pm0.37}$ | $93.16_{\pm0.67}$ | $93.17_{\pm0.67}$ | $93.16_{\pm0.7}$ | $75.63_{\pm0.22}$ | $75.56_{\pm0.14}$ | $75.67_{\pm0.24}$ |

Table 15: Final accuracies of RanPAC and LoRanPAC with pre-trained ViTs. RanPAC takes its default choice $E = 10K$, while for LoRanPAC we set three different values for $E$: $E = 10K$, $E = 50K$, and $E = 100K$.

| (Part 1) | CIFAR100 (B-0) | | | ImageNet-R (B-0) | | | ImageNet-A (B-0) | | | CUB-200 (B-0) | | | Avg. |
|---|---|---|---|---|---|---|---|---|---|---|---|---|---|
| | Inc-5 | Inc-10 | Inc-20 | Inc-5 | Inc-10 | Inc-20 | Inc-5 | Inc-10 | Inc-20 | Inc-5 | Inc-10 | Inc-20 | |
| RanPAC | 86.71 | 87.02 | 87.10 | 71.90 | 71.97 | 72.50 | 56.48 | 62.34 | 61.75 | 88.08 | 87.15 | 88.13 | 76.76 |
| LoRanPAC ($E = 10K$) | 87.48 | 87.49 | 87.42 | 70.77 | 70.85 | 70.48 | 58.85 | 58.46 | 59.38 | 86.73 | 86.85 | 87.19 | 76.00 |
| LoRanPAC ($E = 50K$) | 88.13 | 88.05 | 88.04 | 73.05 | 73.07 | 73.05 | 62.80 | 62.80 | 62.48 | 89.23 | 89.23 | 89.19 | 78.26 |
| LoRanPAC ($E = 100K$) | 88.18 | 88.18 | 88.21 | 73.67 | 73.72 | 73.63 | 62.74 | 63.20 | 63.20 | 89.36 | 89.27 | 89.23 | 78.55 |
| (Part 2) | ObjectNet (B-0) | | | OmniBenchmark (B-0) | | | VTAB (B-10) | | | StanfordCars (B-16) | | | Avg. |
| | Inc-5 | Inc-10 | Inc-20 | Inc-5 | Inc-10 | Inc-20 | Inc-5 | Inc-10 | Inc-20 | Inc-5 | Inc-10 | Inc-20 | |
| RanPAC | 58.77 | 57.66 | 57.69 | 77.63 | 77.63 | 77.46 | 91.15 | 91.58 | 91.58 | 58.03 | 71.40 | 71.40 | 73.50 |
| LoRanPAC ($E = 10K$) | 57.97 | 57.82 | 58.06 | 76.79 | 76.96 | 76.91 | 91.89 | 91.81 | 91.91 | 64.03 | 64.43 | 64.72 | 72.78 |
| LoRanPAC ($E = 50K$) | 59.41 | 59.4 | 59.54 | 79.33 | 79.35 | 79.43 | 92.48 | 92.47 | 92.54 | 73.32 | 73.66 | 73.47 | 76.20 |
| LoRanPAC ($E = 100K$) | 60.83 | 60.86 | 60.77 | 79.50 | 79.60 | 79.70 | 92.46 | 92.55 | 92.56 | 74.21 | 74.39 | 74.39 | 76.82 |

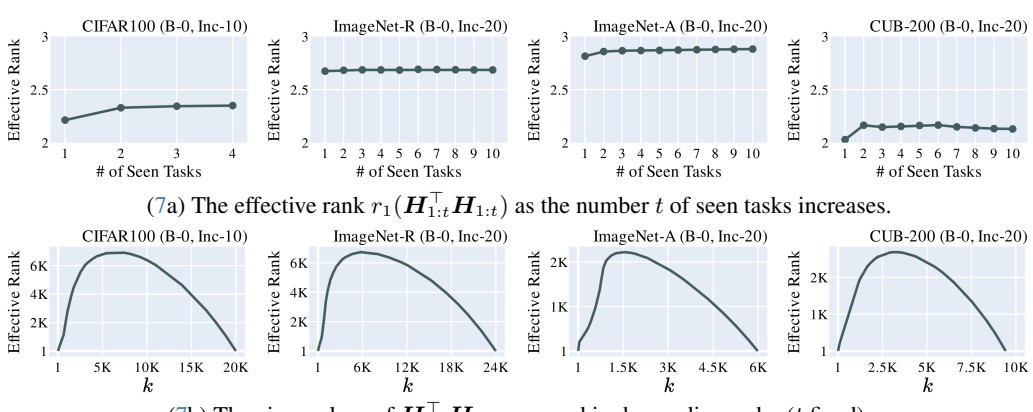

(7a) The effective rank $r_1(\boldsymbol{H}_{1:t}^\top \boldsymbol{H}_{1:t})$ as the number $t$ of seen tasks increases.

(7b) The eigenvalues of $\boldsymbol{H}_{1:t}^\top \boldsymbol{H}_{1:t}$ arranged in descending order ($t$ fixed).

Figure 7: The effective ranks of $\boldsymbol{H}_{1:t}^\top \boldsymbol{H}_{1:t}$.

## K.8 MORE FIGURES FOR DATA ANALYSIS

Recall $\mu_k(\cdot)$ denotes the $k$-th largest eigenvalue of a matrix and $M_t := m_1 + \cdots m_t$. Define the notion of *effective rank*, as by Tsigler & Bartlett (2023):

$$r_k\left(\boldsymbol{H}_{1:t}^\top \boldsymbol{H}_{1:t}\right) := \frac{\sum_{j \geq k} \mu_j\left(\boldsymbol{H}_{1:t}^\top \boldsymbol{H}_{1:t}\right)}{\mu_k\left(\boldsymbol{H}_{1:t}^\top \boldsymbol{H}_{1:t}\right)}, \quad \forall k = 1, \ldots, M_t \tag{29}$$

Note that $r_1(\cdot)$ is the standard definition of effective rank (sometimes called *stable rank*), while $r_k(\cdot)$ generalizes it by only considering the eigenvalues starting from the $k$ largest.

Fig. 7a plots the effective rank $r_1(\boldsymbol{H}_{1:t}^\top \boldsymbol{H}_{1:t})$ as the number of tasks increases and Fig. 7b plots $r_k(\boldsymbol{H}_{1:t}^\top \boldsymbol{H}_{1:t})$ for different values of $k$. We observe similar curves on different datasets: $r_1(\boldsymbol{H}_{1:t}^\top \boldsymbol{H}_{1:t})$ is smaller than 3, and $r_k(\boldsymbol{H}_{1:t}^\top \boldsymbol{H}_{1:t})$ first increases and then decreases as a function of $k$.

Fig. 8 plots the top $k_t$ eigenvalues of $\boldsymbol{B}_t \boldsymbol{B}_t^\top$ (recall that we only preserve top $k_t$ singular values of $\boldsymbol{B}_t$). It shows that the condition number is now of order $10^5$ ($10^{11}/10^6$), which is much smaller than the condition number of $\boldsymbol{H}_{1:t}^\top \boldsymbol{H}_{1:t}$.

Fig. 9 plots the normalized differences $\|\overline{\boldsymbol{\Sigma}}_{1:t} - \widetilde{\boldsymbol{\Sigma}}_{1:t}\|_\infty / \|\overline{\boldsymbol{\Sigma}}_{1:t}\|_\infty$ between the eigenvalues given by LoRanPAC and its continual implementation (Algorithm 4). This empirically verifies that the differences between the two are insignificant compared to the largest eigenvalue $\|\overline{\boldsymbol{\Sigma}}_{1:t}\|_\infty$ (just of order $10^{-3}$). This experiment assists understanding Theorem 6.

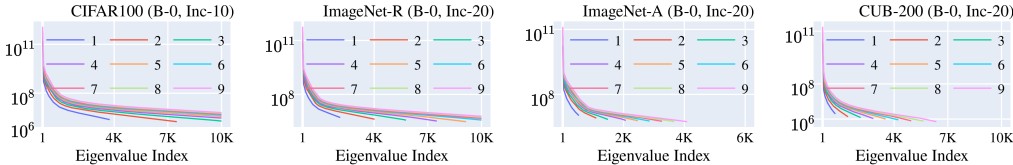

Figure 8: The eigenvalues of $\widetilde{\boldsymbol{\Sigma}}_{1:t}$ for $t = 1, \ldots, 9$. These are by definition the top $k_t$ eigenvalues of $\boldsymbol{B}_t \boldsymbol{B}_t^\top$. It shows that our continual implementation (Algorithm 4) prunes the smallest eigenvalues (of order $10^{-5}$) so that the condition number is now of order $10^5$ ($10^{11}/10^6$); compare this figure with Fig. 1a. In this experiment we truncate 25% of the eigenvalues; that is, given $M_t$, we select $k_t$ such that $k_t/M_t = 75\%$.

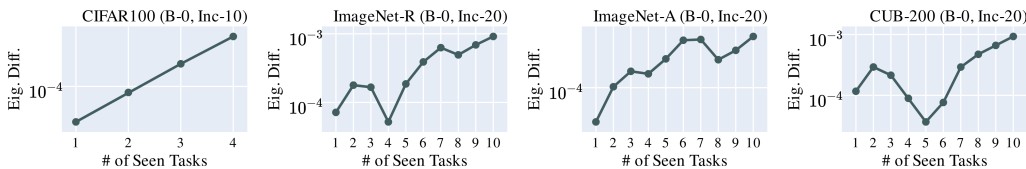

Figure 9: Normalized differences $\|\overline{\boldsymbol{\Sigma}}_{1:t} - \widetilde{\boldsymbol{\Sigma}}_{1:t}\|_\infty / \|\overline{\boldsymbol{\Sigma}}_{1:t}\|_\infty$ between the eigenvalues given by LoRanPAC and its continual implementation (Algorithm 4). This empirically verifies that the differences between the two are insignificant compared to the largest eigenvalue $\|\overline{\boldsymbol{\Sigma}}_{1:t}\|_\infty$ (just of order $10^{-3}$). See Fig. 8 and Fig. 1a. See also Theorem 6 where we formally bound the distances $\|\overline{\boldsymbol{\Sigma}}_{1:t} - \widetilde{\boldsymbol{\Sigma}}_{1:t}\|_\infty$ for every $t$.

Fig. 10 plots the eigenvalues of $\boldsymbol{H}_{1:t}^\top \boldsymbol{H}_{1:t} \in \mathbb{R}^{M_t \times M_t}$ ($M_t \leq 10^4$) with the embedding dimension $E$ varying in $\{10000, 25000, 50000, 75000\}$. It shows that the "shape" of the spectrum is similar for different values of $E$ and on different datasets (see also Fig. 1a for the case $E = 10^5$).

Fig. 11 depicts how the random embedding $\boldsymbol{P} \in \mathbb{R}^{E \times d}$ and ReLU layer affect the spectrum of the features. In Fig. 11a we plot the output features $\boldsymbol{X} \in \mathbb{R}^{d \times M}$ of the ImageNet-A dataset from pre-trained ViTs ($d = 768$, $M = 5981$, $E = 10^5$). We see $\boldsymbol{X}\boldsymbol{X}^\top$ is relatively well-conditioned: Its maximum eigenvalue is of order $10^5$ and minimum eigenvalue of order 10. Fig. 11b shows that $\boldsymbol{X}^\top \boldsymbol{P}^\top \boldsymbol{P} \boldsymbol{X} \in \mathbb{R}^{E \times M}$ is ill conditioned. This is because $\boldsymbol{P}\boldsymbol{X}$ has rank at most $d$, and the smallest $M - d$ eigenvalues of $\boldsymbol{X}^\top \boldsymbol{P}^\top \boldsymbol{P} \boldsymbol{X}$ should be zero, while we get these small and non-zero eigenvalues in Fig. 11b due to numerical errors in (incremental) SVD; these eigenvalues should be truncated (set to zero), in order to solve Min-Norm accurately. Fig. 11c shows that $\mathrm{relu}(\boldsymbol{P}\boldsymbol{X})$ also has these small and non-zero eigenvalues. While the rank of $\mathrm{relu}(\boldsymbol{P}\boldsymbol{X})$ is unclear, its smallest yet non-zero eigenvalues are likely inherent from $\boldsymbol{P}\boldsymbol{X}$, and we suggest truncating them as well. See also Table 16.

### K.9 ABLATION STUDY ON RANDOM RELU FEATURES

In Table 16 we study the effects of the random ReLU model. Recall that, given the output features $\boldsymbol{X}_t$ of the data of task $t$ from a pre-trained model, we use the random ReLU features $\boldsymbol{H}_t := \mathrm{relu}(\boldsymbol{P}\boldsymbol{X}_t)$ and labels $\boldsymbol{Y}_t$ to train a linear classifier via continually solving Min-Norm or LoRanPAC. We could

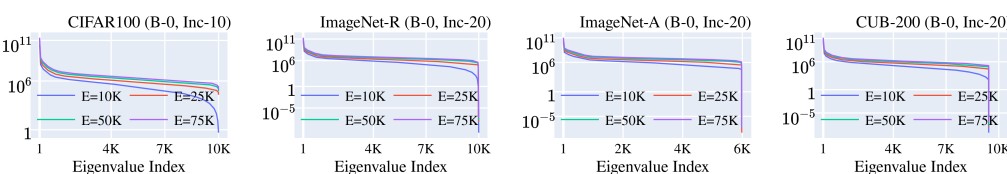

Figure 10: Eigenvalues of $\boldsymbol{H}_{1:t}^\top \boldsymbol{H}_{1:t} \in \mathbb{R}^{M_t \times M_t}$ ($M_t \leq 10^4$) with the embedding dimension $E$ varying in $\{10000, 25000, 50000, 75000\}$. We find that the "shape" of the spectrum is similar for different $E$ and on different datasets (see also Fig. 1a for the case $E = 10^5$).

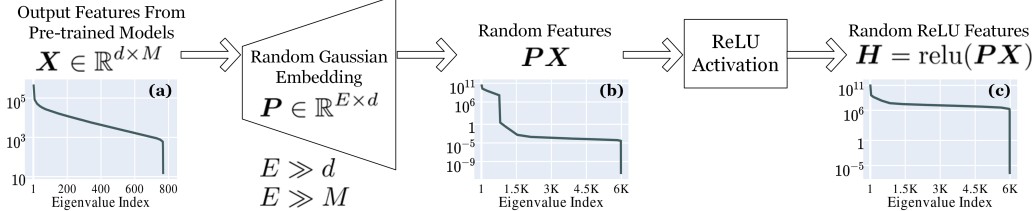

Figure 11: Feeding $M$ data samples to a pre-trained model gives its $d$-dimensional output features $X \in \mathbb{R}^{d \times M}$. Passing it through a random embedding layer $P$ and a ReLU layer yields the random ReLU features $H \in \mathbb{R}^{E \times M}$. Plotted in (a), (b), (c), respectively, are eigenvalues of $XX^\top$, $X^\top P^\top PX$, and $H^\top H$ in descending order, where $X$ consists of pre-trained ViT features of the ImageNet-A dataset ($d = 768, M = 5981, E = 10^5$). It is seen that $PX$ and , $H$ are more ill-conditioned than $X$. See also Table 16.

instead use $X_t$ or $PX_t$ or $\text{relu}(X_t)$ as the features to train the linear classifier. To see the effects of these alternative choices, we make Table 16 from which we have the following observations:

- The random ReLU features $H_t$ gives the highest accuracy, while using the ReLU layer alone or random embedding alone does not make improvements over the original pre-trained features $X_t$.

- Solving Min-Norm via Incremental SVD exhibits numerical failures as soon as we use random embedding $P$. This is because $PX \in \mathbb{R}^{E \times M}$ has rank at most $d$ and we would get some $M - d$ small yet non-zero eigenvalues accounting for the numerical errors of (incremental) SVD solvers (see, e.g., Fig. 11). While they damage the accuracy, truncating these singular values and the corresponding singular vectors restores the performance.

Table 16: Final accuracy of Min-Norm and LoRanPAC when using different features, $X_t \in \mathbb{R}^{d \times m_t}$, $\text{relu}(X_t)$, $PX_t$, and $H_t := \text{relu}(PX_t)$. Here $P \in \mathbb{R}^{E \times d}$ is a random Gaussian matrix with $\mathcal{N}(0, 1)$ entries and $E = 10^5$, and $H_t$ consists of random ReLU features we use by default. Note that both Min-Norm and LoRanPAC are solved by the incremental SVD method, with a difference that the latter truncates the SVDs. Incremental SVD without truncation is not scalable enough to handle all 50000 data samples of CIFAR100, so we mark "N.A." in the table for Min-Norm.

|  | CIFAR100 (B-0, Inc-10) | ImageNet-R (B-0, Inc-20) | ImageNet-A (B-0, Inc-20) | CUB (B-0, Inc-20) | Avg. |
|---|---|---|---|---|---|
| | | *Final Accuracy of Min-Norm* | | | |
| $X_t$ | 85.11 | 69.22 | 58.92 | 84.90 | 74.54 |
| $\text{relu}(X_t)$ | 84.07 | 66.43 | 55.23 | 84.14 | 72.47 |
| $PX_t$ | N.A. | 1.35 | 2.37 | 0.55 | N.A. |
| $H_t$ | N.A. | 0.42 | 0.92 | 0.72 | N.A. |
| | | *Final Accuracy of LoRanPAC* | | | |
| $X_t$ | 84.61 | 68.23 | 59.45 | 84.14 | 74.11 |
| $\text{relu}(X_t)$ | 83.51 | 66.20 | 56.62 | 84.01 | 72.59 |
| $PX_t$ | 78.96 | 48.50 | 42.73 | 63.15 | 58.33 |
| $H_t$ (default) | 88.18 | 73.65 | 63.20 | 89.23 | 78.57 |

### K.10 EXTRA EMPIRICAL STUDY OF RANPAC

In Appendix K.10.2, we analyze the training losses of RanPAC. In Appendix K.10.3 we show RanPAC is unstable with respect to small increments, while LoRanPAC is more stable.

### K.10.1 RANPAC IS UNSTABLE WITH RESPECT TO REGULARIZATION PARAMETER

In Fig. 12a, we show that RanPAC is unstable as it fails if the regularization $\lambda$ is small. In Fig. 12b, we extend the solution of (4) into $J_t \widetilde{U}_{1:t}(\widetilde{\Sigma}_{1:t}^2 + \lambda I_E)^{-1}\widetilde{U}_{1:t}^\top$ for ridge regression; this amounts to RanPAC with truncation. This extension stabilizes RanPAC and is practically immune to changes in the regularization parameter $\lambda$.

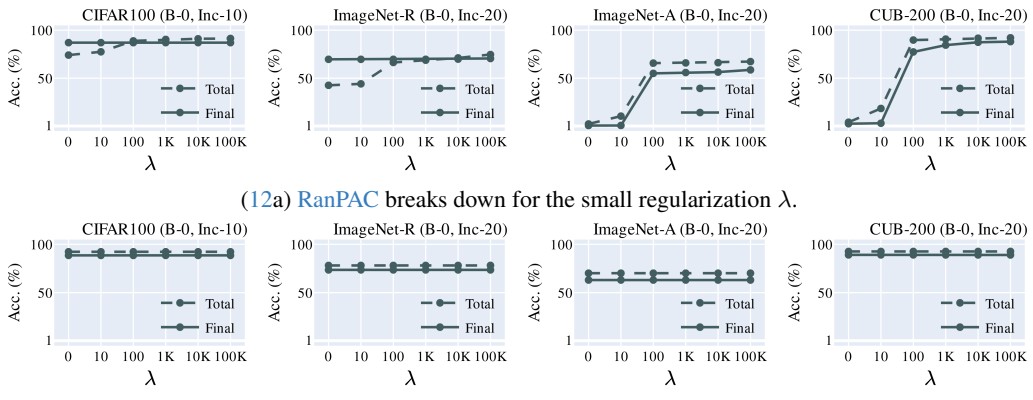

(12a) RanPAC breaks down for the small regularization $\lambda$.

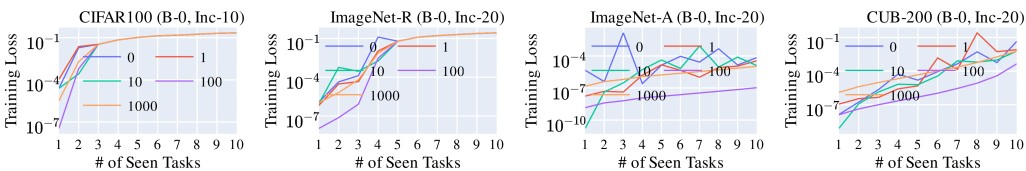

(12b) RanPAC with truncation (25%) has stable performance for varying regularization parameter $\lambda$.

Figure 12: We extend the solution of (4) into $\boldsymbol{J}_t \widetilde{\boldsymbol{U}}_{1:t} (\widetilde{\boldsymbol{\Sigma}}_{1:t}^2 + \lambda \boldsymbol{I}_E)^{-1} \widetilde{\boldsymbol{U}}_{1:t}^\top$ for ridge regression; this amounts to RanPAC with truncation. This extension stabilizes RanPAC.

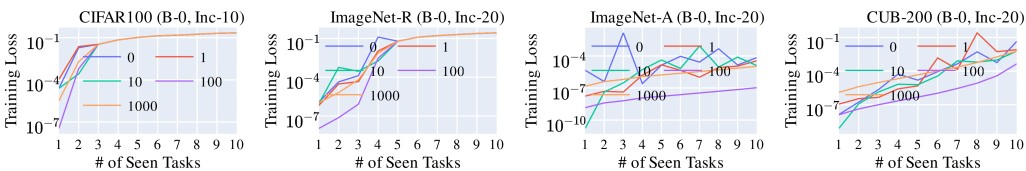

Figure 13: The average training MSE loss $\frac{1}{M_t} \| \boldsymbol{W} \boldsymbol{H}_{1:t} - \boldsymbol{Y}_{1:t} \|_{\mathrm{F}}^2$ of RanPAC for different ridge regularization parameters $\lambda \in \{0, 1, 10, 100, 1000\}$. Compare this with Fig. 12 and Fig. 4.

### K.10.2 TRAINING LOSSES OF RANPAC

Fig. 13 plots the training loss of RanPAC for different ridge regularization parameters $\lambda \in \{0, 1, 10, 100, 1000\}$. We observe that, In the case of $\lambda = 0$, the training loss of RanPAC is smaller than 1. Fig. 4 shows that the incremental SVD implementation of Min-Norm could have its training loss larger than $10^{10}$. This difference is because the incremental SVD implementation (without truncation) can be unstable and accumulates errors over time, while RanPAC is implemented by solving the normal equations $\boldsymbol{W}(\boldsymbol{H}_{1:t}\boldsymbol{H}_{1:t}^\top + \lambda \boldsymbol{I}_E) = \boldsymbol{Y}_{1:t}\boldsymbol{H}_{1:t}^\top$ in variable $\boldsymbol{W}$ (the covariances $\boldsymbol{H}_{1:t}\boldsymbol{H}_{1:t}^\top$ and $\boldsymbol{Y}_{1:t}\boldsymbol{H}_{1:t}^\top$ are updated continually). The advantage is that maintaining $\boldsymbol{H}_{1:t}\boldsymbol{H}_{1:t}^\top$ and $\boldsymbol{Y}_{1:t}\boldsymbol{H}_{1:t}^\top$ is easy and does not entail numerical errors, so solving the normal equations $\boldsymbol{W}(\boldsymbol{H}_{1:t}\boldsymbol{H}_{1:t}^\top + \lambda \boldsymbol{I}_E) = \boldsymbol{Y}_{1:t}\boldsymbol{H}_{1:t}^\top$ directly is expected to be stable, as long as the solver invoked is numerically stable (the built-in PyTorch solver is used). This appears to be the case, as RanPAC maintains small training errors. On the other hand, the corresponding test accuracy can be nearly zero with small $\lambda$ (e.g., when $\lambda = 0, 1$ as shown in Fig. 12).

### K.10.3 RANPAC IS UNSTABLE FOR CIL WITH THE SMALLEST INCREMENTS

In the experiments of the main paper, we see that RanPAC is unstable for small increments (e.g., Inc-5). Moreover, its instability is exacerbated in the extreme case Inc-1 (Table 2).

Here, we show similar experimental results in Figs. 14 to 17, suggesting that RanPAC is significantly more unstable for Inc-1, Inc-2, and Inc-4. In particular, in the extreme case of Inc-1, RanPAC presents failures on all datasets except CIFAR100 (as indicated by the verticle blue line in Figs. 14 to 17). On the contrary, these figures, including Fig. 18, show that our LoRanPAC method is stable for different small increments (1, 2, 4, 5).

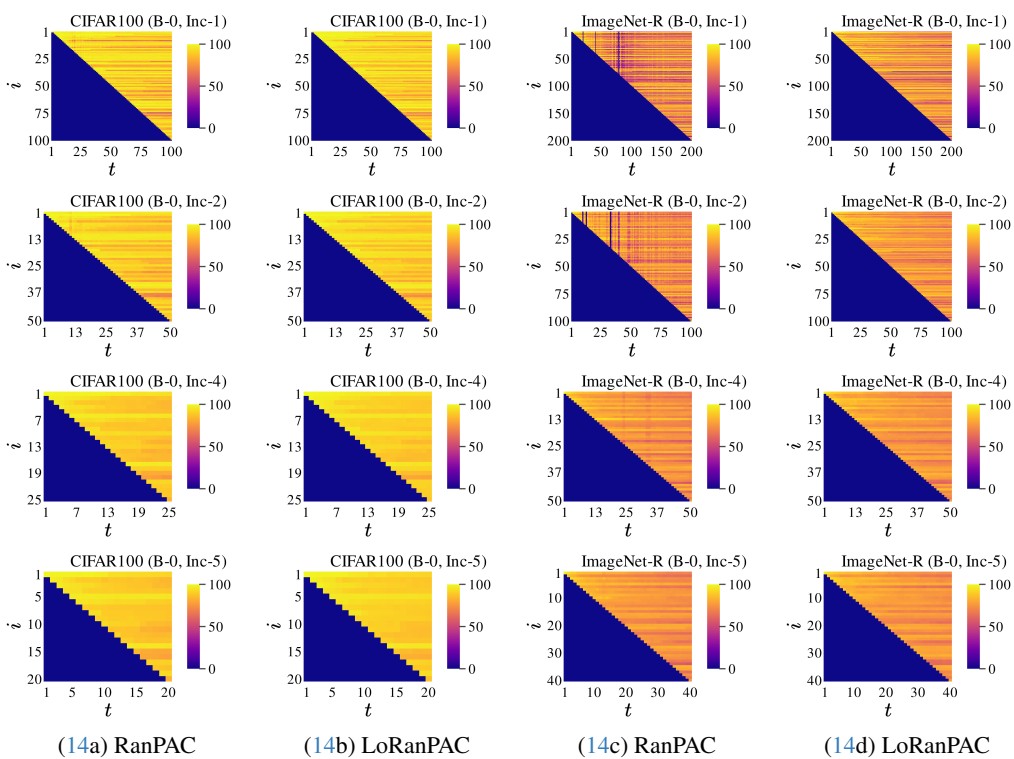

Figure 14: Upper triangular accuracy matrices on CIFAR100 and ImageNet-R (Inc-1, 2, 4, 5).

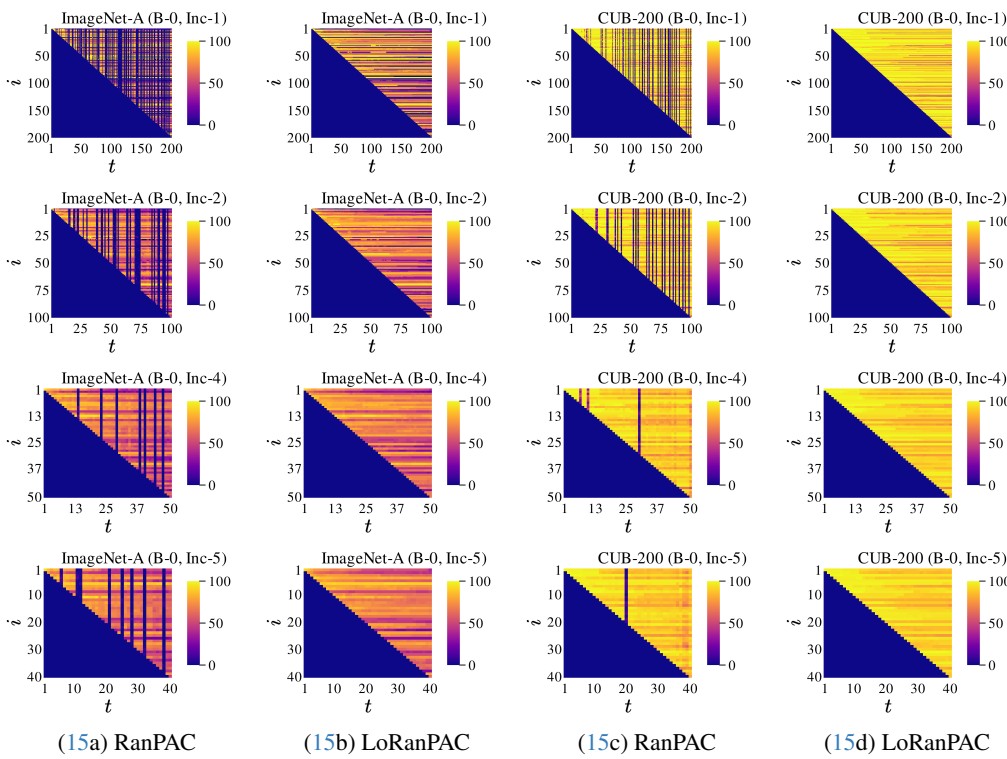

Figure 15: Upper triangular accuracy matrices on ImageNet-A and CUB-200 (Inc-1, 2, 4, 5).

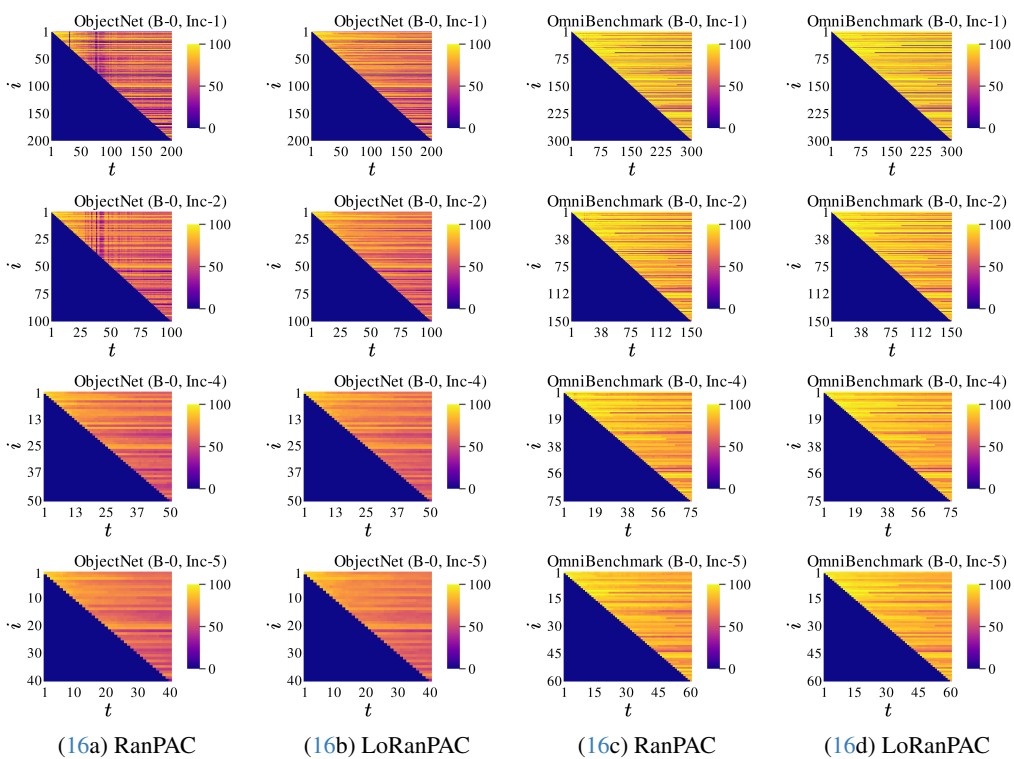

Figure 16: Upper triangular accuracy matrices on ObjectNet and OmniBenchmark (Inc-1, 2, 4, 5).

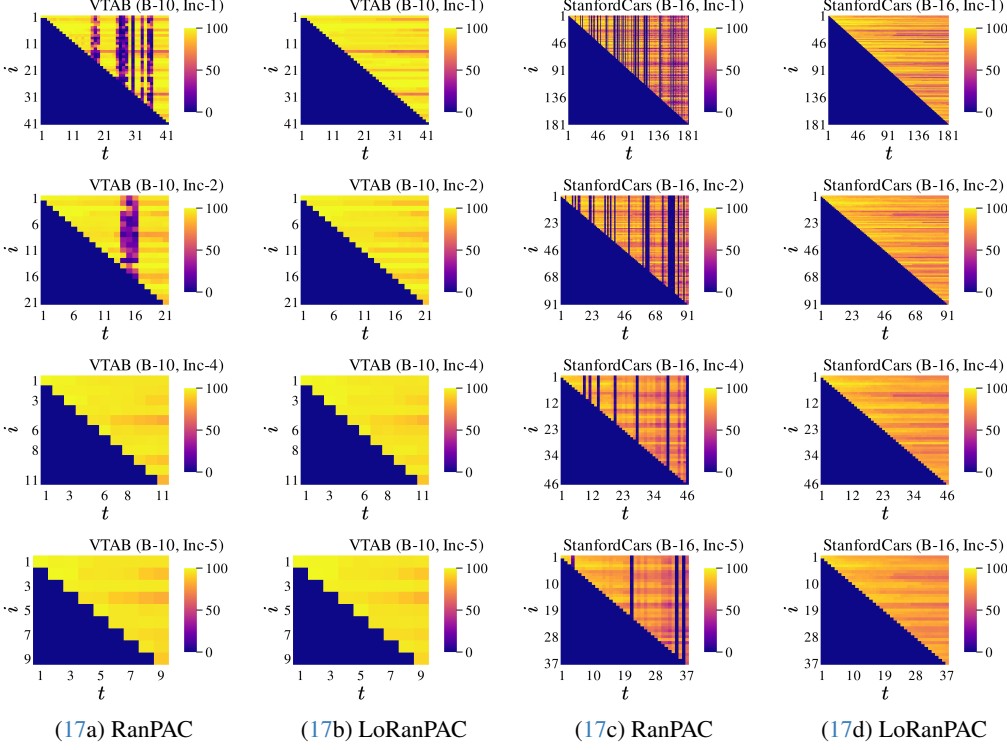

Figure 17: Upper triangular accuracy matrices on VTAB and StanfordCars (Inc-1, 2, 4, 5).

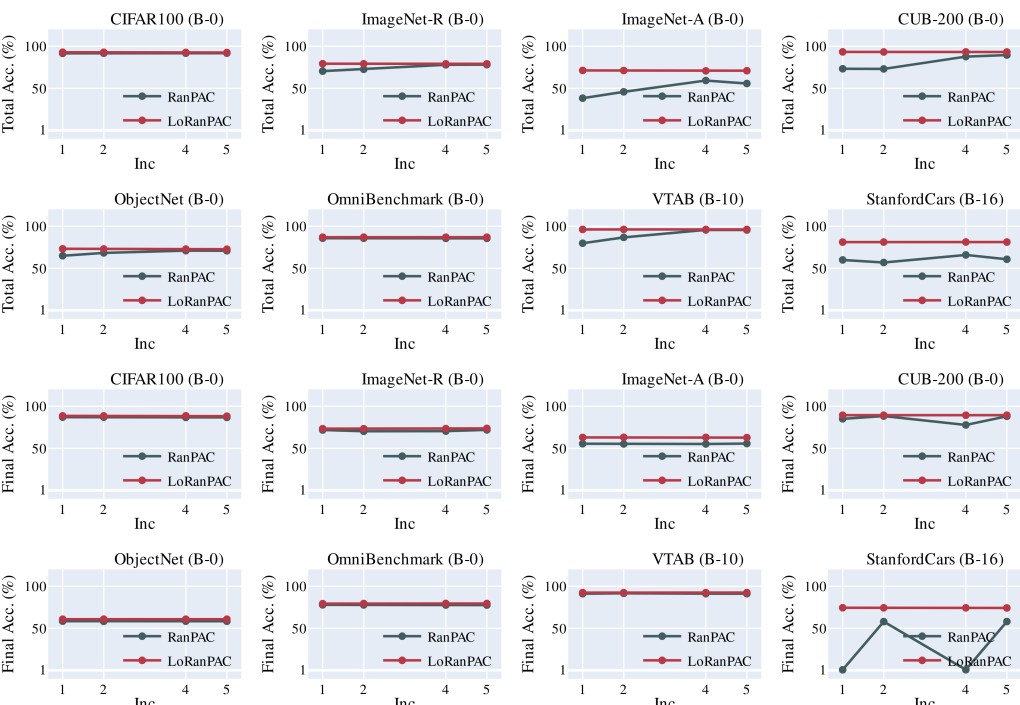

Figure 18: Our LoRanPAC method is stable with respect to class increments of each task (Inc-1, 2, 4, 5), while the accuracy of RanPAC drops for smaller increments. The first two rows plot the total accuracy. The last two rows plot the final accuracy. The figures here essentially plot the averages of the upper triangular accuracy matrices (total accuracy) or its last column (final accuracy) of Figs. 14 to 17. The numerical values of the total and final accuracy for Inc-1 are shown in Table 2.

