# OpenReview forum: "LoRanPAC: Low-rank Random Features and Pre-trained Models for Bridging Theory and Practice in Continual Learning"
_ICLR.cc/2025/Conference — ICLR 2025 Poster_

### Official Review · Reviewer_P9vj · 2024-10-23

**Soundness:** 2
**Presentation:** 3
**Contribution:** 2
**Rating:** 6
**Confidence:** 4

**Summary:**

The paper tackles the problem of class-incremental learning (CIL).
To this end, the paper improves RanPAC (McDonnell et al., 2023), making it more numerically stable as a least-squares problem, by truncating small singular values from the SVD of the (pretrained) embeddings of the samples from *all* previous tasks. The truncation might also positively impact the generalization itself.
The authors link their method to a theoretical framework (ICL by Peng et al., ICML 2023) which derived sufficient conditions to preventing forgetting in linear models, and use this connection to derive some bounds on the training and generalization error of the proposed algorithm.

**Strengths:**

1. The proposed method seems to empirically improve upon the existing RanPAC method (McDonnell et al., NeurIPS 2023).
1. The proposed method is easy to implement.

**Weaknesses:**

# Major concerns

1. **Proposed analysis feels contrived.**
To me, the analytical connection between ICL (Peng et al., ICML 2023) and the proposed method feels overly strained.
    1. To begin with, the ICL formulations presented in Line 111 and 116, are purely theoretical (formalizing a sufficient condition to prevent forgetting), and *cannot* be considered continual learning approaches, since they require storing the data from all previous tasks.
    1. In Line 105 (Eq. 1), the raw feature matrices $X_t$ are randomly mapped to the "effective" matrices $H_t=\text{relu}(PX_t)$ using a random matrix $P\in\mathbb{R}^{E\times d}$ where $E\gg d$.
To facilitate the comparison to the ICL framework, in Line 113, the authors assume that $E$ is so large that it makes the *entire* task sequence feasible (jointly!). However, for most datasets, this would practically make the time and space requirements of the proposed method infeasible.
    1. Theorem 1 also implicitly requires such extreme overparameterization or that the data can already be explained linearly almost perfectly. Otherwise, $\Vert\mathcal{E}_{1:t}\Vert$ would be large, making the bound uninformative ($\frac{M_t-k_t}{M_t}$ is not small).
However, if the data is linearly explained by some model $W^{*}_t$, then ICL trivially obtains 0 loss (since all previous data is stored and optimized simultaneously) and all that is left to analyze is the error stemming from the truncation (which implicitly assumes that the smaller singular values are very small, which might not be the case for arbitrary data).
In my opinion, this is **not** a significant theoretical contribution.

1. **Unclear if accuracy improvements are significant.**
 It seems like the results in the main Table 1 are of a single run per model-dataset combination. No standard deviation is reported. Given that most of the improvements upon RanPAC are somewhat small, it is unclear if these are indeed statistically significant.

---

## Summary of weaknesses

 Considering all of the above, I believe the paper constitutes a nice incremental improvement over RanPAC (using truncation instead of regularization), but I am not convinced that the analysis makes the paper valuable enough for ICLR.

-----

# Minor concerns

I do not expect any response on the following concerns but I recommend addressing them in any future revised version.

1. Many times (e.g., Line 160), the authors specify constant values like $\lambda = 10^{4}$ and $\lambda = 10^{-5}$  and refer to them as being very small or very large. However, I find these references problematic since the regularization strength in least-squares problems is meaningless without taking into account the scale of the data and the fitting error. Similarly, I would even go further and say that the main benefit from the proposed method is not that it gets rid of the smallest singular values, but that it remedies the *condition number* of the matrices. The smallest singular values can only be considered "small" when comparing to the largest singular values. An alternative method could have limit the condition number instead of using a fixed percentage $\zeta$.

1. I think Figure 1c would be better if it showed the training accuracy, since this is the origin of the problem there.

1. The time complexity analysis should appear immediately after the algorithm, rather than as a side note on page 9. Space complexity should also be specified. Moreover, at some point, given enough tasks, $M_t = k_{t-1} + m_t$ can become larger than $E$, which then makes the time complexity $E^3$ (notice that the $M_t - $E$ smallest singular values are zeros), just like in RanPAC.

1. In Line 130, the authors say "the theoretical justification for the strong empirical performance of RanPAC remains limited". I kindly disagree and claim that it is trivially justified, since according to the formula in Line 124, it just minimizes *all* the losses simultaneously.

1. To be slightly more precise, step 3 in Algorithm 1 should use the floor operator $\lfloor(1-\zeta)M_t\rfloor$.

1. The paper would be more readable if the dimensions are specified when matrices are first presented, e.g., the authors should specify the dimensions of ${\widetilde{U,\Sigma,B}_{1:t}}$ in Eq. (3).

1. The sentence in Line 319 (about the values of $\delta$) is unclear.

1. The message of Figure 6 is unclear.

1. In Theorem 1, I believe $\frac{M_t-k_t}{M_t}$ is simply $(1-\zeta)$.

1. In Theorem 1, specify the subscript for the norm used for $\mathcal{E}_{1:t}$.

1. The presentation is often overcomplicated and inconsistent. For instance, in line 105 (Eq. 1), a projection matrix $P$ is introduced, while most of the analysis throughout the paper, especially in Section 5, completely ignores that matrix and deals with arbitrary deterministic feature matrices $H_t$. Moreover, I would consider presenting the ICL details only after the algorithm itself, as a retrospect.

**Questions:**

Other than what I wrote above,

1. I could not immediately understand why both RanPAC and the proposed method outperform the $LC(H_{1:T})$ benchmark (trained with the original cross entropy loss like explained in Appendix J). Even if the data matrices are ill-conditioned, early stopping (e.g., when performing a fixed number of steps) should create a remedying effect and help generalization. Is it possible that a better hyperparameter tuning is needed here? Did the authors check the training *and* test curves throughout the training of this competitor?

1. In Figure 4a, what is going on with the scaling of the x-axis? What was the maximal truncation percentage that you examined? I would expect seeing a sharp decrease at the for high values, but it is not apparent. Moreover, why can we truncate so much without losing performance even though the eigenvalues do not decrease so quickly (as seen from Figure 1a)?

---

> ### Author Response · Authors · 2024-11-13
> **Reply to Reviewer P9vj (Major Concern 1)**
>
> Dear Reviewer P9vj,
>
> Thank you for your thoughtful comments. We understand your concerns. We believe they are due to some misunderstandings of our algorithm and motivations. Some misunderstandings arose as we lacked space in the main paper, and the other was caused by slight inaccuracy on our part. However, all these are easy to fix, and we take the opportunity to respond to your comments and justify our method. In our revision, we will make some space (e.g., put Fig 6 in the Appendix) and revise accordingly.
>
> **Comment 1**: *...the ICL formulations in Line 111 and 116 ... require storing the data from all previous tasks.*
>
> **Reply 1**: While these formulations require access to past data, and similarly in the RanPAC formulation (in Line 124), they admit continual implementations. We chose to state their offline versions for conciseness.
>
> **Comment 2**: *...the authors assume that $E$ is so large that it makes the entire task sequence feasible (jointly!)*
>
> **Reply 2**: We address it by considering the following two points:
> - First, as indicated in the top post, a larger $E$  gives higher accuracy (Fig 8, Appendix K.4). This is a practical motivation rather than a theoretical convenience for ICL to be feasible.
> - Second, assuming $E$ is large allows us to review ICL and RanPAC *very quickly* in a unified manner and with minimum texts and minimum complication.
> - Third, our method works for any $E$. Indeed, although the formulations presented in Lines 111, 116, 243 require a large $E$ to be feasible, the computational solutions stated in Eqs. 2 and 4 do not. In particular, when we have $M\_t > E$, the solution  $\overline{W}\_t\gets Y_{1:t}\overline{V}\_{1:t}\overline{\Sigma}\_{1:t}^{-1}\overline{U}\_{1:t}^\top$ in Eq. 2 is the solution to the normal equations $W\tau\_{k\_t}(H\_{1:t})\tau\_{k\_t}(H\_{1:t})^\top = Y_{1:t}\tau\_{k\_t}(H\_{1:t})^\top$, which means the offline version of ICL-TSVD would minimize the MSE loss $\\| W \tau\_{k\_t}(H\_{1:t}) - Y\_{1:t} \\|_F$. Therefore, while we agree that ICL is *purely theoretical*, it is precisely our contribution to propose ICL-TSVD that makes it practical.
>
> **Comment 3**: *Assuming E is too large would practically make the time and space requirements of the proposed method infeasible for most datasets*.
>
> **Reply 3**: If we agree that the case of large $E$ is practically motivated, and that prior methods can't handle it, then it is precisely our contribution to address the computational challenges with large $E$ via a method of complexity $O(E(k\_{t-1}+m\_t)^2)$ per task. We showed on multiple datasets that our method is feasible for $E$ large.
>
>
> **Comment 4**: *Theorem 1 implicitly requires such extreme overparameterization or that the data can already be explained linearly almost perfectly. Otherwise, ... making the bound uninformative ($\frac{M\_t-k\_t}{M\_t}$ is not small)*
>
> **Reply 4**: We are motivated to consider $E$ large, but the theorems do not assume it. We should have made clear that theorems hold for any $E,M_t$, and any $k\_t\in [0, \min\\{ E, M\_t\\})$, not just for $k\_t=(1-\zeta)M\_t$. The term $\frac{M\_t-k\_t}{M\_t}$ vanishes if $M\_t \to \infty$ and if we only truncate the smallest eigenvalues (and in our experiments, there are just a few of them, which means $M\_t-k\_t=o(M\_t)$); this is when the theorems can be informative.
>
> **Comment 5**: *all that is left to analyze is the error stemming from the truncation. In my opinion, this is not a significant theoretical contribution.*
>
> **Reply 5**: While we appreciate your very high standard for *significant theoretical contributions*, we'd like to argue the following:
> - such analysis is not entirely trivial, in our humble opinion; e.g., the upper bounds have complicated dependencies on quantities such as $a\_t$ and $\gamma\_t$.
> - our theory is the first such result, to the best of our knowledge, for truncated incremental SVD (since Bunch & Nielsen,1978), for continual learning with pre-trained models, and for continual principal component regression.
> - our contribution is not purely theoretical, and our theory is practically motivated: It analyzes the effects of continual truncation to provide extra insights into our method, as we believe our algorithmic contribution is very strong and improves prior works in many important ways.
>
> **Comment 6**:  *... which implicitly assumes that the smaller singular values are very small, which might not be the case for arbitrary data ...*
>
> **Reply 6**: We agree that our theorems might not be informative for all data. This is inevitable, as the truncation throws away information, and we often need the assumption of large eigengaps in this type of analysis. For us, deriving a theorem that is tight for all data is an ambition; we simply want to truncate the smallest singular values (cf. Fig 1), and our theory just shows that it works.
>
> **Please feel free to let us know which part of your major concern 1 remains alive. We'd be glad to have further exchanges.**

---

> ### Author Response · Authors · 2024-11-13
> **Reply to Reviewer P9vj (Major Concern 2)**
>
> The other major concern is that it was *unclear to the reviewer if accuracy improvements are significant*, the reason being:
> > It seems like the results in the main Table 1 are of a single run per model-dataset combination. No standard deviation is reported. Given that most of the improvements upon RanPAC are somewhat small, it is unclear if these are indeed statistically significant.
>
> **Our Reply**: Thank you for raising this point. We argue that the results in Table 1 are statistically significant, as it covers various datasets under different settings. Also, note that in Table 1 we use the same random seed and task order and split as prior works, and we use hyperparameters directly from prior works, without taking advantage of tuning them for our method.
>
> Such uniform improvements in the CIL setting (Table 1) are not seen in the follow-up works of RanPAC, many of which have been accepted to top ML/CV conferences. Some prior works have complicated continual learning strategies that still couldn't outperform RanPAC in the CIL setting, indicating that  RanPAC is a *very strong* baseline (see, e.g., Table 1 of  Zhou et al. 2024a). It is only in our paper that all comparisons to RanPAC are highlighted with a detailed analysis of why our method works better (Line 159, Line 482).
>
> Furthermore, Fig 5 and Table 2 show that our method is significantly more scalable and more accurate.
>
> On the other hand, we agree that reporting standard deviation is useful as it can better illustrate the strength of our method (e.g., more stable than RanPAC). In Table 16, Appendix L.3, of our revision, we run the same experiments as in Table 2 multiple times, with different random seeds, exposing that RanPAC has huge standard deviations on several datasets.
>
> **We hope this gives a refresher view of our algorithmic contributions. Please also take a look at our top post, Thank you for your patience thus far.**

---

> > ### Comment · Reviewer_P9vj · 2024-11-25
> > **Re: Major concern 2**
> >
> > I thank the authors for the revisions in the experimental part.
> >
> > 1. Table 16 is a good improvement upon Table 2. Why not simply updating Table 2?
> >
> > 1. I am not sure why the authors insist on not adding STDs to Table 1.
> > I was not convinced by their response that such an addition is methodologically unnecessary.

---

> ### Author Response · Authors · 2024-11-19
> **Reply to Reviewer P9vj (Questions and Minor Concerns)**
>
> **Question 1**: *Why do both RanPAC and the proposed method outperform the LC ($H_{1:T}$)*?
>
> **Answer 1**: We checked the test accuracies of  LC ($H_{1:T}$) at every epoch. It was no better than the presented results. We trained it for 20 epochs in the presented experiments, and we found the accuracy got slightly worse when we trained it for 30 epochs; this might be some form of early stopping you suggested. A similar phenomenon is shown in Table 1 of the RanPAC paper (https://arxiv.org/abs/2307.02251) for LC ($X_{1:T}$), called joint linear probe in that table.
>
> The plausible explanation is that SGD (with decreasing stepsizes) only finds the exact global minimizer as the number of iterations tends to infinity, while RanPAC and ICL-TSVD find it in closed-form. Another factor to consider is that we used the cross-entropy loss for SGD, while RanPAC and ICL-TSVD have MSE losses. It is not clear how the two losses would compare beyond Remark 2. (We tried MSE losses for SGD, and the accuracy was very low after running more than 20 epochs.)
>
> **Question 2**: *In Figure 4a, what is going on with the scaling of the x-axis? What was the maximal truncation percentage that you examined? I would expect seeing a sharp decrease at the for high values, but it is not apparent. Moreover, why can we truncate so much without losing performance even though the eigenvalues do not decrease so quickly (as seen from Figure 1a)?*
>
> **Answer 2**: There is no issue with the x-axis; we tested a range of truncation percentages from [0\%,90\%], and for clarity, we only reported the results for [0\%, 5\%, 25\%, 50\%, 90\%] in Fig 4a.
>
> This figure shows that, as we truncate more, the method runs faster without sacrificing the accuracy too much. We hope this convinces you that the complexity of time and space can be very low by controlling the truncation percentage. In RanPAC, there is no such control, and its implementation has $O(E^3)$ complexity.
>
> Actually, the eigenvalues do decrease very quickly as per Figure 1a (and this is perhaps why the test accuracy is not affected too much):
> - There are a few very small eigenvalues of order $10^{-5}$; they're likely due to numerical errors, and they might have just been $0$ if we could compute the exact SVD.
> - There are just a few top eigenvalues (of order $10^{11}$), and they quickly decrease to much smaller values $10^{6}$; we believe the eigenvalues of order $10^{6}$ are the actual *tail* of the spectrum. Also note that the effective rank (or stable rank) is just 2 or 3 (cf. Fig 9a, Appendix K.5).
>
> **Minor Concerns**: Thank you very much for all the details. We understand them, and we agree with the comments to a great extent. We will address all of them after the decisions are made or in the next public version of our manuscript. Please understand that, like all ICLR authors, we are running busy and crazy with the rebuttal.

---

> ### Comment · Reviewer_P9vj · 2024-11-25
> **Re: Major concern 1**
>
> I have read the authors' long response.
> However, I do not believe it answers my Major Concern in the original review.
>
> Specifically,
> 1. The authors make an effort to convince me that the proposed method is feasible for diverse regimes of $d, E, M_t$, but this is *not* my concern; I do *not* doubt the feasibility of the proposed method nor its reported empirical success.
>
> 2. I *do* doubt that the connection to ICL adds anything meaningful to the paper and to the empirical success of the method.
> As I explained in my review, I find Theorem 1 informative only when $\Vert \mathcal{E}_{1:t}\Vert$ is small (which holds information only about the data, not the algorithm).
> In such cases, where the labels are approximately explained linearly by the data, line 7 in the algorithm essentially says "we construct the solution using a (sequentially-built) low-rank approximation of the (left-)singular values of the 'accumulated' feature matrix of *all* tasks seen so far". For linear data this is reasonable and very straightforward since it approximates a perfect solution so far, but I do not see what we learn from "linking" it to ICL.
> Moreover, the first term in the theorem simply says "if we throw away only insignificant directions it will incur a low loss". I do not feel like it says a lot more than classical results for stationary settings (where removing insignificant directions incur a loss defined by the truncated singular values).
> I do not believe that my standard here is "very high".
>
> 3.  The authors also mentioned that their "upper bounds have complicated dependencies on quantities such as $a_t$".
> Do the authors find $a_t$ to be an informative quantity?
>
> 4. Coming to think of it again, all of this sounds *closely* related to PCA-OGD (Doan et al., 2020) and their proven bounds, which are not discussed at all.

---

> ### Author Response · Authors · 2024-11-25
> **Authors' reply: Major concern 2**
>
> Thank you for the follow-up comments on the experimental part.
>
> 1. *Table 16 is a good improvement upon Table 2. Why not simply updating Table 2?*
>
> - We wanted to keep the original manuscript untouched so that the reviewers and ACs have better access to the original version. We will, of course, replace Table 2 with Table 16, after the decisions are out or in our next revision.
>
> 2. *I am not sure why the authors insist on not adding STDs to Table 1. I was not convinced by their response that such an addition is methodologically unnecessary.*
> - We are not resisting it. We do agree that having standard deviations shown is important; this is why we have run Table 16.
> - We had concerns that Table 1 is too large to include more details. Also, the experiments would take a lot of time as many methods are slow (everybody was running experiments for the rebuttal, and we had limited GPU resources). Our plan was to revise Table 1 in the next major revision.
> - That being said, we still have two days to respond; please stay tuned, and by Nov 27, we will at least include a comparison between RanPAC and our method with standard deviations under the same setting as in Table 1.

---

> ### Author Response · Authors · 2024-11-25
> **Authors' reply: Major concern 1**
>
> Dear Reviewer P9vj,
>
> Thank you for your time in reading all of our responses and for agreeing with the empirical success of the ICL-TSVD method.
>
> We also thank you for the follow-up interrogations and the opportunity for us to clarify.
>
> For the new concern in relation to PCA-OGD, we kindly refer you to our reply to **Reviewer XHQX**. There, we discussed PCA-OGD in detail and how our paper improves it in both theory and practice, which we hope you will appreciate. In fact, given PCA-OGD, we believe we can now better address your concerns.
>
> **Let us begin with the comments on $a\_t$:**
> - By saying that *the upper bounds have complicated dependencies on quantities such as $a\_t$*, we mean the upper bounds are not trivial to prove. Complicated dependencies on $a\_t$ do not contradict the fact that $a\_t$ is an intuitive and suggestive quantity, as it is just the sum of the maximum eigenvalues continually truncated.
> - Comparing our theorems to the corresponding result of PCA-OGD, we hope you would appreciate that we are addressing a more challenging problem (continual truncation, as we explained to **Reviewer XHQX**), and that our results are much simpler with only two intuitive quantities $a\_t$ and $\gamma\_t$; the corresponding result in PCA-OGD has more than 8 quantities. Similarly, more complicated are the quantities defined in Theorem 3.1 of a recent paper accepted to ICML 2024 (https://arxiv.org/abs/2405.17583).
>
> **Next, thank you for agreeing that we can do nothing about the noise term without extra assumptions on data. We now explain the links to ICL from our perspective:**
> -  **Reviewer XHQX** justified the connection between ICL and the minimum norm solution.
> - ICL (with its minimum-norm version) is how we begin to think about the problem, and RanPAC inspires us to consider the network architecture. Sections 2 and 3 introduce them and their instability. There, the links to ICL (and RanPAC) are important as they help develop the story and motivate our approach.
> - We name our method *ICL-TSVD* as its connection to ICL is closer than RanPAC (cf. Lines 116, 242, Eqs 2, 4). It is just a name. We are open to other names.
> - **Our theorems are proved independently and do not rely on ICL in any way** (similarly, PCA-OGD is inspired by OGD, while its theorems do not rely on the OGD method). We mention this to clarify the following point (we somehow overlooked this comment, and maybe this is why we are confused by other comments):
> > *The authors link their method to a theoretical framework (ICL by Peng et al., ICML 2023) ... and use this connection to derive some bounds on the training and generalization error of the proposed algorithm.*
>
> **Please note that we didn't use the connection to ICL to derive theoretical bounds.** (The paragraph at Line 336 is only to give a feeling about the effects of truncation, if that is the source of confusion.)
>
> Given these considerations, we couldn't see how the link to ICL becomes a **major** concern, as the link does not have any impact on our theory or experimental results, and it only serves together with RanPAC as a motivation to develop our approach. We expect it to be addressed easily by revising the manuscript and rephrasing (if you also think so, please give us more details).
>
> Please let us know whether this addresses your concern and whether you have new ones.

---

> ### Author Response · Authors · 2024-11-25
> **Authors' reply: Major concern 1 (continued)**
>
> Finally, we address the last comment:
> > *I do not feel like it says a lot more than classical results for stationary settings (where removing insignificant directions incur a loss defined by the truncated singular values).*
>
> We note that we are addressing the continual learning case, where we do not maintain the past data and we have minimum assumptions. In this case, there is no hope of getting a result stronger than what the "classical results for stationary settings" can suggest. However, our theorems precisely capture the effects of continual truncation (which is unknown previously), in such a way that:
> - the bound of the training MSE loss is zero if we do not truncate;
> - the bound of the generalization MSE loss coincides with Huang et al. (2022) in the offline case (a single task) under their exact assumptions. This result of Huang et al. (2022) is more recent than classic.
>
> In retrospect, the information you already got from the theorem is that the continual truncation would perform comparably with the offline static case if we truncate insignificant directions. This is previously unknown. We proved it.
>
> Why is such a result, then, a *major* concern rather than a nice, novel contribution where we can recover the classical results for stationary settings with a more general result for continual truncation?
>
> Respectfully, we invite you to consider the following two analogies:
> - In continual learning practice, researchers will be very proud if their method has similar accuracy to the static baseline (i.e., joint training).
> - gradient descent for convex smooth optimization has (unaccelerated) convergence rate $O(1/k)$ where $k$ is the number of iterations. This result has been known for decades. Many papers, including several seminar works, have been published to show that gradient descent variants (e.g., coordinate descent methods [1,2]) have a convergence rate of $O(1/k)$. If we were to apply your standard and your argument, all these subsequent works would be rejected as **they are not valuable enough for ICLR: the proved bound on the convergence rate didn't say too much beyond the $O(1/k)$ rate we already know about gradient descent**.  But we hold a different view. We appreciate that these methods are different from gradient descent, and in fact generalize it. In our case, we analyze a continual learning method, different from an offline method, and obtain a result that generalizes the bounds in the offline case or the case of truncating only once.
>
> We are truly truly confused. We look forward to your reply so we can find a common ground.
>
> > [1] Yu. Nesterov, Efficiency of Coordinate Descent Methods on Huge-Scale Optimization Problems (https://epubs.siam.org/doi/10.1137/100802001)
>
> > [2] Amir Beck and Luba Tetruashvili, On the convergence of block coordinate descent type methods (https://epubs.siam.org/doi/abs/10.1137/120887679)

---

> ### Comment · Reviewer_P9vj · 2024-11-26
>
> I decided to raise my score to 6.
>
> - The authors agreed to publish the STDs in the next version of the manuscript. This seems important to me since the whole point is that RanPAC is unstable and the improvements of ICL-TSVD upon it are often small.
>
> - I was convinced during the discussion with the authors that I was too harsh in my initial review.
>
>      - Personally I still believe that the connections to ICL are rather contrived and mostly make the presentation cumbersome (I had other remarks in my initial review that the authors promised to address in future revisions). I would have more easily accepted this paper as a short paper focusing mostly on the algorithm itself. However, the authors convinced me that the theory they developed may be interesting for the community and is not trivial. At this point, it seems like a matter of personal taste.
>      - I would still recommend refining the messages of the paper, and its (more interesting) perspectives as a sequential truncation methods and PCA-OGD (see next).
>
> - In another [reply](https://openreview.net/forum?id=bqv7M0wc4x&noteId=LRxUXxa0TC) the authors discussed the relations to PCA-OGD:
>     - I find the algorithmic "difference" between the methods to be rather small. The difference between GD and SVD in a linear setting (or layer) is insignificant and often boils down to mere implementation details. However, I do agree now that there are differences in the truncation policies, e.g., since PCA-OGD doesn't store the singular values and applies PCA only to the current task's gradients. This could easily be adjusted to be very similar to ICL-TSVD, but accumulates to a sufficient "delta" that can be considered a different algorithm.
>     - I agree that the theorem in the paper reviewed here is easier to understand than in the other paper and is somewhat more elegant.
>     - I suggest elaborating on these connections in the updated manuscript (at least in an appendix).

---

> > ### Author Response · Authors · 2024-11-28
> > **Thank you for the reply**
> >
> > Dear Reviewer P9vj,
> >
> > Thank you for being open-minded and revising your assessment of our work. We are grateful for your expertise and insights, which have helped improve our paper.
> >
> > Based on the discussions, we agree to revise the message by reorganizing the paper and changing its structures (see Appendix L.4 for full details):
> > - We will introduce the method directly (after section 1 and some basic motivations), by merging Section 4 and part of Sections 2, 3.
> > - In the next section, we will provide a complete picture and discuss its connection to all the mentioned methods, including RanPAC, PCA-OGD, ICL, and more.
> > - Based on the above two sections, we will revise the introduction accordingly, in order to make the story consistent.
> >
> > Finally, just in time, we now have multiple runs of RanPAC and ICL-TSVD under the same setting Table 1. The results, with standard deviation, are reported in Table 17, Appendix L.3, of our revised manuscript. All other methods in Table 1 will be included in Table 17 with standard deviations given more time.
> >
> > Thank you again for your time and feedback.

---

### Official Review · Reviewer_iVyo · 2024-11-01

**Soundness:** 3
**Presentation:** 3
**Contribution:** 2
**Rating:** 5
**Confidence:** 3

**Summary:**

This paper introduces a new algorithm, ICL-TSVD, designed to bridge the gap between theory and practice in continual learning (CL) with pre-trained models. It integrates an empirically strong approach RanPAC within the Ideal Continual Learner (ICL) framework. Specifically, ICL-TSVD lifts pre-trained features into a higher dimensional space, then addresses the ill-conditioning of these lifted features through continually SVD truncation, and solves the truncated form of an over-parameterized minimum-norm least-squares problem. Theoretically, it is shown that the method achieves small training and generalization error under certain conditions. Empirically, ICL-TSVD demonstrates stability across various hyperparameter settings and outperforms other CL baselines across multiple datasets.

**Strengths:**

1. The paper clearly demonstrates the contribution of the proposed approach and is easy to follow.

2. Extensive experiments across multiple datasets show that ICL-TSVD remains stable regarding hyperparameter selection (including regularization parameter $\lambda$ and truncation percentage $\zeta$) and consistently outperforms previous CL baselines, particularly RanPAC.

3. It provides theoretical guarantees for the proposed method ICL-TSVD by revealing a recurrence relation throughout the CL process, showing that it has small estimation and generalization errors when a suitable fraction of SVD factors are truncated.

**Weaknesses:**

1. The continual Implementation of $\overline{\bf{W}}_t$ involves the computation of $\widetilde{\bf{B}}_t$, which has the form of $\widetilde{\bf{B}}_t = [\widetilde{\bf{U}}\_{1:t-1}\widetilde{\bf{\Sigma}}\_{1:t-1}, \widetilde{\bf{H}}_t]$ when $t\geq 2$ according to Eq. (3). Appendix C.1 briefly mentions that it is empirically found continually updating all SVD factors $\widetilde{\bf{U}}\_{1:t}, \widetilde{\bf{\Sigma}}\_{1:t}$ and $\widetilde{\bf{V}}\_{1:t}$ lead to large test errors. However, it remains unclear why authors use $\widetilde{\bf{B}}_t = [\widetilde{\bf{U}}\_{1:t-1}\widetilde{\bf{\Sigma}}\_{1:t-1}, \widetilde{\bf{H}}_t]$ instead of $\widetilde{\bf{B}}_t = [\widetilde{\bf{U}}\_{1:t-1}\widetilde{\bf{\Sigma}}\_{1:t-1}\widetilde{\bf{V}}\_{1:t-1}, \widetilde{\bf{H}}_t]$ from a theoretical perspective. What are the difference and connection between these two implementations? Why just maintaining $\widetilde{\bf{U}}\_{1:t-1}$ and $\widetilde{\bf{\Sigma}}\_{1:t-1}$ can serve as a substitution for all SVD factors, what is the motivation for that?

2. In Section 6.2,  ICL-TSVD chooses embedding dimension $E = 10^5$ while RanPAC uses the default choice $E=10^4$. It is not so convincing that this is a fair comparison although the authors claim that ICL-TSVD’s implementation is more scalable and more efficient than RanPAC’s. Figure 8 and Table 13 in Appendix K indeed show that ICL-TSVD does not outperform RanPAC on about half of the datasets.

3.
  - For Theorem 1, take Figures 3 and 4 for example, the truncation percentage is set to 25%. Then based on eigenvalues vs. eigenvalue index shown in Figure 1, $a_t$ should be at least in the order of $10^5 \sim 10^6$, and $\gamma_t$ is around or slightly larger than 1. In this case, the bound for the estimation error can be very large even if the unknown ground-truth $\\|\mathbf{W}_t^*\\|_F^2$ and noise $\\|\mathcal{E}\_{1:t}\\|^2$ are reasonably small, which means that the theoretical guarantee does not match with empirical results. And similar issue exists in Theorem 2 as well.
  - Furthermore, without independent i.i.d. Gaussian vectors assumption on $\bf{h}$ and the columns of $\bf{H}\_{1:t}$ (which is unrealistic for practice), we may not guarantee that $\\|\Lambda - \frac{1}{M_t}\bf{H}\_{1:t}\bf{H}\_{1:t}^{\top}\\|$ converges to 0 as $M_t\rightarrow\infty$.

4. Figure 5 shows the comparison of training times for varying embedding dimensions $E$ between ICL-TSVD and RanPAC. How about comparison with other CL baselines in terms of training time? Typically, the computational cost of SVD-type algorithms is large, is ICL-TSVD still faster than other CL methods under the current experimental settings?

**Questions:**

1. In Table 2, why the final accuracy of RanPAC on StanfordCars dataset is only 1.19%? Could you provide an explanation for it?

2. Figure 5 does not plot the training time for RanPAC when embedding dimension $E = 50\mathrm{K}, 100\mathrm{K}$, is this because the training process is too slow?

3. How about the running time of ICL-TSVD compared to other CL baselines, especially those that do not utilize SVD in their implementation?

4. For original definition of $\overline{\bf{W}}_t$ in Eq. (2) and practical implementation of $\widetilde{\bf{W}}_t$ in Eq. (4), do they close to each other? In other words, is there any theoretically guarantee showing that $\\|\overline{\bf{W}}_t - \widetilde{\bf{W}}_t\\|_F$ is small in general?

**Minor issues and typos**

- In Eq. (5) of Section 5, the denominator of $\gamma_t$ should be $\max\_{1\leq i\leq t-1}\mu\_{k_{i+1}}(\widetilde{\bf{B}}_i\widetilde{\bf{B}}_i^{\top})$ instead of $\max\_{1\leq i\leq t-1}\mu\_{k\_{i}+1}(\widetilde{\bf{B}}_i\widetilde{\bf{B}}_i^{\top})$.

**Details Of Ethics Concerns:**

No ethics concerns.

---

> ### Author Response · Authors · 2024-11-13
> **Reply to Reviewer iVyo (Weaknesses, Part 1)**
>
> Dear Reviewer iVyo,
>
> Thank you for your time on the paper and the appendices. We appreciate your comments on weaknesses and questions, as well as the deep insights shown there. Our rebuttal takes your comments into full consideration, and we hope it will earn your support for acceptance.
>
> We will revise our paper accordingly based on your comments and our responses. Please feel free to raise any further concerns, and we will be happy to address them during the discussion phase.
>
>
> **Comment 1**: *... it remains unclear why authors use $\widetilde{B}\_t:=[\widetilde{U}\_{1:t-1}\widetilde{\Sigma}\_{1:t-1}, H\_t]$ [as defined in Eq. 3], instead of  $\widetilde{C}_t:=[\widetilde{U}\_{1:t-1}\widetilde{\Sigma}\_{1:t-1}\widetilde{V}\_{1:t-1}^\top, H\_t]$*.
>
> **Reply 1**: This is a good point that we could have clarified in Appendix C.1. We apologize for the confusion. We address it below:
> - First, one might consider maintaining all TSVD factors $\widetilde{U}\_{1:t},\widetilde{\Sigma}\_{1:t},\widetilde{V}\_{1:t}$ and labels $Y_{1:t}$, and forming the classifier $\widetilde{W}\_t\gets Y_{1:t}\widetilde{V}\_{1:t}\widetilde{\Sigma}\_{1:t}^{-1}\widetilde{U}\_{1:t}^\top$ (cf. Eq. 2). Empirically, this gives large test errors, and it is the method we implied in Appendix C.1.
> - This led us to the strategy indicated in Eqs 2 and 4, that is to maintain $\widetilde{U}\_{1:t},\widetilde{\Sigma}\_{1:t}$ and form a solution as per Eq. 4. To do so, the first idea is to update TSVD factors using $\widetilde{C}_t$; interestingly, this is mathematically equivalent to using $\widetilde{B}_t$. Indeed, with the QR decomposition $Q\_t R\_t = (I_E - \widetilde{U}\_{1:t-1}\widetilde{U}\_{1:t-1}^\top) H_t$, we have $$\widetilde{C}_t= [\widetilde{U}\_{1:t-1}\widetilde{\Sigma}\_{1:t-1} \widetilde{V}\_{1:t-1}^\top,\ H\_t  ] =  [\widetilde{U}\_{1:t-1},\ Q\_t  ]  \begin{bmatrix} \widetilde{\Sigma}\_{1:t-1}  & \widetilde{U}\_{1:t-1}^\top H\_t \\\ 0 & R_t \end{bmatrix}  \begin{bmatrix} \widetilde{V}\_{1:t-1}^\top  &0 \\\ 0 & I\_{m\_t} \end{bmatrix}   $$
> Similarly, $\widetilde{B}_t$ equals the product of the first two matrices on the right-hand side. Since the first and third matrices are orthogonal, it suffices to calculate the SVD of the middle matrix on the right-hand side. Since the final classifier in Eq. 4 can be calculated without $\widetilde{V}\_{1:t}$, we could just maintain $\widetilde{U}\_{1:t},\widetilde{\Sigma}\_{1:t}$ and ignore $\widetilde{V}\_{1:t}$. We hope it is clear now.
>
> **Comment 2**: *ICL-TSVD uses $E=100K$ while RanPAC uses its default $E=10K$. It is not so convincing that this is a fair comparison although ... Fig. 8 & Table 13 show that ICL-TSVD does not outperform RanPAC on about half of the datasets.*
>
> **Reply 2**: As indicated in the top post and our answer to your question 2 below, RanPAC is too slow for larger $E$. Here is what we could argue if both methods use the same $E\geq 10000$:
> - ICL-TSVD and RanPAC would have comparable accuracy (cf. Fig. 8, Table 13). RanPAC might even have a slight advantage (when it works stably), for two reasons. First, it performs cross-validation, while we simply use $\zeta$=25% without any specific attention paid to tuning it. Second, RanPAC's implementation has *less continual learning flavor*: It updates covariances $H\_{1:t} H\_{1:t}^\top$ and $Y\_{1:t} H\_{1:t}^\top$ and solves the normal equations $W (H\_{1:t} H\_{1:t}^\top + \lambda\_t I)= Y\_{1:t} H\_{1:t}^\top$ with a selected regularization strength $\lambda\_t$; this is what an offline ridge regression method would solve. By doing so, numerical errors would not accumulate over time, while the issue is that it takes $O(E^2)$ space and $O(E^3)$ time, which makes RanPAC infeasible for large $E$. Given these, we found it remarkable that our online ICL-TSVD implementation has accuracy comparable to an almost offline RanPAC implementation.
> - ICL-TSVD is more stable and will outperform RanPAC a lot in CIL with smaller increments (e.g., the setting of Table 2), even with the same $E=10K$.
> - ICL-TSVD is up to 1000 times faster for $E=50K$.
> - RanPAC runs out of memory for $E=100K$.

---

> ### Author Response · Authors · 2024-11-15
> **Reply to Reviewer iVyo (Weaknesses, Part 2)**
>
> **Comment 3**: *(Part 1) The quantities $a\_t$ and $\gamma\_t$ can be large* & *(Part 2) Gaussian assumptions can be unrealistic, and $\| \Lambda - \frac{1}{M_t} H_{1:t} H_{1:t}^\top \|$ might not converge to $0$ as $M_t \to \infty$.*
>
> **Reply 3 (Part 1)**: Indeed $a\_t$ and $\gamma\_t$ can be large, and in this case we need a large $M_t$ to have small upper bounds. This is also why the theory section is titled *Provably Controlled Estimation and Generalization*, i.e., the upper bounds are controlled by where we truncate. More specifically, we take the following perspectives:
> - Our theory is informative in the following sense: the errors are small if we truncate at the position that exhibits a large eigengap and we truncate only small eigenvalues. This suggests we could set $k_t$ by detecting the eigengap and truncating only small eigenvalues. Our theory can provide informative bounds for this strategy. This is nice, as our original motivation is to eliminate the extremely small eigenvalues (cf. Fig 1).
> - However, since there are typically just a few such small eigenvalues, this strategy would truncate only a few eigenvalues, while, empirically, truncating more would gain efficiency without affecting generalization too much (Fig 4a).
> - Despite these thoughts, we make a simple choice: truncate 25% of them. We believe this is not a big limitation, as we have illustrated the effect of choosing $k\_t$, and in our code (to be released), the user can choose different truncation strategies that fit their needs.
>
> **Reply 3 (Part 2)**: We agree that Gaussian assumptions can be unrealistic, so we didn't use them in the theorem. Next we explain why we mentioned Gaussianity and why this assumption is not necessarily needed in our case:
> - We mentioned in the text that  $\| \Lambda - \frac{1}{M\_t} H\_{1:t} H\_{1:t}^\top \|$ goes to $0$ as $M\_t\to \infty$ when $H\_{1:t}$ consists of i.i.d. Gaussian vectors and $\Lambda$ satisfies certain boundedness conditions (Koltchinskii & Lounici 2017). This is the simplest possible explanation we know that allows the reader to quickly understand the term $\| \Lambda - \frac{1}{M\_t} H\_{1:t} H\_{1:t}^\top \|$, without getting into any technical details. We believe this would facilitate digesting our theorem.
> - On the other hand, we can still guarantee $\| \Lambda - \frac{1}{M\_t} H\_{1:t} H\_{1:t}^\top \| \to 0$ even if the Gaussian assumption on $H\_{1:t}$ is replaced with a much weaker condition called *hypercontractivity*; see, e.g., [1]. Specifically, assuming every column of $H\_{1:t}$ is an i.i.d. copy of some random vector $h$, the hypercontractivity condition only requires $h$ to satisfy $$\sup\_{v:\\|v\\|=1} \\frac{\\mathbb{E}^{1/p}| h^\top v|^p }{ \\mathbb{E}^{1/2}| h^\top v|^2 } <\infty$$
> for some $p > 4$. We hope it is now clear that Gaussian assumptions are indeed not required in our results.
>
> [1] Jirak et al., "Concentration and moment inequalities for heavy-tailed random matrices" (https://arxiv.org/abs/2407.12948)
>
>
> **Comment 4**: *Figure 5 shows the comparison of training times for varying embedding dimensions $E$
>  between ICL-TSVD and RanPAC. How about comparison with other CL baselines in terms of training time? Typically, the computational cost of SVD-type algorithms is large, is ICL-TSVD still faster than other CL methods under the current experimental settings?*
>
> **Reply 4**: This comment relates to the following question and here we address them together.
> > **Question 3**: *How about the running time of ICL-TSVD compared to other CL baselines, especially those that do not utilize SVD in their implementation?*
>
> Thank you for raising this point. We mention the following points:
> - First, we justified in the above post that we essentially perform (incremental) *low-rank* SVD, which makes our method faster than conventional SVD-based methods.
> - Then, we note that we didn't compare the running times of other methods in the paper, primarily because our method outperforms them by a large margin and other methods are of different types.
> - On the other hand, we are happy to compare the running times with more baselines. This is shown in Table 15, Appendix L.2, of our revision. In that table, our method is shown to be multiple times faster than prompt-based methods (e.g., L2P, CodaPrompt) and adapter-based methods (e.g., EASE). These methods do not utilize SVDs. They are slower as they need to backpropagate the gradients throughout the large pre-trained ViTs for multiple epochs, to train learnable prompts or adapters (with the ViT parameters frozen).
>
> **Reply to minor typos**: Thank you for reading the manuscript carefully and pointing out the typo. We are now fully aware of it and will make a revision accordingly.

---

> ### Author Response · Authors · 2024-11-15
> **Reply to Reviewer iVyo (Questions)**
>
> **Question 1**: *In Table 2, why the final accuracy of RanPAC on StanfordCars dataset is only 1.19%?*
>
> **Reply 1**: Good question. We explained why RanPAC is unstable and fails in the general case in the post above (*Our Algorithmic Contributions and Improvements in Comparison to RanPAC*).
>
> Below, we aim to explain why it fails specifically in the case you mentioned.
>
> - First, in Table 2, we are given one class at a time. This means the training set is small for every task $t$, so the validation set of RanPAC is also small. In this case, its cross-validation strategy struggles at choosing a good regularization parameter.
> - Then, note that RanPAC chooses $\lambda\_t$ for every task $t$ encountered. A wrongly chosen $\lambda\_t$ would result in nearly zero accuracy on all seen tasks, while the accuracy would recover in the next task, if $\lambda_{t+1}$ is chosen suitably.
> - For StanfordCars, RanPAC fails at the last task in the present setting, which makes the 1.19% final accuracy in Table 2. Figure 18 (B-16, Inc-1) plots the full accuracy matrices of RanPAC on StanfordCars, and the blue columns imply failures and nearly zero accuracies on all seen tasks. In particular, note that the last column of that figure is in blue.
> - Finally, note that Table 2 only contains a single run of the experiments with a seed given by prior work (default in the code repo). In Table 16, Appendix L.3 of our revision, we show the same experiments with 3 random seeds, and report the mean and standard deviations of the accuracies. In that table, the final accuracy of RanPAC improves, as for some other seeds, it does not necessarily fail at the lat task. However, the table also shows RanPAC exhibits large standard deviations on StanfordCars. We will present the new table instead, in our revision.
>
> **Question 2**: *Figure 5 does not plot the training time for RanPAC when embedding dimension $E=50K,E=100K$, is this because the training process is too slow?*
>
> **Reply 2**: Yes, RanPAC is too slow for large $E$. E.g., it takes more than 20 hours with $E=50K$ on ImageNet-A, while our method takes about 1 minute (RanPAC is similarly slow on other datasets). This was also mentioned in the post at the top.
>
> We hope this clarifies why our comparison with RanPAC is fair, even with different $E$. If both methods use the same $E$, we end up with the conclusion that our method is up to  $1000$ times faster, while being more stable with comparable performance.
>
> **Question 3**: *How about the running time of ICL-TSVD compared to other CL baselines, especially those that do not utilize SVD in their implementation?*
>
> **Reply 3**: This was addressed above, together with *Weakness 4*.
>
> **Question 4**: For $\overline{W}_t$ and $\widetilde{W}_t$ defined in Eq. 2 and Eq. 4, respectively, is there any theoretical guarantee on $\\| \overline{W}_t - \widetilde{W}_t  \\|\_F$?
>
> **Reply 4**: Beautiful question. The answer is yes, and we provide such a bound in Theorem 7, Appendix L.1 of our revision. Note that the upper bound of Theorem 7 is meaningful and not loose if we truncate at a position $k\_t$ that exhibits large eigengaps. In fact, to get meaningful bounds, it is in some sense necessary to truncate at such a position $k\_t$ that exhibits large eigengaps; for a (remote) counterexample, see, e.g., Section 2.3.2 in the following reference:
> > [1] Yuxin Chen, Yuejie Chi, Jianqing Fan, Cong Ma, et al. Spectral methods for data science: A statistical
> perspective. Foundations and Trends® in Machine Learning, 14(5):566–806, 2021. (https://arxiv.org/abs/2012.08496)

---

> > ### Comment · Reviewer_iVyo · 2024-12-03
> >
> > I thank the authors for their rebuttal. I will keep my rating.

---

### Official Review · Reviewer_XHQX · 2024-11-04

**Soundness:** 3
**Presentation:** 3
**Contribution:** 3
**Rating:** 6
**Confidence:** 2

**Summary:**

This paper proposes a new method to address instability issues of RanPAC. Part of the contribution is a diagnosis of the cause of the instability which is attributed to the condition number of the features exploding as the number of tasks increases. The proposed method truncates the smaller singular values of the features in an online or continual way, so that it doesn't have to store all of the previous tasks samples. Effectiveness of the new method is validated empirically.

**Strengths:**

The problem and setup is well motivated. The results are natural and the writing is clear. The experimental results are extensive and seem significant.

**Weaknesses:**

It seems to me that the main contribution of this work is an algorithm for continual principal component regression, with an application to training with random features. Specially, the properties of the features that arise from pretraining, random projections and nonlinearity don't seem to matter much to the derivations theoretically. On the other hand, if I view the work mainly as an algorithm for continual principal component regression, then the guarantees given in theorems 1 and 2 are a black box, since the quantities $\gamma_t$ and $a_t$ which have an unpacked recursive form. It is not clear to me how they behave in general.

**Questions:**

In the ICL framework of Peng et al. (2023), they are maintaining a minimum norm solution based on proof of Proposition 5 in that work where they are using the pseudoinverse. So I think the statements on top of page 3 might not be quite accurate.


In lines 202-208 the cause of instability of RanPAC is attributed to the double descent phenomenon. I think it would be helpful to explain this more to the reader.


New notation that is used in lines 242-243 has not been introduced yet.

To judge the significance of the experimental results, it would be helpful if the reported numbers included also a measure of uncertainty.

---

> ### Author Response · Authors · 2024-11-12
> **Reply to Reviewer XHQX**
>
> Dear Reviewer XHQX,
>
> We appreciate your comments on the paper's strengths, and thank you for recommending acceptance with a score of 6. We reply to your comments below. Do let us know if you have further concerns and if we can improve in any other aspects.
>
> **Comment 1**: *It seems to me that the main contribution of this work is an algorithm for continual principal component regression (continual PCR), with an application to training with random features.*
>
> **Reply 1**: Good point. We did mention the connection to continual PCR in Remarks 4 and 6, and, to our knowledge, the notion of *continual PCR* is not mentioned elsewhere; please let us know if that is not the case. We believe the connection we established is important, as many continual learning methods lack theory, while PCR is an important topic in the theory community (or, at least, the references cited in Remarks 4 and 6 are focused on theory); this connection will facilitate our understanding of continual learning by extending existing theoretical results in overparameterized linear models and random feature models to the online/continual case.
>
> **Comment 2**: *... the properties of the features that arise from pretraining, random projections and nonlinearity don't seem to matter much to the derivations theoretically.*
>
> **Reply 2**: We agree with you. On the other hand, note that random projections and nonlinearity affect the spectrum of $H\_{1:t}$, so they play implicit roles in the values of $a\_t$ and $\gamma\_t$ in the upper bounds. We believe a full analysis that takes random projections and nonlinearity into account is possible in a continual learning setting, e.g., by extending existing theorems for random feature models. Such analysis is well-suited for future work, and we will discuss more related works in more detail and mention this future direction in Appendix H.2.
>
>
> **Comment 3**: *... the quantities $\gamma\_t$ and $a\_t$ have an unpacked recursive form. It is not clear to me how they behave in general.*
>
> **Reply 3**: We agree that the appearance of $\gamma\_t$ and $a\_t$ in the upper bound is unconventional, while we argue that we can understand how they behave through the spectrum of $H\_{1:t}$ (or $\widetilde{B}\_t$). E.g., as justified in the paper, $\gamma\_t$ would be very large and $a\_t$ would be small, if we truncate at a position that exhibits large eigengap and the eigenvalues truncated are very small, then the overall upper bound is small. Furthermore, $a\_t$ and $\gamma\_t$ are computable, so we could in principle just plot their values and analyze their behaviors for different truncation strategies. We'd be happy to include such plots if you deem them necessary.
>
>
> **Comment 4**: *In the ICL framework of Peng et al. (2023), they are maintaining a minimum norm solution based on proof of Proposition 5 in that work where they are using the pseudoinverse. So I think the statements on top of page 3 might not be quite accurate.*
>
> **Reply 4**: We apologize for the confusion. The constraint $x\in \mathcal{K}\_{t-1}$ in Proposition 5 of Peng et al. (2023) can be viewed as the constraint $WH\_{1:t-1}=Y\_{1:t-1}$ in the over-parametrized regime.
>
> **Comment 5**: *In lines 202-208 the cause of instability of RanPAC is attributed to the double descent phenomenon. I think it would be helpful to explain this more to the reader.*
>
> **Reply 5**: This would indeed be helpful. In Appendix K.7.1, we had partial empirical justifications regarding this point. To be more accurate, when we say *double descent*, we mean the empirical observation that RanPAC has small training losses yet large test losses.
>
> **Comment 6**: *New notation that is used in lines 242-243 has not been introduced yet.*
>
> **Reply 6**: Sorry for the confusion. If we understand it correctly, *the new notation used in lines 242-243* refers to $\tau_{k_t}(\cdot)$. This is defined in Line 240 by words. It maps a matrix to its best rank-$k_t$ approximation.
>
> **Comment 7**: *To judge the significance of the experimental results, it would be helpful if the reported numbers included also a measure of uncertainty.*
>
> **Reply 7**: This is a good point. Below we justify our original setup and then mention new experiments with a measure of uncertainty:
> - In our original manuscript, we follow the experimental setting of prior work (with the same random seed), and we believe that our experiments are significant as they cover many datasets under various settings.
> - On the other hand, we agree with your advice to include a measure of standard deviation, as this can better highlight the strength of our method. We performed such an experiment in Table 16, Appendix L.2, of our revision. That table shows RanPAC has large standard deviations on multiple datasets, while our method is much more stable.
>
> We hope this alleviates your concern. Given more time, we will include or revise more experiments that we deem necessary to measure standard deviations.

---

> > ### Comment · Reviewer_XHQX · 2024-11-25
> >
> > Hi,
> >
> > I'm not very familiar with the literature on PCR, but there is quite bit of work on online PCA which could be used for regression. PCA has also been studied theoretically for continual learning before in [1]. My understanding is that there it is used to project gradients. Overall, I think a thorough discussion and comparison to  the work related to online PCA would be necessary.
> >
> > Comment 3, I understand that they depend on the spectrum of $H_{1:t}$, so why can't they be written out in terms of (top k ?)  singular values of $H_{1:t}$? Is there no closed form solution ?
> >
> >
> > Comment 4: It is true that the statement of Proposition 5 in Peng et al. (2023) in its' own does not imply that the solution is minimum norm, however in the proof, where they describe how this is solution is computed, they are using the pseudo inverse in a way that would result in the minimum norm solution.
> >
> >
> > Comment 5: Usually double descent refers to the curve of generalization error against model complexity, where the first part is U shaped and then the generalization error decreases (depending on the spectrum). I'm still confused how this relates to that.
> >
> >
> > My comment about the experiments seems addressed.
> >
> >
> > [1]Thang Doan, Mehdi Abbana Bennani, Bogdan Mazoure, Guillaume Rabusseau, and Pierre Alquier. A theoretical analysis of catastrophic forgetting through the ntk overlap matrix. International Conference on Artificial Intelligence and Statistics 2021

---

> ### Author Response · Authors · 2024-11-25
> **Reply to Reviewer XHQX**
>
> Dear Reviewer XHQX,
>
> Thank you for the follow-up comments. We are grateful for your reminder of [1] (PCA-OGD), as there is indeed an interesting connection. We are aware of [1] and cited it, but we overlooked its PCA part. PCA-OGD collects and orthogonalizes eigenvectors to project the gradients. In Algorithm 1 of [1], the set $S\_J$ of eigenvectors grows in size, previous elements are not modified, and *the PCA step only truncates the SVDs of the gradients for the current task*; in our notation, it considers $\tau\_{d}(H\_t)$ for every $t$ to preserve top $d$ eigenvectors. To our knowledge, PCA-OGD has the following differences compared to our method (please let us know if that is inaccurate):
> - (**different algorithms**) PCA-OGD is a variant of gradient descent, while our method is based on SVDs. PCA-OGD does use the SVDs, but its purpose is to construct the eigenspaces where gradients are projected.
> - The set $S\_J$ keeps growing and then PCA-OGD would eventually be infeasible. Also, it does not maintain any information about singular values, so it is unclear which eigenvectors should be discarded when memory is insufficient. In contrast, our method keeps track of singular values, and can truncate SVD factors easily.
> - (**different truncation strategies**) The truncation of each task in PCA-OGD is independent, therefore it amounts to truncating *only once*. This is simpler than the case we consider, as we need to analyze the effects of $\tau\_{k\_t}(\widetilde{B}\_t)$ defined in Eq 3. Such analysis was mentioned in a follow-up work as an open problem [2] (we are happy to discuss further the differences from [2] if needed).
> - (**different theorems**) The type of theorems is different. In [1], the term analyzed is the forgetting, while we provide bounds on training/generalization errors on the joint loss. The analysis of [1] couldn't be transferred to our case in the under-parameterized regime, as OGD inherently requires overparameterization (otherwise, the orthogonal space might be empty). On the contrary, our theorems can be extended to bound the forgetting.
>
> > [2] Youngjae Min, Kwangjun Ahn, and Navid Azizan. One-pass learning via bridging orthogonal
> gradient descent and recursive least-squares. In IEEE Conference on Decision and Control, 2022.
>
> Comment 3: To be more precise, they depend on the spectrum of $\widetilde{B}\_t$, and hence, implicitly, the spectrum of $H\_{1:t}$. We need extra assumptions to relate the spectrums of $\widetilde{B}\_t$ and $\widetilde{H}\_{1:t}$ (similarly to the eigengap assumptions in Theorem 6), and if we do so, we could *potentially* express our results in terms of the spectrum of $\widetilde{H}\_{1:t}$. We didn't go into these complications and only argued in Line 287 that, empirically, the spectrums of $\widetilde{B}\_t$ and $\widetilde{H}\_{1:t}$ are close to each other. In general, there is no closed-form relation (equality) between the spectrums of $\widetilde{B}\_t$ and $\widetilde{H}\_{1:t}$ due to continual truncation, and this is also a main difference from PCA-OGD.
>
> Comment 4: You are correct. We appreciate your insights there. The program in Proposition 5 of Peng et al. (2023) would indeed result in a minimum solution if solved via SVDs. We didn't mention this in the main paper as that would take a little bit more to explain. We promise to mention the connection in the revision in the form of a simple lemma in the appendix.
>
> Comment 5: We apologize for the confusion regarding double descent. The simplest way to address it is to revise our wording. We will remove the phrases about double descent and say RanPAC (in some cases) produces small training errors yet large test errors.
>
> *My comment about the experiments seems addressed.*: Thank you for letting us know:)
>
> Thank you again for your time and follow-up comments, which have improved our work. Many of the above discussions will be included in our paper. Please feel free to let us know if there is still confusion remaining.

---

### Official Review · Reviewer_EXjv · 2024-11-04

**Soundness:** 3
**Presentation:** 3
**Contribution:** 2
**Rating:** 5
**Confidence:** 1

**Summary:**

This paper proposes a method for continual learning. In this method, the singular values are continuously truncated. The authors claim that their method has theoretical guarantees as well as good practical performance.

**Strengths:**

The authors provided comprehensive theoretical and empirical results.

**Weaknesses:**

A simple baseline in continual learning is experience replay. But I did not see the authors providing the results for experience-replay-based method.

**Questions:**

Did the authors compare with experience replay method?

---

> ### Author Response · Authors · 2024-11-12
> **Reply to Reviewer EXjv**
>
> Dear Reviewer EXjv,
>
> Thank you for your comments. We are happy to read the strength that **The authors provided comprehensive theoretical and empirical results.**
>
> **The question is whether experience-replay-based methods are compared. The answer is yes (this is implicit in the paper).**
>
> **Details**. We compared our approach with the joint training method (that replays all training data, rather than just a small fraction). In some sense,  joint training serves as a performance upper bound for experience-replay-based methods.
>
> Specifically, LC ($X_{1:T}$) refers to joint training of a linear classifier using vanilla ViT features $X_{1:T}$, while LC ($H_{1:T}$) is the same approach, but using the random ReLU features $H_{1:T}$. In Table 1, we show our method outperforms both approaches, without storing any of the samples or features $X_{1:T}$.
>
> There could be other variants of joint training. For example, one could consider joint training of a two-layer ReLU network using ViT features $X_{1:T}$. We didn't show its performance in the paper, but we confirm that it is comparable to LC ($X_{1:T}$) and LC ($H_{1:T}$). We'd be happy to add such a result in the revision.
>
> **Please**: Feel free to raise any further concerns, as we will be available to respond in the next few days. For example, if you are interested in seeing a specific kind of experience replay methods, let us know and we will compare them.
>
> **Please**: Dear reviewer, as you've seen, we received divergent reviews. We would be extremely grateful if you could look further at the paper and update your thoughts.
>
>
> Thank you very much,
>
> Authors

---

### Author Response · Authors · 2024-11-13
**Our Algorithmic Contributions and Improvements in Comparison to RanPAC**

We begin by claiming that our method improves over RanPAC in two major ways:
- **Our method is more scalable: Fig 5 shows our method is up to 100-1000 times faster for large embedding dimension $E$**.
- **Our method is more stable: It has 10 points higher accuracy on average than RanPAC in Table 2.**

These improvements are not seen in follow-up works such as Ahrens et al. (TMLR 2024) and Prabhu et al. (NeurIPS 2024), which we discussed in detail in the manuscript.

Below, we explain why RanPAC is unstable and inefficient and how our method improves it.

# RanPAC is unstable and inefficient
- RanPAC is unstable (cf. Lines 197-200), and it fails when its validation set (chosen to be a small fraction of a training set of the current task) is small and not representative of the test data. The validation set is indeed small in class-incremental learning with small increments. **This is a new understanding of RanPAC's failure pattern**, to our knowledge.
- RanPAC is inefficient for large embedding dimension $E$ (Remark 8, Appendix C). For each task $t$, it solves a system of linear equations that amounts to inverting the $E\times E$ (regularized) covariance matrix $H\_{1:t} H\_{1:t}^\top + \lambda I$. This would take $O(E^3)$ time in general. Moreover,  for every task $t$ it solves such equations more than $10$ times (for different $\lambda$ taken from a predefined set) for the purpose of cross-validation.

# Why we need large embedding dimension $E$ and why prior methods couldn't handle large $E$
Roughly speaking, a larger $E$ yields higher accuracy (cf. Fig 8, Appendix K.4). This motivates developing scalable methods for handling cases with large $E$.

On the other hand, as **Reviewer P9vj** rightly pointed out, a larger $E$ would bring significant computational challenges. Indeed, RanPAC and its variants (Ahrens et al. TMLR 2024, Prabhu et al. NeurIPS 2024) all have $O(E^3)$ complexity and can not handle large $E$. More specifically:
- The running times of RanPAC with $E=10K$ (default) and $E=25K$ (the choice of Prabhu et al.) are in Fig 5. In this case our method is up to $100$ times faster.
- For $E=50K$, RanPAC is too slow. E.g., it takes more than **20 hours** on ImageNet-A in the setting of Fig 5 (similarly on other datasets). In contrast, our method takes about **1 minute** on ImageNet-A under the same setting. (This answers a question of **Reviewer iVyo**)
- For $E=100K$, RanPAC runs out of memory.

# Our contribution: a scalable method that works for extremely large embedding dimension $E$

For task $t$, our method performs QR decomposition on an $E\times  (k\_{t-1}+m\_t)$ matrix and SVD on an $(k\_{t-1}+m\_t)\times  (k\_{t-1}+m\_t)$ matrix, where $k\_{t-1}$ is the number of SVD factors preserved after task $t-1$ and $m_t$ is the number of samples in task $t$. This takes $O( E(k\_{t-1}+m\_t)^2)$ and $O( (k\_{t-1}+m\_t)^3 )$ time, respectively.

Importantly, we can control the time complexity by adjusting the values of $k\_{t-1}$ and $m\_t$. E.g., we can set $k\_{t-1}$ small, or if $m\_t$ is too large, we can split task $t$ into several subtasks. In particular, $k_{t-1}$ can be extremely small, as Figure 4a shows the accuracy is not affected too much even if we truncate 90\% SVD factors.

- **Reviewer iVyo** is concerned that SVDs can be costly. This is true in general. But here, essentially, our method performs *low-rank* SVD, and its complexity $O( (k\_{t-1}+m\_t)^3 )$ could be affordable when $k\_{t-1}+m\_t$ is not large, and it is advantageous compared to $O(E^3)$ algorithms when $E$ is far larger.

- **Reviewer P9vj** is concerned that $k\_{t-1} +m\_t$ and $k_t$ would eventually be larger than $E$. But note that we need at least $k_t < \min\\{ E, M_t\\}$ for the truncation to have effects; here $M_t$ is the total number of samples seen after task $t$. This is why, in our full implementation, Algorithm 4 of Appendix C (Line 1098), we have an extra hyperparameter $r\_{max}\in[0,E)$, which denotes the *maximum allowable rank*, and we set $$k_t\gets \min (r\_{max}, (1- \zeta) M_t),$$ We apologize for this not being clear in Algorithm 1.

- **Reviewer P9vj** is concerned that a large $E$ would make *the proposed method infeasible for most datasets*. This would indeed be an issue for other methods, but not for ours. We showed our method is feasible and fast on multiple datasets with large $E$, as it sets $k\_{t-1}+m\_t$ small and has a linear time complexity in $E$.

---

### Author Response · Authors · 2024-11-13
**Our Theoretical Contributions to CL with Pre-trained Models: Three-stage Interrogation**

We consider a 3-stage interrogation of our theoretical contributions. By doing so, we hope the reviewers could appreciate our perspectives and theoretical results a little bit more.

# 1. Status of the field (continual learning)
Here we rehearse what we wrote in Lines 054-059:
- Many methods have recently been developed for CL with pre-trained models (cf. Table 1). **However, almost all these methods in Table 1 are purely empirical, and there is little theory derived in their paper.** These include RanPAC and its follow-up works (Ahrens et al. TMLR 2024) and (Prabhu et al. NeurIPS 2024).

- On the other hand, **theoretical studies on CL have been focused on algorithms or models that have little to no implications in practical CL scenarios such as class-incremental learning**. These include Evron et al. (COLT 2022), Lin et al. (ICML 2023),  and Peng et al. (ICML 2023).

# 2. Significance of our theory

Given the current status of the field, and while many CL papers provide theorems that do not match the algorithms that are in practical use, **we believe our theoretical results are the first of their kind in the field, in the sense that they apply precisely to the proposed CL algorithm, which has strong performance at the same time**.  This reduces the gap between theory and algorithms. We believe this is an important conceptual contribution and will inspire follow-up works to consider similar frameworks to develop empirically strong CL methods with guarantees.

Furthermore, we claim that our theory is novel, useful, and of sufficient technical depth.

- **Our theory is novel** in light of related works even in the statistics community (in Remarks 4 and 6). To our knowledge, **our theory is the first such result that analyzes the errors of incremental truncated SVD**, even though the basic techniques of incremental SVD have been known for decades since Bunch & Nielsen (1978) (cf. Appendix C).
- **Our theory is useful** as the upper bound is small if we truncate suitably. This can be understood from Section 5 and all the figures.
- **One theory is of sufficient technical depth**, as we analyze the effects of continual truncation, while prior published works in the theory community truncate only once (the offline case); cf. Remarks 4 and 6.

While **Reviewer P9vj** was *not convinced that the analysis makes the paper valuable enough for ICLR*,  and while we appreciate the very high standard **Reviewer P9vj** holds for theoretical works, we should note that our work is not purely theoretical, and we consider our algorithmic contributions to be even more important, as we will justify in the next post (*Our Algorithmic Contributions and Improvements in Comparison to RanPAC*).

# 3. Limitations of our theory
As the reviewers acutely pointed out, our theorems have limitations, e.g., the bounds might be loose; the linear model assumptions might be restrictive. We will address them in our responses to individual reviewers. Here, in order to better appreciate these limitations, we clarify why our theorems are as such and the rationale behind them:
- We have a method that truncates SVDs continually, so **we want to analyze the effects of continual truncation**. However, such analysis in its full generality is a formidable task, as the full network architecture is complicated: the input images have a complicated distribution; they are fed to a pre-trained model to yield pre-trained features $X\_{1:t}$, which is lifted to random ReLU features $H\_{1:t}$; our method continually truncates $H\_{1:t}$.
- Since we don't know the distribution of $X\_{1:t}$, any randomness assumptions on $X\_{1:t}$ would sound unrealistic to practitioners. Hence, we want to derive theorems with $X\_{1:t}$ being deterministic, even though a rich body of theory work needs Gaussian-type assumptions on $X\_{1:t}$ or $H\_{1:t}$ (cf. Remarks 4 and 6). Crucially, we find we can do the analysis under a linear model assumption conditioning on $H\_{1:t}$. While the linear model assumption might be considered limited, note that:
    - **doing so achieves our goal, as the resulting theorems do reveal the effects of continual truncation**. Specifically, the role of continual truncation is made explicit via two quantities, the accumulative error $a\_t$ and eigengap $\gamma\_t$. Moreover, we can understand the behaviors of $a\_t$ and $\gamma\_t$ via the spectrum of (truncated) random ReLU features, which we visualized and analyzed.
    - **the roles of pre-trained models, random embedding, and nonlinearity are also captured by our theory (implicitly)**, as they affect the spectrum of (truncated) random ReLU features and thus the values of $a\_t$ and $\gamma\_t$.
    - **it puts no restrictions on our method**. Rather, it *predicts* the errors our method could produce. This is important, as the prior work, RanPAC, is unpredictable and unstable.

---

### Author Response · Authors · 2024-11-14
**Summary of Reviews and Our Rebuttal**

We thank all reviewers for their time in reading the manuscript and writing comments.

First of all we are happy to read that the reviewers commented on the following strengths of our paper:
- Experiments are extensive, comprehensive, or significant (**Reviewer EXjv**, **Reviewer XHQX**, **Reviewer iVyo**)
- The writing is clear and easy to follow (**Reviewer XHQX**, **Reviewer iVyo**). All reviewers scored *3 (good)* for *Presentation*.

We received multiple comments on weaknesses from the reviewers, and we also received multiple questions. To address them, we write our rebuttal that consists of:
- a statement at the end of this post, identifying our non-conventional, “bridging fields" contributions;
- one post that clarifies our theoretical contributions (*Our Theoretical Contributions to CL with Pre-trained Models: Three-stage Interrogation*);
- one post that clarifies our algorithmic contributions (*Our Algorithmic Contributions and Improvements in Comparison to RanPAC*);
- posts to individual reviewers addressing their specific concerns;
- a revised version of our paper based on the comments. We made minimum modifications to the original submission (from the main paper to Appendix K). All revisions and new contents needed to address reviewers' concerns are in Appendix L. We believe this would make our revisions more accessible, and can be easily compared to the original version. Minor comments, such as grammar or typos, will be fully addressed later (e.g., after the decisions are out).

We are glad that most comments are constructive and can be addressed to their full extent with our best efforts. Resolving some other comments does require interacting with the reviewers and relying on their active participation in the discussion phase. Given the current divergent scores, and given our belief in the quality of our work, we would be grateful if the reviewers could join the discussion, and if **ACs** could take a look at the discussions, our rebuttal, and our manuscript (especially, the intro, Fig 5, Table 2, Table 16).


# Non-conventional, “bridging fields" contributions

This year, AISTATS helps the reviewers identify *non-conventional contributions* by the following questions:
> *Does the submission contain non-conventional research contributions? Examples: novel ideas with not widely accepted assumptions, new problems and/or tasks, “bridging fields” contributions.*

We bring it to the attention of our ICLR reviewers, and state our non-conventional, “bridging fields" contributions that we indicated in the title (as they are often overlooked):
- Our paper bridges the gap between the theoretical studies and empirical methods in continual learning;
- Our paper connects continual learning to recent theoretical studies on random feature models and overparameterized linear models (cf. Remarks 1, 4, 6, and Appendix H.2). Given this connection, we believe existing theory in random feature models and overparameterized linear models will play an important role in enhancing theoretical foundations of continual learning. Conversely, as **Reviewer XHQX** recognizes, our theory can also be viewed as advances in principal component regression, an important topic in statistics.

---

> ### Author Response · Authors · 2024-11-28
> **Happy Thanksgiving**
>
> It is right before Thanksgiving, and we take the opportunity to say thank you to all reviewers for their intellectual input. We also want to express our special gratitude to **Reviewer XHQX** and **Reviewer P9vj** for participating in the discussion and supporting acceptance.
>
> It is also in the middle of the discussion phase, so we want to have a quick summary of where we are now:
> - **Reviewer XHQX** replied to our rebuttal and suggested that the concern on experiments was addressed. We believe the remaining concerns are somehow minor (but if it is not, please let us know!). For example, regarding the confusion on *double descent*, we have addressed it by revising our manuscript and removing related contents; in retrospect, we just wanted to say that RanPAC could have large test losses yet small training losses. Finally, we refer the reviewer to Appendix L.4, in which we aim to address the comments on continual PCA to its full extent.
> - **Reviewer P9vj** revised the score from 3 to 6. We highly appreciate it. After discussion, we are glad that **Reviewer P9vj** finds our theory *more elegant* and our experiments *an empirical success*. We believe the main concerns have been addressed, and we promise to make the necessary revisions that would improve the paper (See Appendix L.4).
> - While **Reviewer EXjv**  and **Reviewer iVyo** haven't engaged with us during the discussion phase, this is understandable. For comments of **Reviewer iVyo**, we have derived a new theorem in Appendix L.1, and we have run new experiments in Appendix L.2. For comments of **Reviewer EXjv**, we have made a justification about experience replay methods. We believe we have also addressed their comments to the fullest extent; if that is inaccurate, please feel free to let us know.

---

> ### Public Comment · ~Jiao_Chen3 · 2025-02-15
>
> I truly appreciate your insightful work on this topic. It might be helpful to reference [1] in the camera-ready version to further enrich the literature review. Thank you for considering this suggestion.
>
> [1] ACIL: Analytic class-incremental learning with absolute memorization and privacy protection. NeurIPS 2022.

---

> ### Public Comment · ~Liangzu_Peng2 · 2025-02-15
> **Reply**
>
> We appreciate your comments.
>
> Please find the paper you mentioned cited and discussed at page 21 of the manuscript; see the paragraph at line 1122.
>
> Before submission, we implemented an RLS type method using the Woodbury matrix identity to invert the regularized covariance (which is a generalized version of acil). It didn't work well due to numerical instability, a known issue of the matrix inverse formula.
>
> We will further talk about it in the main paper of our camera ready, in a unifying manner. Please stay tuned.

---

### Public Comment · ~Xiaoyu_Sun6 · 2025-12-31
**About the title difference between the paper and BibTex**

Dear authors:
Thank you for your inspiring work! I was copying the bibtex provided and found the title in bibtex "TSVD: Bridging Theory and Practice in Continual Learning with Pre-trained Models" is different from the paper title: "LoRanPAC: Low-rank Random Features and Pre-trained Models for Bridging Theory and Practice in Continual Learning". I wonder is this bibtex valid or I need to change the title field to LoRanPAC title? Thank you very much for your reply.

---

> ### Public Comment · ~Rene_Vidal1 · 2026-01-02
> **Re: About the title difference between the paper and BibTex**
>
> Please use "LoRanPAC: Low-rank Random Features and Pre-trained Models for Bridging Theory and Practice in Continual Learning"

---

### Meta-Review · Area_Chair_gr2U · 2024-12-20

**Metareview:**

This paper combines RanPAC into the Ideal Continual Learner (ICL) framework, providing theoretical results and experiments. The new method (and its empirical performance), along with theoretical results, are key contributions of the paper.

Reviewers were generally more positive for this paper after rebuttal than before. Key strengths are that the method is well-motivated, the paper is well-written, and the experiments are extensive. The experiments are particularly impressive given the number of pages devoted to theory in the paper; perhaps the authors could consider giving more space to the results.

There are key questions regarding the theoretical contribution of this work, however. Reviewer XHQX is missing comparison to online PCA algorithms: the authors should clearly compare their theoretical analysis to existing work in this space (such as the PCA-OGD algorithm that has been discussed already). Reviewer P9vj does not find the theoretical results very useful, and notes that the authors have promised many changes to the paper. My own estimation of the theoretical contribution is somewhere between the authors' and the reviewers'. Overall, given the lack of such results in continual learning, I think the field would benefit from this paper.

**Additional Comments On Reviewer Discussion:**

I am ignoring Reviewer EXjv as, like they said, they do not have the expertise to properly review this paper. Reviewer iVyo wrote a detailed review, but did not engage in a detailed discussion after. In my view, the authors' response sufficiently addressed most points, and the paper will be stronger as a result.

Various questions and concerns were brought up about the theoretical contributions. The authors addressed many of the questions, and it looks like this should improve the quality of the writing. Concerns about the significance of the theory remain, but overall I think the field would benefit from this theoretical analysis.

The authors have promised a number of clarifications and changes during the discussion period. I strongly suggest that the authors put some time to integrate these into the main paper, and not just the Appendix.

---

### Decision · Program_Chairs · 2025-01-22

Accept (Poster)